# *Decomposed Prompting*: A MODULAR APPROACH FOR SOLVING COMPLEX TASKS

**Tushar Khot♣, Harsh Trivedi♡, Matthew Finlayson♣, Yao Fu♠,**\*
**Kyle Richardson♣, Peter Clark♣, Ashish Sabharwal♣**
♣Allen Institute for AI       ♡Stony Brook University       ♠University of Edinburgh
tushark@allenai.org, hjtrivedi@cs.stonybrook.edu, matthewf@allenai.org, yao.fu@ed.ac.uk,
kyler@allenai.org, peterc@allenai.org, ashishs@allenai.org

## ABSTRACT

Few-shot prompting is a surprisingly powerful way to use Large Language Models (LLMs) to solve various tasks. However, this approach struggles as the task complexity increases or when the individual reasoning steps of the task themselves are hard to learn, especially when embedded in more complex tasks. To address this, we propose Decomposed Prompting, a new approach to solve complex tasks by decomposing them (via prompting) into simpler sub-tasks that can be delegated to a shared library of prompting-based LLMs dedicated to these sub-tasks. This modular structure allows each prompt to be optimized for its specific sub-task, further decomposed if necessary, and even easily replaced with more effective prompts, trained models, or symbolic functions if desired.

We show that the flexibility and modularity of Decomposed Prompting allows it to outperform prior work on few-shot prompting using GPT-3. On symbolic reasoning tasks, we can further decompose sub-tasks that are hard for LLMs into even simpler solvable sub-tasks. When the complexity comes from the input length, we can recursively decompose the task into the same task but with smaller inputs. We also evaluate our approach on textual multi-step reasoning tasks: on long-context multi-hop QA, we can more effectively teach the sub-tasks via our separate sub-tasks prompts; and on open-domain multi-hop QA, we can easily incorporate a symbolic information retrieval module within our decomposition framework, leading to improved performance on both tasks.[1]

## 1 INTRODUCTION

Large Language Models (LLMs) such as GPT-3 (Brown et al., 2020) have been shown to solve various tasks given only a few examples as prompts, also referred to as in-context learning. These models can even perform more complex reasoning tasks when shown the sequence of simple reasoning steps needed to perform the complex task as a prompt (Wei et al., 2022; Nye et al., 2021). In essence, the sequence of reasoning steps, such as in Chains-of-Thought (CoT) prompting (Wei et al., 2022), demonstrates how to decompose the complex task as well as how each reasoning step should be performed. However, as tasks become more complex, few demonstrations of the complex task aren't sufficient for current models to learn to perform all necessary reasoning steps. E.g., few-shot demonstrations of concatenating the $k^{\text{th}}$ letter of words in a string is insufficient for GPT-3 to learn to extract the $k^{\text{th}}$ letter, or learn to answer hard single-hop questions when only provided a few demonstrations of multi-hop questions. Additionally, it is unclear whether tasks such as document retrieval and integration, for knowledge-intensive tasks, can even be done by few-shot prompts.

To address these limitations, we propose **Decomposed Prompting** (DECOMP), a new approach to solve complex tasks by instead decomposing them into simpler sub-tasks and delegating these to sub-task specific LLMs, with both the decomposer and the sub-task LLMs (henceforth, *sub-task handlers*) having their own few-shot prompts. Fig 1 illustrates our approach. The decomposer

---

\*Work done during internship at Allen Institute for AI
[1]Datasets, Code and Prompts available at `https://github.com/allenai/DecomP`.

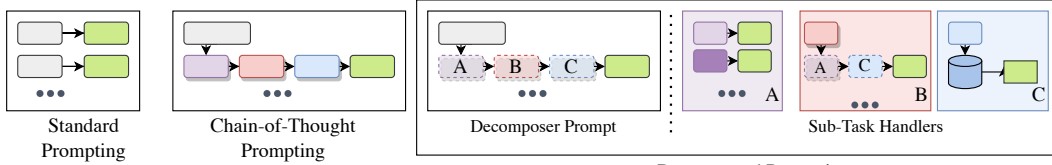

Figure 1: While standard approaches only provide labeled examples (shown as a grey input box with green label box), Chain-of-Thought prompting also describes the reasoning steps to arrive at the answer for every example in the prompt. Decomposed Prompting, on the other hand, uses the decomposer prompt to only describe the procedure to solve the complex tasks using certain sub-tasks. Each sub-task, indicated here with A, B and C is handled by sub-task specific handlers which can vary from a standard prompt (sub-task A), a further decomposed prompt (sub-task B) or a symbolic function such as retrieval (sub-task C)

prompt only describes a sequence of sub-tasks (A, B, and C) needed to solve the complex tasks, indicated with the dashed lines. Each sub-task is then delegated to the corresponding sub-task handler shown on the right.

Using a software engineering analogy, the decomposer defines the top-level *program* for the complex task using interfaces to simpler, sub-task functions. The sub-task handlers serve as modular, debuggable, and upgradable *implementations* of these simpler functions, akin to a software library. If a particular sub-task handler, say the one for identifying the $k^{\text{th}}$ letter or retrieving a document, is not performing well enough, we can debug this handler in isolation, explore alternative prompts or implementations, and seamlessly plug the improved module back into the overall system, as a systematic way to try to improve performance on the complex end-task.

This approach has several advantages over prior work (as also shown in the figure). The sub-task handlers can be shown a broader and richer set of examples (of the simpler task) than the specific ones needed for the complex task prompt (task A). If a sub-task is too complex, it can be further decomposed into simpler sub-tasks (task B). Similar to software libraries, these sub-task handlers can be shared across multiple tasks; e.g., here tasks A and C are reused in the model for task B. As noted above, a sub-task handler can be easily swapped with an improved implementation without any change to the rest of the system. Few-shot prompt based LLMs can be even replaced with a symbolic system for tasks more suited for non-neural methods; e.g., task C uses a symbolic retrieval system such as Elasticsearch that can handle very large-scale corpora. Lastly, we can even improve upon prior work by simply adding an *error-correcting* sub-task handler as a post-processing step.

To illustrate these advantages of DECOMP, we empirically evaluate it against prior work on eight challenging datasets using GPT3 models: (1) On a task of concatenating the $k^{\text{th}}$ letter, we show that our approach of factoring out each sub-task allows us to more effectively teach the sub-problem of extracting the $k^{\text{th}}$ letter(specifically, by decomposing it into even easier sub-tasks). (2) On a task of reversing a list, we show that DECOMP allows us to extend the capabilities of a weaker model and build a scale-invariant system by recursively decomposing the task into reversal of smaller and smaller lists. (3) On a task of long-context QA (Khot et al., 2022), our approach allows each sub-task handler to accommodate more examples than feasible with CoT prompting leading to better QA performance. (4) On three multi-hop open-domain QA datasets (Yang et al., 2018; Ho et al., 2020; Trivedi et al., 2022), we can incorporate a symbolic retrieval (ElasticSearch) API as the handler for the retrieval sub-task leading to better results than CoT. (5) On two Math QA datasets (Cobbe et al., 2021; Roy & Roth, 2015), we can post-process CoT to easily fix frequent formatting errors, resulting in a surprisingly high improvement of 14-17 pts.

## 2 RELATED WORK

**Few-shot Prompts for Multi-Step Reasoning** Large-scale Language models (LLMs) have been shown to learn various NLP tasks given just few examples as prompts (Brown et al., 2020). Recently, they have also been successfully applied to various multi-step reasoning tasks by providing the intermediate reasoning steps, i.e. Chain-of-Thought (Wei et al., 2022; Chowdhery et al., 2022), needed to arrive at the answer. An alternate approach has been to compose multiple LLMs or LLMs with symbolic functions to perform multi-step reasoning (Jung et al., 2022; Creswell et al., 2023;

Press et al., 2022; Parisi et al., 2022; Gao et al., 2022; Schick et al., 2023, inter alia). We view these prior works as specialized systems with a pre-defined decomposition structure.

The closest works to our approach are the ideas of least-to-most prompting (Zhou et al., 2023) and successive prompting (Dua et al., 2022) where one prompt/model is used to generate the sub-questions needed to answer a complex question and a second prompt/model sequentially answers these sub-questions. In contrast, our approach allows for diverse decomposition structures including recursion and other non-linear decomposition structures. E.g., by definition, least-to-most asks questions from easiest to the hardest and requires an LLM to eventually answer the complete question ("most" in least-to-most) whereas we have no such restriction. Additionally, we iteratively generate new questions based on previous answers (similar to successive prompting) and can explicitly assign different prompts or symbolic systems to answer each sub-question.

**Modular Approaches for Multi-Step Reasoning** Our work follows a long literature in NLP on neural modular modeling architectures (Andreas et al., 2016; Talmor & Berant, 2018; Min et al., 2019; Jiang & Bansal, 2019; Gupta et al., 2020; Perez et al., 2020; Khot et al., 2021; Levine et al., 2022) for question-answering and other tasks. We take particular inspiration from the *Text Modular Networks* approach of Khot et al. (2021), whereby problem decomposition consists of a learned *next question* generator trained to generate questions in the language of a collection of textual and symbolic agents. Best-first search strategy was used to explore the space of possible decompositions during inference. In contrast to this work, which largely centered around supervised training of the next-question generator *given existing agents*, we leverage the power and recent successes of few-shot LLMs to build both the decomposer and the sub-task agents that best fit the ideal decomposition. This has the advantage of obviating the need for specialized supervised training data that may not always be available for all sub-tasks – a key bottleneck of this prior work.

## 3 Decomposed Prompting

As with conventional *few-shot* prompting, the goal is to teach an LLM to find an answer $A$ to a query $Q$ using a small set of *in-context* examples $D = \{E_1, ..., E_{|D|}\}$. The answer $A$ is obtained from the underlying distribution $p(A \mid Q, D, \theta)$ (Dohan et al., 2022). In the most basic few-shot setup, examples take the form $E_j = (Q_j, A_j)$. In the case of CoT-style prompting, the goal is to obtain answers by first generating a sequence or chain of intermediate reasoning steps or "thoughts" $T$, and then deriving the final answer based on $T$. To teach this ability, one uses more sophisticated in-context examples that take the form $E_j = (Q_j, (T_{j,1}, \ldots, T_{j,k}), A_j)$.

In Decomp, the core is a *decomposer* LLM that tries to solve a complex task by generating a **prompting program** $P$ for it. Each step of $P$ directs a simpler sub-query to a function in an auxiliary set of **sub-task functions** $\mathcal{F}$ available to the system. Given a query $Q$ whose answer is $A$, the program $P$ is a sequence of the form $\big((f_1, Q_1, A_1), ..., (f_k, Q_k, A_k)\big)$ where $A_k$ is the final answer predicted by $P$ and $Q_i$ is a sub-query directed to the sub-task function $f_i \in \mathcal{F}$. $P$ is executed by a high-level imperative **controller**, which passes the inputs and outputs between the decomposer and sub-task handler until a stopping condition in $P$ is met and the final output obtained.

To teach the decomposer LLM in a few-shot prompting manner, we use in-context examples that take the form $E_j = \big((Q_j, \big((f_{j,1}, Q_{j,1}, A_{j,1}), ..., (f_{j,k_j}, Q_{j,k_j}, A_{j,k_j})\big)\big)$ where $A_{j,k_j} = A_j$ is the final answer for $Q_j$ and $(Q_{j,1}, \ldots, Q_{j,k_j})$ is a decomposition of $Q_j$. Each sub-task function $f$, in turn, is operationalized via a sub-task handler as an in-context prompting LLM (e.g., a separate CoT-style prompt or a additional prompting program dedicated to that sub-task), or any other symbolic or learned function (e.g., a calculator or specialized supervised trained model).

### 3.1 Decomposed Prompts

To illustrate this with an example, consider a multi-step task such as "Concatenate the first letter of every word in $str$ using a space". We can solve this task by decomposing it into a sequence of three simple sub-tasks: 1) Collect the list of words in the $str$; 2) For each word, extract the third letter; 3) Concatenate the extracted letters using space as the separator. Fig. 2 shows an example decomposition prompt for this task. Much like a conventional structured program, the top-level `decomp` prompt provides an example program $E_j$ using three sub-task functions: $f_1$ `:split` that

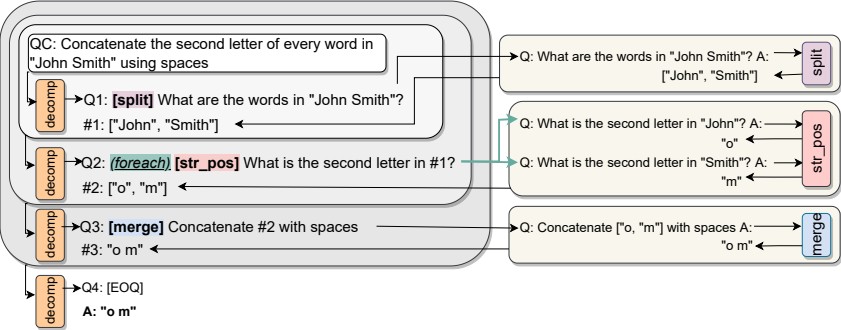

Figure 2: Prompts for the decomposer and the `split` and `merge` sub-tasks used by the decomposer. The decomposer specifies the sequence of questions and corresponding sub-tasks (within square braces). The sub-task prompts can be written independent of the complex task examples and can even capture generalizations, e.g., letters in word (`split`) and no delimiter (`merge`).

*splits words in an input string*, $f_2$ :`str_pos` that *finds character positions in strings* and $f_3$ :`merge` that *concatenates characters*. In this case, we operationalize each sub-task function as a separate in-context prompt (e.g., using a standard prompting approach for `split` and `merge` on the right side), each containing a set of in-context examples that are independent of the original complex task.

In addition to the three functions described above, additional control structure is included, such as the symbolic function `foreach`, which iterates over arrays and references to previous answers such as #1. We note that such a helper function is not strictly necessary (e.g., we could directly generate "Q2': What is the first letter of Jack?" and "Q3': What is the first letter of Ryan?" instead of Q2 in the figure) and is added to reduce the manual effort needed to specify the decomposition and also reduce potential errors during decomposition. In our experiments we use two of the compositional operators defined by Khot et al. (2022) (see appendix for details), although it is capable of using all their operators (which also capture the QDMR operators from Wolfson et al. (2020)).

Figure 3: The inference procedure in DECOMP iteratively calls the decomposer prompt to generate the next question and sub-task at each step, given the current history of question and answers. The generated question is then routed to the assigned sub-task handler (with some handling of special operators, when needed). When the special end-of-questions [EOQ] marker is generated, the previous answer is returned as the final prediction.

## 3.2 PROMPT EXECUTION AND INFERENCE

Given a new question and a set of background in-context examples $D$, the inference (i.e., the program construction and execution) process is illustrated in Fig. 3. The new complex question is fed to the decomposer prompt to get the first sub-question to be asked to the `split` prompt. With the help of our symbolic controller, the answer generated from this prompt is then appended to the decomposer prompt to get the second sub-question, $Q2$. Due to the `foreach` operator in the generated question, $Q2$ results in two questions (one for each word in #1) to be fed to the `str_pos` prompt. The answers are combined into an array to get the answer #2. The entire decomposition history is used to generate $Q3$ and passed to the `merge` prompt to get the final answer. Since the task has been solved, the decomposition prompt produces the special end-of-sequence marker([EOQ]) and the last answer is returned as the final answer. Formally, performing inference involves finding the best answer $A$ to a new query $Q$, which in the simplest form involves computing the MAP answer using the

LLMs predictive distribution for $A$, i.e., $\hat{A} = \arg\max_A p(A \mid D, Q, \theta)$ (Dohan et al., 2022). For practicality, such computations are approximated using greedy search in our experiments.

### 3.3 DECOMP CAPABILITIES

**Hierarchical Decomposition** Certain sub-tasks, even when given many examples, are not solvable with few-shot prompting. E.g., we found identifying the $k^{\text{th}}$ letter of a string to be challenging for the GPT3 `text-davinci-002` model. In such a scenario, we can decompose the sub-task prompt further, to first identify the letters and their position and then select the $k^{\text{th}}$ element of this array (see Fig. 4). We can also re-use existing sub-task prompts in our framework. E.g., the `split` prompt can be reused since it was developed for the general task of splitting strings.[2]

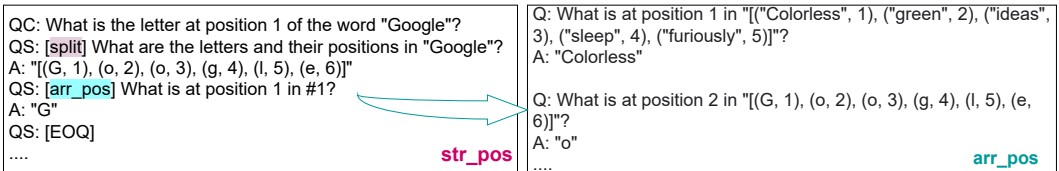

Figure 4: Since identifying the $k^{th}$ character is challenging for GPT3 `davinci-002` model, we further decompose it into two simpler sub-tasks: split the word into its letters (using the shared sub-task `split`) and then return the $k^{th}$ item of this list using the `arr_pos` prompt.

**Recursive Decomposition** Some problems can be naturally broken down into one or more smaller problems of the same form. Recursive algorithms such as merge sort use this idea to solve large problems efficiently, using a succinctly described method. We apply this same principle in DECOMP by allowing the decomposer prompt to recursively call itself, as shown in Fig. 5 for the task of list reversal. By using recursion, we are able to generalize any base prompting approach (CoT in this figure) to much longer lists by breaking the input into smaller and smaller lists till we reach a list length where the model is highly accurate. Such recursive approaches can not be described by current methods such as CoT and standard prompting. Least-to-most prompting (Zhou et al., 2023) also proposes a similar solution but differs in two key aspects (a) it has to identify all the sub-problems in one-shot instead of our iterative top-down decomposition (b) it has to learn to identify the relevant answers from the previous solutions which we get for free from our decomposition.

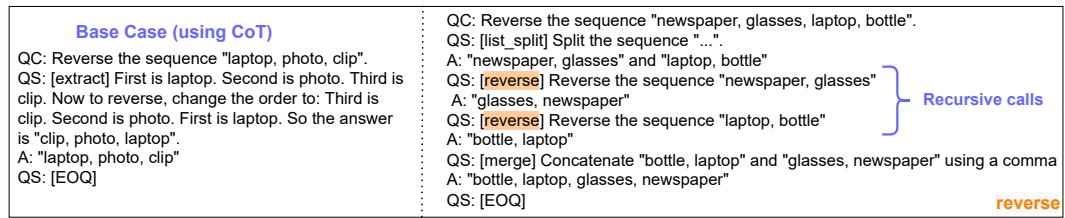

Figure 5: Sample prompt for recursive decomposition for reversing lists. Each list is split into two halves and each half is reversed and concatenated in the reverse order. We can recursively split a list till we hit the base case (lists of length 3 here) where existing approaches such as CoT are accurate.

**External API Calls** In certain cases, the sub-tasks may not be feasible to solve using only a LLM. E.g., retrieving knowledge from a KB or large corpus. Such sub-tasks, however, can be easily solved using existing systems such as retrieving documents using an Elasticsearch index or webpages using Google search (Lazaridou et al., 2022). Fig. 6 shows how DECOMP can easily use such a system to retrieve the relevant documents and answer a single-hop open-domain question.

---

[2]Appendix G contains the complete `split` prompt which has examples for questions such as "Q: What are the letters and their positions in "Mathison"?".

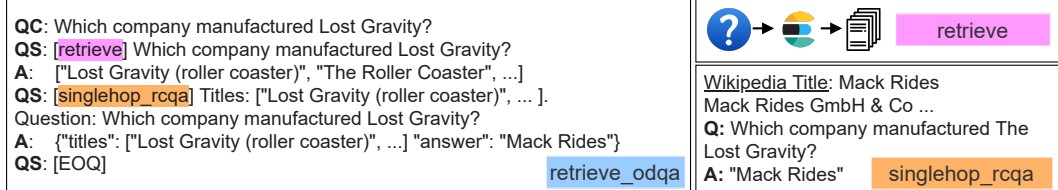

Figure 6: A Decomposed Prompt to answer open-domain questions using Elasticsearch-based retrieval. Full usage of this prompt for open-domain multihop questions is given in Fig. 11.

## 4    CASE STUDIES

We showcase DECOMP's strengths through four tasks; two symbolic manipulation tasks similar to those investigated by Wei et al. (2022) and two existing textual multi-hop reasoning tasks. Unless specified, we use `text-davinci-002` InstructGPT3 model (Ouyang et al., 2022) as the LLM and report the Exact Match (EM) numbers, following prior work. For order-independent list answers, we evaluate set equality as EM. We compare our approach to CoT rather than each specific decomposition structure used in prior work. See App. G for the complete prompts for all our tasks.

### 4.1    $k^{\text{th}}$ LETTER CONCATENATION (HIERARCHICAL DECOMPOSITION)

We compare DECOMP to CoT prompting for concatenating letters at the $k^{\text{th}}$ position. All prompts contain examples of concatenating letters in position 1, 4, and last position of strings with 3 words. We create three different prompts for all our baselines and present the average to account for variance due to the choice of examples following Perez et al. (2021). We use the `decomp`, `split`, `str_pos` (further decomposed as shown in Fig. 4), and `merge` prompts for decomposition prompting. We adapt the CoT for last letter concatenation from prior work (Wei et al., 2022) for this task as shown below. In addition, we consider a *rolled out* version of our decomposition prompts in terms of a CoT, i.e., we describe the entire decomposition process (identify words, split each word into letters, take $k^{\text{th}}$ letter and concatenate) as a single CoT. e.g, for the question "Take the letters at position 4 of the words in "Herbert Alexander Simon" and concatenate them using a space.", we use the CoT:

| **Chain-Of-Thought** | **Chain-Of-Thought (rolled out)** |
|---|---|
| The letter at position 4 of "Herbert" is "b". The letter at position 4 of "Alexander" is "x". The letter at position 4 of "Simon" is "o". Concatenating "b", "x", "o" using a space leads to "b x o". So, "Herbert Alexander Simon" outputs "b x o". ... | The words in "Herbert Alexander Simon" are "Herbert", "Alexander", and "Simon". The letters and their positions in "Herbert" are "[(H, 1), (e, 2), (r, 3), (b, 4), (e, 5), (r, 6), (t, 7)]". The letter at position 4 in this sequence is "b". ⋯ outputs "b x o". ... |

We similarly adapt the least-to-most prompt (Zhou et al., 2023) to include rollout. (see App. G). We compare these four prompting techniques on 4 datasets to evaluate generalization along 3 axes: (1) new letter position $k = 3$;[3] (2) longer inputs, #words=4 and 5; (3) new delimiter ";". The words in the test examples come from a list of most popular first and last names.[4] All evaluation datasets have 100 examples. We present results on space as a delimiter averaged across three prompts in Fig. 7.[5]

**DECOMP outperforms chain-of-thought and least-to-most prompting**, even when the prompt uses the same reasoning procedure as the rolled out decomposition. This shows that the separate prompts are more effective at teaching hard sub-tasks than a single CoT prompt.

**DECOMP generalizes perfectly to longer sequences.** As the length of the input sequence increases, our approach continues to achieve close to 100% accuracy on this task.[6] The CoT-based approaches drop noticeably in their scores with longer input lengths, widening the performance gap.

---

[3]Note that none of the sub-task prompts contain examples for this position.

[4]`forebears.io/earth/forenames` and `forebears.io/earth/surnames`

[5]We obtain similar results with semi-colon shown in Fig. 22 in the appendix.

[6]Note that we report aggregate metrics for DECOMP too but the std. dev is zero here

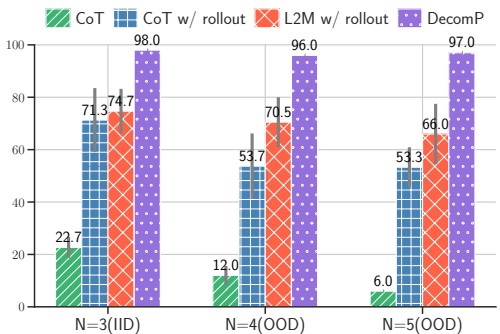

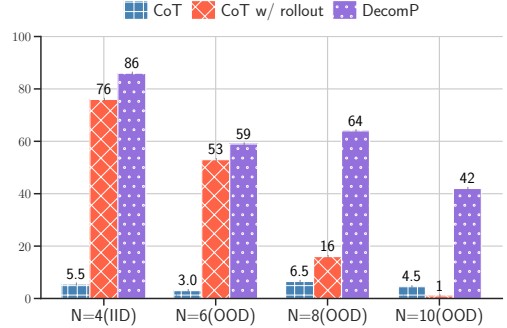

Figure 7: EM Results on the $k^{th}$ letter concatenation task (k=3) using space as delimiter with different number of words in the input. DE-COMP outperforms and generalizes better than CoT as well as Least-to-most prompting.

Figure 8: EM results on reversing sequences. Incorporating CoT in DECOMP greatly increases the ability of the model to generalize to new sequence lengths.

## 4.2 LIST REVERSAL (RECURSIVE DECOMPOSITION)

We use the task of reversing lists of words[7] to show how recursive DECOMP enables length generalization. We adapt the relevant CoT prompt from Wei et al. (2022), and integrate it in a decomposed prompt. As a control, we also compare to a CoT version w/ rollout of our decomposed prompt. All prompts contain the same 3 examples of reversing word sequences with 3-5 items. We evaluate all prompts for generalization to 4, 6, 8, and 10-item sequences. Here we use `davinci-001` to show that DECOMP enables a weaker model approach `davinci-002`'s performance (which does solve this task). We use the strategy from Fig. 5 and provide our prompts in App. G. Fig. 8 shows the results of the prompting strategies on different input lengths.

**DECOMP improves the length generalization of few-shot prompting.** While our base CoT prompt does not generalize at all to longer sequences, our approach can recursively decompose the problem and achieve better length generalization. Moreover, the CoT version of our decomposition strategy fails because the unrolled prompt becomes too long and convoluted without the ability to abstract away sub-modules.

## 4.3 LONG-CONTEXT QUESTION ANSWERING

We next evaluate on the CommaQA-E dataset (Khot et al., 2022) under the reading comprehension setting. The dataset consists of synthetically generated entities (e.g. Erowid award), facts ("Wetherality was an actor in the movie Dewbar.") and multi-hop questions (e.g., "What awards have the actors of the Erowid winning movies received?"). Due to the presence of many distractors and, as a result, longer context, this dataset has been shown to be hard for standard LMs even when fine-tuned.

```
What awards have movies produced by people born in 1910 won?
QS: [qa] Who were born in the year 1910?
A: ["Teeplemole", "Muntaril"]
QS: (foreach_merge) [qa] For which movies was #1 the producer?
A: ["Featsaw", "Zalate", "Premercy"]
QS: (foreach_merge) [qa] Which awards were given to #2?
A: ["Zorgion", "Chowwurst", "Hallowcock"]
QS: [EOQ]
...                                                    coarse
```
```
What awards have movies produced by people born in 1910 won?
QS: [simp_qa] Who were born in the year 1910?
A: ["Teeplemole", "Muntaril"]
QS: (foreach_merge) [pos_qa] For which movies was #1 the producer?
A: ["Featsaw", "Zalate", "Premercy"]
QS: (foreach_merge) [aw_qa] Which awards were given to #2?
A: ["Zorgion", "Chowwurst", "Hallowcock"]
QS: [EOQ]
...                                                        fine
```

Figure 9: Sample prompts used for the CommaQA dataset. On the left, the coarse-grained decomposition defines a single QA sub-task with all single-hop questions being delegated to a single sub-task handler. On the right, the fine-grained decomposition assigns questions to three different sub-tasks (see App. G for their prompts) depending on the question type. This allows us to provide more examples for each question type allowing the model to learn the sub-task more effectively.

To fit these questions within GPT3's context limit (2049 tokens), we generate a smaller version of the CommaQA-E dataset and of the compositional generalization split such that we can fit at least

---

[7]We use the vocabulary from Wei et al. (2022): `https://www.vocabulary.com/lists/189583`

four examples in the context for CoT prompts. The CoT prompts describe the sequence of facts needed to arrive at the answer (see App. G for all the prompts).

For DECOMP, we can separate the task of decomposition (independent of the context) from the sub-tasks of single-hop question answering. As shown in Fig. 9, we provide examples of the context-independent decomposition in the decomposer prompt and use the separate sub-task prompts to teach the QA skill over the given context. Additionally, we can choose the granularity of decomposition to trade off human effort for increased accuracy. For example, we could have single QA prompt to handle all the questions or create QA prompts for different classes of questions. In our experiments, each sub-task prompt contains 8 QA examples (2 questions/para). We evaluate three different prompts and report the average results in Fig. 10.

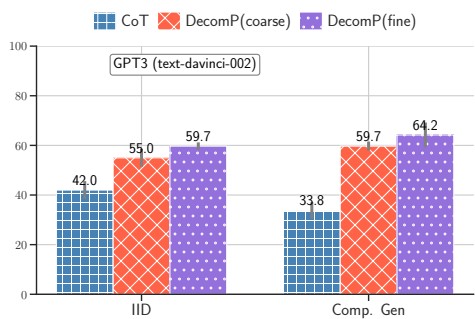

Figure 10: EM results on the CommaQA-E datasets. DECOMP always outperforms CoT, with fine-grained marginally out-performing coarse-grained decomposition.

We make three observations on CommaQA. **DECOMP is more accurate than CoT** irrespective of the granularity of decomposition or the evaluation split. **Finer grained decomposition can help improve task performance** by providing more examples for each class of questions, which in turn increases single-hop QA accuracy. **DECOMP generalizes to new compositions** such as the compositional generalization split of CommaQA, which tests models on unseen compositions of relations observed in the training set. While CoT has a drop in score, both decomposition-based approaches actually get a small bump (the subset of relations used in this split are easier for our QA models).

### 4.4 OPEN-DOMAIN QUESTION ANSWERING

Next, we demonstrate the ability of our approach to integrate external API calls on the task of open-domain multihop question answering. We evaluate our approach on three datasets: (1) 2Wiki-MultihopQA (Ho et al., 2020) (2) MuSiQue (Trivedi et al., 2022) (3) HotpotQA (Yang et al., 2018). We describe the open-domain versions of these datasets in more detail in App. A We use the Codex (code-davinci-002) model here since it can fit the much longer contexts needed. We also evaluate the impact of model scale on DECOMP by using models from the Flan-T5 family: Flan-T5-Large (0.7B), Flan-T5-XL (3B), and Flan-T5-XXL (11B).[8]

**QC**: In what country was Lost Gravity manufactured?
**QS**: [retrieve_odqa] Which company manufactured Lost Gravity?
**A**: {"titles": ["Lost Gravity (roller coaster)" ... ], "answer": ["Mack Rides"]}
**QS**: [retrieve_odqa] The company Mack Rides is from which country?
**A**: {"titles": ["Mack Rides", ...], "answer": ["Germany"]}
**QS**: [multihop_rcqa] Titles: ["Lost Gravity (roller coaster)" ..., "Mack Rides" ...].
Question: In what country was Lost Gravity manufactured?            decomposer

Wikipedia Title: Lost Gravity (roller ... )
Lost Gravity is a steel roller coaster ...
Wikipedia Title: Mack Rides
Mack Rides GmbH & Co ...
...
**Q**: In what country was Lost Gravity manufactured?
**A**: "Germany"            multihop_rcqa

Figure 11: The prompt used to answer open-domain multihop questions using Elasticsearch-based retrieval. The retrieve_odqa prompt is given in Fig. 6.

Fig. 11 shows the decomposition prompt we use. The decomposer generates (singlehop) sub-questions and delegates them to retrieve_odqa (described in Fig. 6). As we showed earlier, this module retrieves relevant documents then uses an RC model to answer. retrieve_odqa returns both the answer and the documents, allowing subsequent sub-questions to use the answers (e.g. "Mack Rides") and the multihop_rcqa model to use the documents. The final multihop_rcqa model is prompted to produce the answer directly or using CoT given K paragraphs.

---

[8]We still use GPT3-sized models for decomposition since only these models are reliably able to produce the required structured outputs.

We compare our approach against two baselines: **A. No Context (No-Ctxt),** A closed-book setting baseline where the model must rely only on its parametric knowledge. **B. NoDecomp Context (NoDecomp-Ctxt),** A simple retrieval baseline where we retrieve K paragraphs using the multi-hop question as the input and use that as context. For both NoDecomp-Ctxt and Decomp-Ctxt, K is selected by hyperparameter tuning (App. A). We manually annotate CoTs and decompositions for 20 training set questions, and sample 3 prompts of 15 questions each for all approaches. The detailed prompts are given in the Appendix G. We evaluate on 300 held-out dev questions in each dataset.

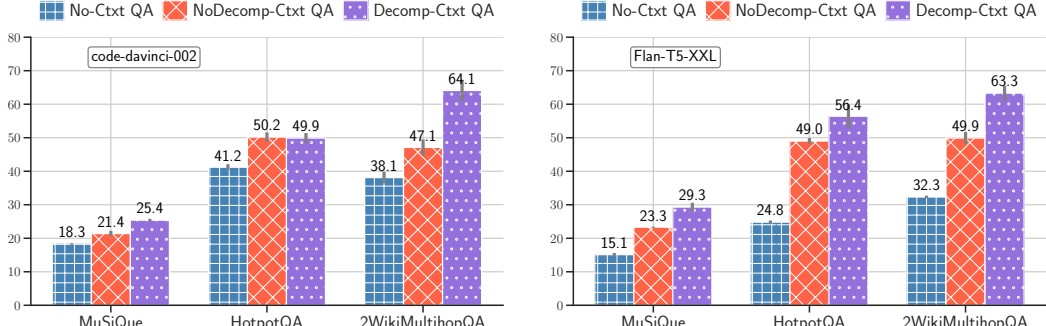

Figure 12: Answer F1[9] on three open-domain QA datasets using two base LMs: Codex (left) and Flan-T5-XXL (right) with direct prompting. Decomp-Ctxt models (ours) significantly outperforms the No-Ctxt models (no retrieval) in all settings and also outperforms our strong retrieval baseline (NoDecomp-Ctxt QA), with the exception of Codex on HotpotQA where it is comparable. See App. A.3 for results on smaller Flan-T5 models and CoT prompting.

We present results on all three datasets with direct QA prompts in Fig. 12 with other results in App. A. The Decomp-Ctxt models performs significantly better than No-Ctxt models in all the settings showing that external knowledge can be leveraged to improve few-shot models on open-domain mulithop QA. Furthermore, we show that our Decomp-Ctxt models outperform the strong retrieval baseline (NoDecomp-Ctxt) in all settings except one (Codex with HotpotQA). Finally, we show that even with the much smaller Flan-T5-XXL model, Decomp-Ctxt outperforms all the baselines and can even achieve scores comparable to the Codex-only systems.

## 4.5 ADDITIONAL RESULTS

**Post-processing CoT for error correction** DECOMP also allows us to create a targeted sub-task handler to focus on the source of error in any system. For example, CoT for arithmetic reasoning often rely on patterns (`answer is .*`) to extract answers but the CoT does not always fit this pattern. Instead, we can assign the answer extraction to a better sub-task handler (GPT3) and reduce these types of errors. This results in a 17 pt improvement on MultiArith ($78 \rightarrow 95$) and 14 pt improvement on GSM8K ($36 \rightarrow 50.6$) compared to CoT prompting (details in App. B).

While DECOMP outperforms the baselines in aggregate, we also see the **gains of DECOMP are consistent across prompt choices** (see App. D) **and decomposition schemes** (see App. E).

## 5 CONCLUSION

We proposed a new approach, Decomposed Prompting, to solve complex tasks using few-shot prompts, by decomposing them into a prompting program built out of simpler sub-tasks. Drawing inspiration from software libraries, our decomposer and shared sub-tasks are designed in a modular fashion: they use their own few-shot prompts, allowing one to independently optimize each prompt, decompose a sub-task further if necessary, or even seamlessly replace it with a symbolic system. We show that Decomposed Prompting outperforms prior work on four different tasks and generalization settings, establishing it as an effective few-shot paradigm for solving complex tasks.

---

[9]Answer F1 is computed by treating prediction and ground truth answer as bags of tokens and computing their precision and recall (Rajpurkar et al., 2016). See HotpotQA (Yang et al., 2018) for details.

ACKNOWLEDGEMENTS

We thank members of the Aristo team at the Allen Institute for AI (AI2) for their constructive feedback and the reviewers for their invaluable suggestions. This work was supported in part by the National Science Foundation under grants IIS2007290.

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

# A OPEN DOMAIN QA DETAILS

## A.1 RETRIEVAL CORPUSES FOR OPEN DOMAIN QA

We use HotpotQA in the fullwiki setting where it comes with the associated Wikipedia corpus for open-domain QA. 2WikiMultihopQA and MuSiQue, however, are originally reading comprehension datasets. Questions in 2WikiMultihopQA and MuSiQue are associated with 10 and 20 paragraphs respectively. To turn these datasets into open-domain QA datasets, we create a corpora for each dataset by combining all the paragraphs in the train, dev and test questions. As a result we get a corpus size of 430,225 paragraphs for 2WikiMultihopQA and 139,416 for MuSiQue.

## A.2 HYPERPARAMETER TUNING FOR OPEN DOMAIN QA

We treat the number of paragraphs to retrieve ($K$) in NoDecomp-Ctxt and Decomp-Ctxt models as a hyperparameter. We select it based on a grid search on a set of values to maximize performance on a held out set of 100 questions for each dataset. For NoDecomp-Ctxt, we search $K \in \{6, 8, 10\}$ for GPT3 models and $K \in 2, 4, 6, 8$ for Flan-T5-* models. For Decomp-Ctxt, we search $K \in \{2, 4, 6\}$ for GPT3 and Flan-T5-* models. Note that the ranges are different between GPT3 and Flan-T5-* as GPT3 can fit in more number of tokens. The ranges are different for NoDecomp-Ctxt and Decomp-Ctxt as $K$ refers to number of paragraphs retrieved in each round of retrieval, and NoDecomp-Ctxt has only one step of retrieval whereas Decomp-Ctxt usually has multiple retrieval steps.

## A.3 ADDITIONAL RESULTS

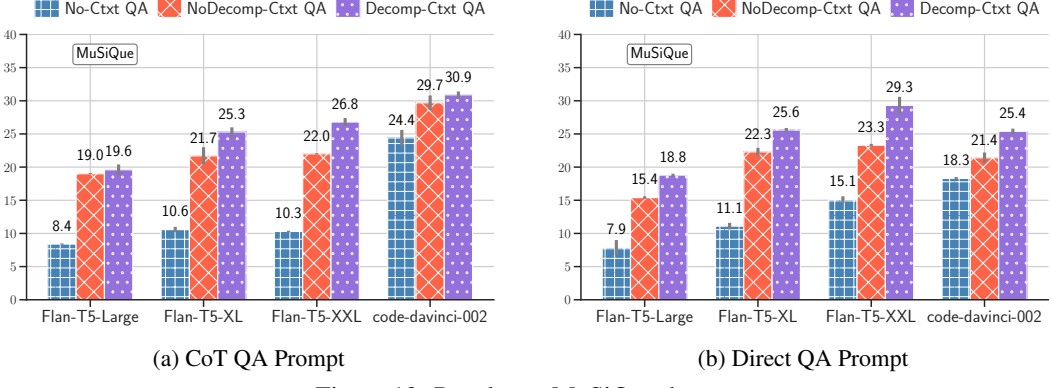

(a) CoT QA Prompt       (b) Direct QA Prompt

Figure 13: Results on MuSiQue dataset

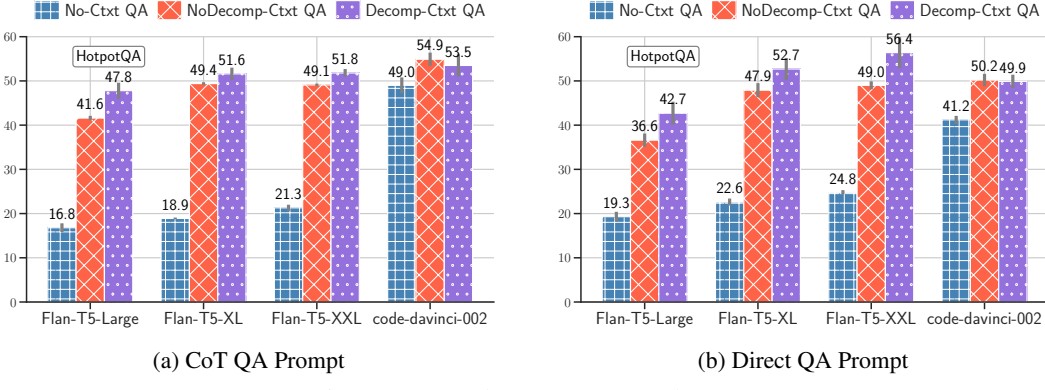

(a) CoT QA Prompt       (b) Direct QA Prompt

Figure 14: Results on HotpotQA dataset

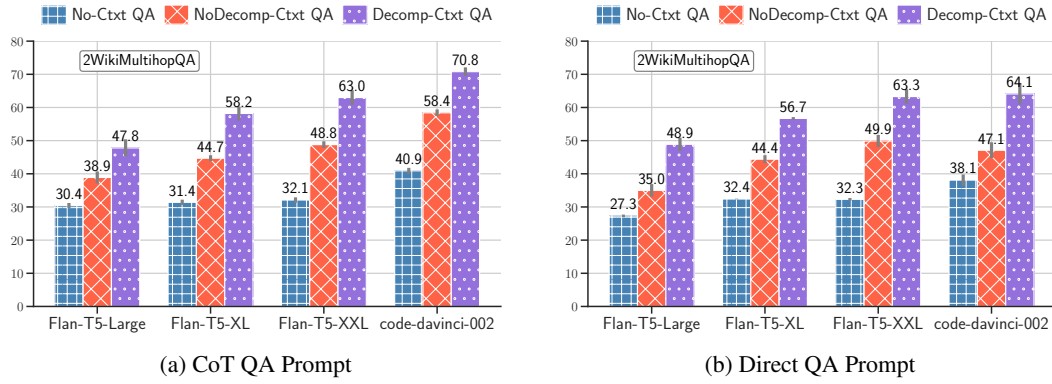

(a) CoT QA Prompt                                     (b) Direct QA Prompt

Figure 15: Results on 2WikiMultihopQA dataset

### A.3.1  MUSIQUE

We present all the results on the MuSiQue dataset in Fig. 13. Across all settings, we can see that retrieval helps substantially (large gains over No-Ctxt QA) with further improvements achieved by our DecomP-based Decomp-Ctxt QA model.

### A.3.2  HOTPOTQA

We present all the results on the HotpotQA dataset in Fig. 14. On this dataset too, we can see large gains by incorporating retrieval but the gains from using DecomP are mostly seen in the smaller models.

### A.3.3  2WIKIMULTIHOPQA

We present all the results on the 2WikiMultihopQA dataset in Fig. 15. On this dataset, we can see large gains by incorporating retrieval and also observe substantial gains by incorporating DecomP (as compared to NoDecomp-Ctxt).

## B  MATH QA

We apply Decomposed Prompting to two math QA datasets: GSM8K Cobbe et al. (2021) and MultiArith Roy & Roth (2015). For Chain-of-thought, we used the original prompts for math reasoning Wei et al. (2022). For example:

> Q: There are 15 trees in the grove. Grove workers will plant trees in the grove today. After they are done, there will be 21 trees. How many trees did the grove workers plant today? A: There are 15 trees originally. Then there were 21 trees after some more were planted. So there must have been 21 - 15 = 6. The answer is 6.

Most CoT systems Wei et al. (2022); Wang et al. (2023) rely on extracting the answer by finding the number following "answer is". However, this may not always be accurate. For example, the following CoT would be unanswerable by relying on simple patterns.

> Parker chews 4 pieces of gum a day. There are 15 pieces of gum in a pack. So he will need 4 * 30 / 15 = 8 packs of gum to last him 30 days.

Rather than relying on patterns with limited generalization, we can use a language model to extract the answer more reliably. Specifically, we use Decomposed Prompting to decompose the task into first identifying the chain-of-thought reasoning and then using a second GPT3-based sub-module to extract the answer from the CoT. We show examples of our prompts here (full prompt in App. G):

**Example from the Decomposition Prompt**

> QC: There are 15 trees in the grove. Grove workers will plant trees in the grove today. After they are done, there will be 21 trees. How many trees did the grove workers plant today?
> QS: [cot] There are 15 trees in the grove. Grove workers will plant trees in the grove today. After they are done, there will be 21 trees. How many trees did the grove workers plant today?
> A: There are 15 trees originally. Then there were 21 trees after some more were planted. So there must have been 21 - 15 = 6 trees planted.
> QS: [gpt_ans] There are 15 trees originally. Then there were 21 trees after some more were planted. So there must have been 21 - 15 = 6 trees planted.
> A: 6
> QS: [EOQ]

**Example from the gpt_ans prompt**

> Q: There are 15 trees originally. Then there were 21 trees after some more were planted. So there must have been 21 - 15 = 6 trees planted.
> A: 6

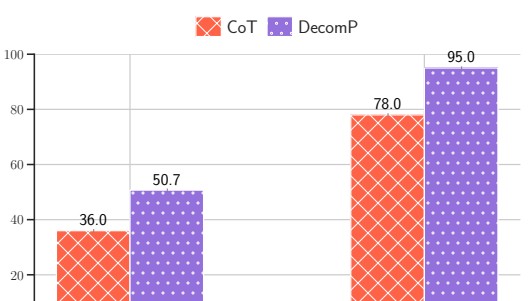

Figure 16: Our simple decomposition results in 14-17 pts on two MathQA datasets: GSM8k and MultiArith.

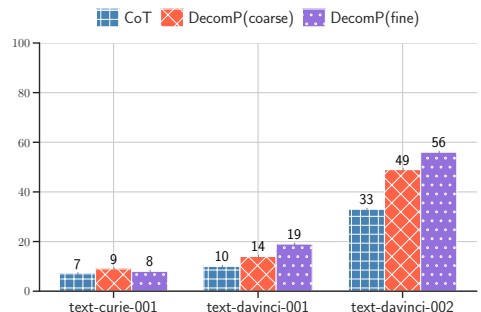

Figure 17: As the models become weaker (davinci-001) and smaller (curie-001), the performance of all the models drop. DE-COMP still outperforms CoT till the performance reaches close to zero with curie.

We present our results in Fig. 16. On the GSM8K data set[10], we outperform CoT by 14 points. On the MultiArith dataset[11], we achieve a 17 pt improvement compare to CoT. While this is a simple change, it illustrates the possibility of using DECOMP for other complex answer types, e.g. non-extractive answer generation from chain-of-thoughts.

## C  EFFECT OF SCALE ON COMMAQA

We evaluate text-curie-001, text-davinci-001 and text-davinci-002 on the CommAQA dataset. Since the curie-001 and davinci-001 have a smaller context window size, we further reduced our prompts to fit within their context windows (2048 tokens). As shown in Fig. 17, both CoT and DECOMP are effected by the model size.

## D  RESULTS ON ALL PROMPTS

### D.1  PER-PROMPT RESULT ON LETTER CONCATENATION

We present the results of the letter concatenation task (with space delimiter) for different values of N in Fig. 18. Our results are stable across the different prompts (P1, P2 and P3) and always outperform CoT and Least-to-Most prompting.

---

[10]We randomly sample 300 examples from the test set due to costs with API usage

[11]We randomly sample 200 examples from the test set due to costs with API usage

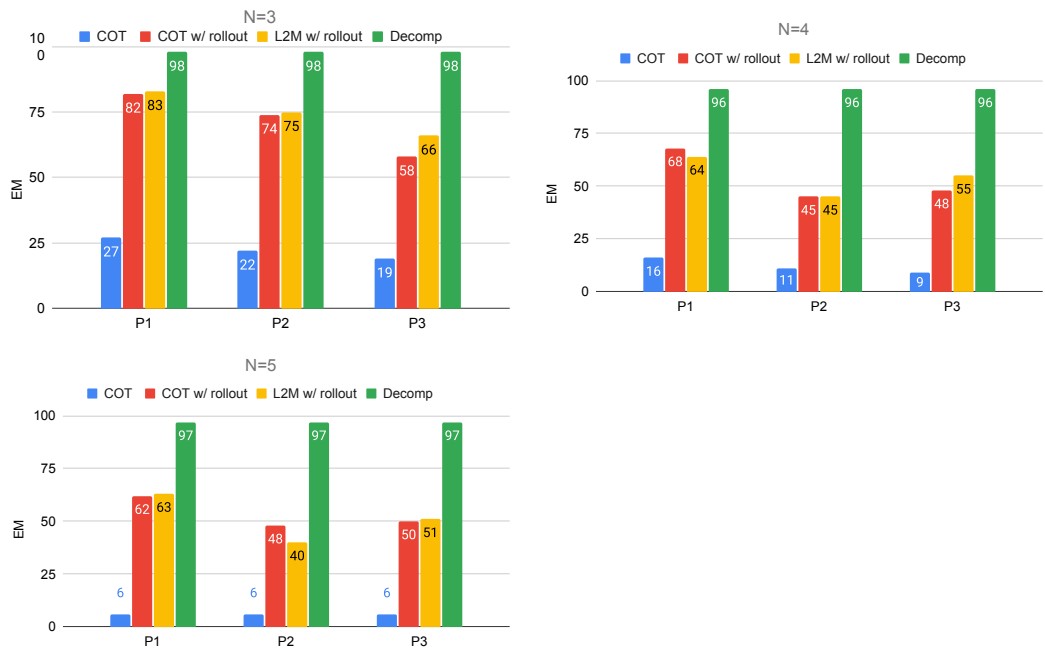

Figure 18: Across all values of N and different prompts (P1, P2 and P3), DECOMP outperform chain-of-thought reasoning and even least-to-most prompting.

## D.2 PER-PROMPT RESULTS ON COMMAQA

We also present the results of all the prompts on the CommAQA dataset in Fig. 19. Here too, we can observe that DECOMP outperforms CoT on each prompt set.

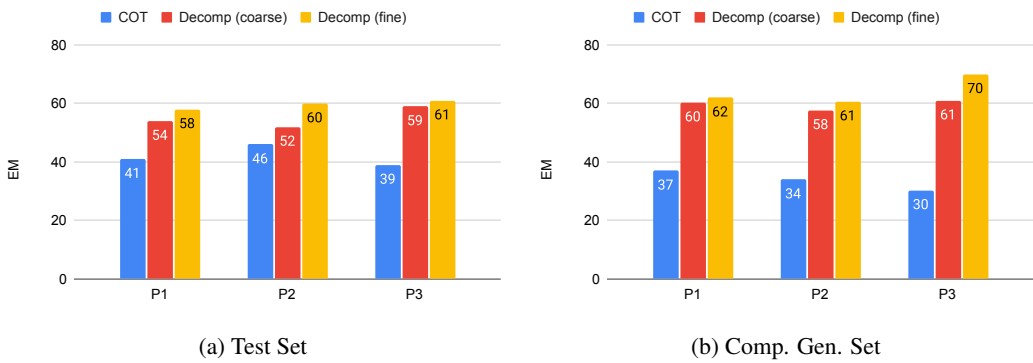

(a) Test Set                    (b) Comp. Gen. Set

Figure 19: Results of different prompts on the CommAQA dataset.

## E EFFECT OF DECOMPOSITION SCHEME

To evaluate the effect of the decomposition scheme, we experiment with two other simple decomposition structures for the letter concatenation and reversal tasks.

**Letter Concatenation** For letter concatenation, we consider an alternate scheme where we use GPT3 to generate each question rather than loop over the answers, e.g.,

QC: Take the last letters of the words in "Augusta Ada King" and concatenate them using a space.
QS: [split] What are the words in "Augusta Ada King"?
A: ["Augusta", "Ada", "King"]
QS: [str_position] What is the last letter in "Augusta"?
A: "a"
QS: [str_position] What is the last letter in "Ada"?
A: "a"
QS: [str_position] What is the last letter in "King"?
A: "g"
QS: [merge] Concatenate ["a", "a", "g"] using a space.
A: "a a g"
QS: [EOQ]

By using the decomposer prompt model to generate the sub-questions, we can be more robust to formatting issues in the output answers, e.g., we can expect GPT3 to still generate the appropriate sub-questions even if the first answer is not a valid array. However, the generated sub-questions may not correctly use all the elements of the list (change in order, missed element, repeated elements, etc).

**List Reversal**   For list reversal, instead of splitting into halves, we take the tail of the list, reverse it and then concatenate it to the head. i.e. reverse(list) = reverse(list[1:]) + list[0]. This requires more GPT3 calls (O(n)) compared to the original approach of splitting the list into halves (O(log(n))).

In both these cases, we noticed that the performance did not drop as shown in Fig. 20 and Fig. 21. On the letter concatenation task, the results were exactly the same. The new reversal decomposition schema was actually stronger on longer inputs at the cost of more calls to GPT3 (O(ln(n)) using binary splits vs O(n) one element at a time). Both these decomposition schemes are still better than CoT.

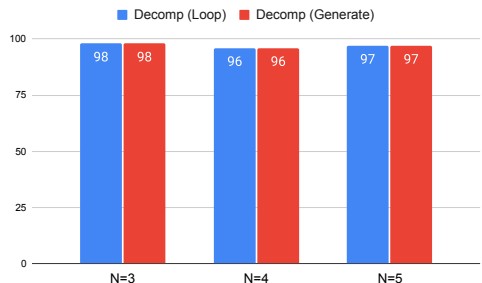

Figure 20: Both decomposition schemes for the letter concatenation task have the same scores.

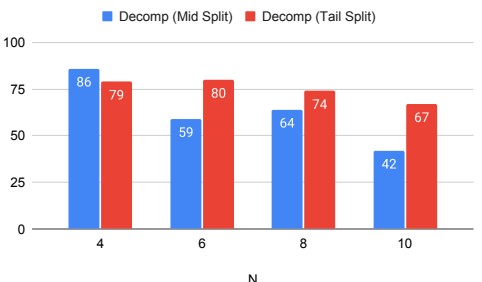

Figure 21: Recursively reversing the tail of a list is more stable at longer lengths but comes at the cost of more calls to GPT3.

# F   ERROR ANALYSIS

## F.1   LETTER CONCATENATION

### F.1.1   DECOMP

We analyzed the errors in DECOMP on the letter concatenation task and only found errors in the sub-task execution.

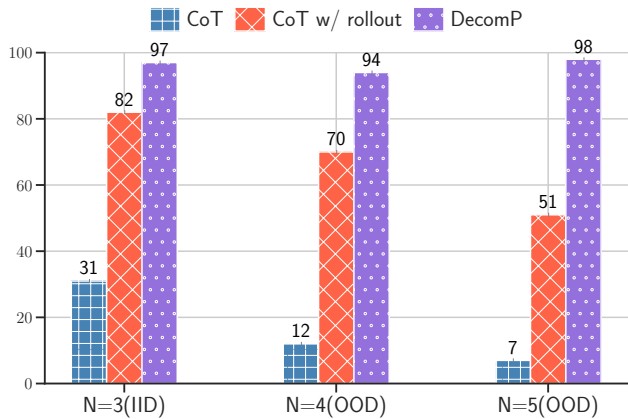

Figure 22: EM Results on the $k^{th}$ letter concatenation task (k=3) using semi-colon as delimiter with different values for N, the number of words in the input. DECOMP always outperforms and generalizes better than CoT.

---

Q: Take the letters at position 3 of the words in "Nancy Samina Abbas Caudhari Bano" and concatenate them using a space.
A: n m b u n
Prediction: c m b u n
Error: Incorrect letter extraction (Sub-task)
*What is at position 3 in "[(N, 1), (a, 2), (n, 3), (c, 4), (y, 5)]"?* ⇒ *"c"*

Q: Take the letters at position 3 of the words in "Orlando Stephen Cho Teixeira Pierre" and concatenate them using a space.
A: l e o i e
Prediction: leoie
Error: Incorrect concatenation (Sub-task)
*Concatenate ["l", "e", "o", "i", "e"] using a space.* ⇒ *"leoie"*

---

### F.1.2 CoT w/ ROLLOUT

We analyzed the errors in CoT on the letter concatenation task and found similar errors during the generation of CoT. But the frequency of these errors was higher than DECOMP, as it is not possible to effectively teach each sub-task with CoT.

---

Q: Take the letters at position 3 of the words in "Sheila Nicolas Verma Sha Sousa" and concatenate them using a space.
A: e c r a u
Pred: i c r a u
Error: Incorrect letter extraction
*...The letters and their positions in "Sheila" are "[(S, 1), (h, 2), (e, 3), (i, 4), (l, 5), (a, 6)]". The letter at position 3 in this sequence is "i"...*

Q:Take the letters at position 3 of the words in "Shobha Kailash Nakamura Peter Benitez" and concatenate them using a space.
A: o i k t n
Pred: o l k t i
Error: Incorrect letter extraction
*..."Benitez" are "[(B, 1), (e, 2), (n, 3), (i, 4), (t, 5), (e, 6), (z, 7)]". The letter at position 3 in this sequence is "i"...*

---

### F.2 COMMAQA

Similarly in CommaQA, the errors are mostly due to sub-task errors, which in this dataset correspond to answering single-hop questions. CoT also makes the same types of errors but they are more frequent since this QA sub-task can not be delegated to a specialized prompt in CoT. Since all errors are of this type, we show only one example here.

---

Q: What awards have movies written by people born in 1933 won?
A: ["Hydrallium", "Pompasole"]
Pred: ["Pompasole"]
Error: Incorrect sub-question answer
*Sub-Q:What movies has Haldron written?*
*Sub-A: ["Polytetrafluoromethane", "Skia", "Warpstone"]*
*Pred: ["Skia", "Warpstone"]*

---

## G  TASK PROMPTS

We have provided the task prompts for all the datasets for COT and our Decomposed Prompting approach.

**CoT**  Since CoT methods also perform 2-step reasoning: first generate the chain-of-thought and second extract the answer from the CoT, we use the same decomposition-based framework for COT baselines too. For example, consider the following example in our COT prompt:

---

QC: Take the letters at position 1 of the words in "Alan Mathison Turing" and concatenate them using a space.
QS: [extract] The letter at position 1 of "Alan" is "A". The letter at position 1 of "Mathison" is "M". The letter at position 1 of "Turing" is "T". Concatenating "A", "M", "T" using a space leads to "A M T". So, "Alan Mathison Turing" outputs "A M T".
A: "A M T"
QS: [EOQ]

---

GPT3 generates the chain-of-thought during the "decomposition" step and a regex-based answer extractor `extract('.* outputs "(.*)"\.')` then takes this CoT and generates the answer. In some cases, the module name is skipped in the prompt (the CoT is sent to the extractor by default).

**Operators**  In this work, we use the same operators as defined by Khot et al.. Their `select` operator is just the basic operator that replaces references to an answer index with its answer. When not specified, `select` is assumed to be the default operator. In addition, we consider two operators in our experiments: `project_values` and `project_values_flat_unique`.

- `project_values`: This operator takes a list answer $\#i = X$ and iterates over it to generate new questions by replacing mentions of $\#i$ i.e. Q = [q.replace(#i, x) for x ∈ X]. The answer to each question is simply concatenated to get the final answer i.e. A = [model(q) for q ∈ Q]. We refer to this as `foreach` for simplicity in the main text.

- `project_values_flat_unique`:  This operator performs the same steps as `project_values` but then additionally flattens the list and only returns the unique entities in the flattened list. We refer to this as `foreach_merge` in the main text for simplicity.

### G.1  LETTER CONCATENATION

We show one of the prompts used for experiments here. The entire set of prompts is provided as supplemetary material.

### G.1.1 DECOMPOSED PROMPTING

decomp

QC: Take the last letters of the words in "Augusta Ada King" and concatenate them using a space.
QS: [split] What are the words in "Augusta Ada King"?
A: ["Augusta", "Ada", "King"]
QS: (project_values) [str_position] What is the last letter in "#1"?
A: ["a", "a", "g"]
QS: [merge] Concatenate #2 using a space.
A: "a a g"
QS: [EOQ]

QC: Take the letters at position 1 of the words in "Alan Mathison Turing" and concatenate them
    using a space.
QS: [split] What are the words in "Alan Mathison Turing"?
A: ["Alan", "Mathison", "Turing"]
QS: (project_values) [str_position] What is the letter at position 1 in "#1"?
A: ["A", "M", "T"]
QS: [merge] Concatenate #2 using a space.
A: "A M T"
QS: [EOQ]

QC: Take the letters at position 4 of the words in "Herbert Alexander Simon" and concatenate
    them using a space.
QS: [split] What are the words in "Herbert Alexander Simon"?
A: ["Herbert", "Alexander", "Simon"]
QS: (project_values) [str_position] What is the letter at position 4 in "#1"?
A: ["b", "x", "o"]
QS: [merge] Concatenate #2 using a space.
A: "b x o"
QS: [EOQ]

split

Q: What are the words in "Alan Mathison Turing"?
A: ["Alan", "Mathison", "Turing"]

Q: What are the letters in "Alan"?
A: ["A", "l", "a", "n"]

Q: What are the letters and their positions in "Mathison"?
A: "[(M, 1), (a, 2), (t, 3), (h, 4), (i, 5), (s, 6), (o, 7), (n, 8)]"

Q: What are the words and their positions in "Herbert Alexander Simon"?
A: "[(Herbert, 1), (Alexander, 2), (Simon, 3)]"

str_position

QC: What is the letter at position 1 of the word "Augusta"?
QS: (select) [split] What are the letters and their positions in "Augusta"?
A: "[(A, 1), (u, 2), (g, 3), (u, 4), (s, 5), (t, 6), (a, 7)]"
QS: (select) [arr_position] What is at position 1 in #1?
A: "A"
QS: [EOQ]

QC: What is the last letter of the word "Mathison"?
QS: (select) [split] What are the letters and their positions in "Mathison"?
A: "[(M, 1), (a, 2), (t, 3), (h, 4), (i, 5), (s, 6), (o, 7), (n, 8)]"
QS: (select) [arr_position] What is the last letter in #1?
A: "n"

QS: [EOQ]

QC: What is the word at the position 4 in "Colorless green ideas sleep furiously"?
QS: (select) [split] What are the words and their positions in "Colorless green ideas sleep furiously
    "?
A: "[(Colorless, 1), (green, 2), (ideas, 3), (sleep, 4), (furiously, 5)]"
QS: (select) [arr_position] What is at the position 4 in #1?
A: "sleep"
QS: [EOQ]



`merge`



Q: Concatenate ["A", "l", "a", "n"].
A: "Alan"

Q: Concatenate ["b", "x", "o"] using a space.
A: "b x o"

Q: Concatenate ["a", "a", "g"] using a comma.
A: "a,a,g"

Q: Concatenate ["Alan", "Mathison", "Turing"] using a space.
A: "Alan Mathison Turing"

Q: Concatenate ["Allen", "Institute"].
A: "AllenInstitute"



`arr_position`



Q: What is at position 4 in "[("Colorless", 1), ("green", 2), ("ideas", 3), ("sleep", 4), ("furiously",
    5)]"?
A: "sleep"

Q: What is at position 1 in "[(M, 1), (a, 2), (t, 3), (h, 4), (i, 5), (s, 6), (o, 7), (n, 8)]"?
A: "M"

Q: What is at the last position in "[(A, 1), (u, 2), (g, 3), (u, 4), (s, 5), (t, 6), (a, 7)]"?
A: "a"

Q: What is at position 1 in "[(Herbert, 1), (Alexander, 2), (Simon, 3)]"?
A: "Herbert"

Q: What is at last position in "[(Allen, 1), (Institute, 2), (for, 3), (Artificial, 4), (Intelligence, 5)]"?
A: "Intelligence"

Q: What is at position 4 in "[(A, 1), (l, 2), (e, 3), (x, 4), (a, 5), (n, 6), (d, 7), (e, 8), (r, 9)]"?
A: "x"

### G.1.2 COT WITH ROLLOUT



`COT w/ rollout`



QC: Take the last letters of the words in "Augusta Ada King" and concatenate them using a space.
QS: The words in "Augusta Ada King" are "Augusta", "Ada" and "King". The letters and their
    positions in "Augusta" are "[(A, 1), (u, 2), (g, 3), (u, 4), (s, 5), (t, 6), (a, 7)]". The last letter in
    this sequence is "a". The letters and their positions in "Ada" are "[(A, 1), (d, 2), (a, 3)]". The
    last letter in this sequence is "a". The letters and their positions in "King" are "[(K, 1), (i, 2), (
    n, 3), (g, 4)]". The last letter in this sequence is "g". Concatenating "a", "a", "g" using a space
     leads to "a a g". So, "Augusta Ada King" outputs "a a g".
A: "a a g"
QS: [EOQ]

QC: Take the letters at position 1 of the words in "Alan Mathison Turing" and concatenate them using a space.

QS: The words in "Alan Mathison Turing" are "Alan", "Mathison", and "Turing". The letters and their positions in "Alan" are "[(A, 1), (l, 2), (a, 3), (n, 4)]". The letter at position 1 in this sequence is "A". The letters and their positions in "Mathison" are [(M, 1), (a, 2), (t, 3), (h, 4), (i, 5), (s, 6), (o, 7), (n, 8)]". The letter at position 1 in this sequence is "M". The letters and their positions in "Turing" are "[(T, 1), (u, 2), (r, 3), (i, 4), (n, 5), (g, 6)]". The letter at position 1 in this sequence is "T". Concatenating "A", "M", "T" using a space leads to "A M T". So, "Alan Mathison Turing" outputs "A M T".

A: "A M T"

QS: [EOQ]

QC: Take the letters at position 4 of the words in "Herbert Alexander Simon" and concatenate them using a space.

QS: The words in "Herbert Alexander Simon" are "Herbert", "Alexander", and "Simon". The letters and their positions in "Herbert" are "[(H, 1), (e, 2), (r, 3), (b, 4), (e, 5), (r, 6), (t, 7)]". The letter at position 4 in this sequence is "b". The letters and their positions in "Alexander" are "[(A, 1), (l, 2), (e, 3), (x, 4), (a, 5), (n, 6), (d, 7), (e, 8), (r, 9)]". The letter at position 4 in this sequence is "x". The letters and their positions in "Simon" are "[(S, 1), (i, 2), (m, 3), (o, 4), (n, 5)]". The letter at position 4 in this sequence is "o". Concatenating "b", "x", "o" using a space leads to "b x o". So, "Herbert Alexander Simon" outputs "b x o".

A: "b x o"

QS: [EOQ]

### G.1.3   COT

```
COT
```

QC: Take the last letters of the words in "Augusta Ada King" and concatenate them using a space.

QS: The last letter of "Augusta" is "a". The last letter of "Ada" is "a". The last letter of "King" is "g". Concatenating "a", "a", "g" using a space leads to "a a g". So, "Augusta Ada King" outputs "a a g".

A: "a a g"

QS: [EOQ]

QC: Take the letters at position 1 of the words in "Alan Mathison Turing" and concatenate them using a space.

QS: The letter at position 1 of "Alan" is "A". The letter at position 1 of "Mathison" is "M". The letter at position 1 of "Turing" is "T". Concatenating "A", "M", "T" using a space leads to "A M T". So, "Alan Mathison Turing" outputs "A M T".

A: "A M T"

QS: [EOQ]

QC: Take the letters at position 4 of the words in "Herbert Alexander Simon" and concatenate them using a space.

QS: The letter at position 4 of "Herbert" is "b". The letter at position 4 of "Alexander" is "x". The letter at position 4 of "Simon" is "o". Concatenating "b", "x", "o" using a space leads to "b x o". So, "Herbert Alexander Simon" outputs "b x o".

A: "b x o"

QS: [EOQ]

### G.1.4   LEAST-TO-MOST W/ ROLLOUT

```
Least-to-most Decomp
```

QC: Take the last letters of the words in "Augusta Ada King" and concatenate them using a space.

QS: [l2m] Take the last letters of the words in "Augusta Ada" and concatenate them using a space.

A: The words in "Augusta Ada" are "Augusta" and "Ada". The letters and their positions in "Augusta" are "[(A, 1), (u, 2), (g, 3), (u, 4), (s, 5), (t, 6), (a, 7)]". The last letter in this sequence

is "a". The letters and their positions in "Ada" are "[(A, 1), (d, 2), (a, 3)]". The last letter in this sequence is "a". Concatenating "a", "a" using a space leads to "a a". So, "Augusta Ada" outputs "a a".

QS: [l2m] Take the last letters of the words in "Augusta Ada King" and concatenate them using a space.

A: "Augusta Ada" outputs "a a". The letters and their positions in "King" are "[(K, 1), (i, 2), (n, 3), (g, 4)]". The last letter in this sequence is "g". Concatenating "a a", "g" using a space leads to "a a g". So, "Augusta Ada King" outputs "a a g".

QS: [ans_ext] So, "Augusta Ada King" outputs "a a g".

A: "a a g"

QS: [EOQ]

QC: Take the letters at position 1 of the words in "Alan Mathison Turing" and concatenate them using a space.

QS: [l2m] Take the letters at position 1 of the words in "Alan Mathison" and concatenate them using a space.

A: The words in "Alan Mathison" are "Alan" and "Mathison". The letters and their positions in "Alan" are "[(A, 1), (l, 2), (a, 3), (n, 4)]". The letter at position 1 in this sequence is "A". The letters and their positions in "Mathison" are [(M, 1), (a, 2), (t, 3), (h, 4), (i, 5), (s, 6), (o, 7), (n, 8)]". The letter at position 1 in this sequence is "M". Concatenating "A", "M" using a space leads to "A M". So, "Alan Mathison" outputs "A M".

QS: [l2m] Take the letters at position 1 of the words in "Alan Mathison Turing" and concatenate them using a space.

A: "Alan Mathison" outputs "A M". The letters and their positions in "Turing" are "[(T, 1), (u, 2), (r, 3), (i, 4), (n, 5), (g, 6)]". The letter at position 1 in this sequence is "T". Concatenating "A M", "T" using a space leads to "A M T". So, "Alan Mathison Turing" outputs "A M T".

QS: [ans_ext] So, "Alan Mathison Turing" outputs "A M T".

A: "A M T"

QS: [EOQ]

QC: Take the letters at position 4 of the words in "Herbert Alexander Simon" and concatenate them using a space.

QS: [l2m] Take the letters at position 4 of the words in "Herbert Alexander" and concatenate them using a space.

A: The words in "Herbert Alexander" are "Herbert" and "Alexander". The letters and their positions in "Herbert" are "[(H, 1), (e, 2), (r, 3), (b, 4), (e, 5), (r, 6), (t, 7)]". The letter at position 4 in this sequence is "b". The letters and their positions in "Alexander" are "[(A, 1), (l, 2), (e, 3), (x, 4), (a, 5), (n, 6), (d, 7), (e, 8), (r, 9)]". The letter at position 4 in this sequence is "x". Concatenating "b", "x" using a space leads to "b x". So, "Herbert Alexander" outputs "b x".

QS: [l2m] Take the letters at position 4 of the words in "Herbert Alexander Simon" and concatenate them using a space.

A: "Herbert Alexander" outputs "b x". The letters and their positions in "Simon" are "[(S, 1), (i, 2), (m, 3), (o, 4), (n, 5)]". The letter at position 4 in this sequence is "o". Concatenating "b x", "o" using a space leads to "b x o". So, "Herbert Alexander Simon" outputs "b x o".

QS: [ans_ext] So, "Herbert Alexander Simon" outputs "b x o".

A: "b x o"

QS: [EOQ]

Least-to-most COT(l2m)

Q: Take the last letters of the words in "Augusta Ada" and concatenate them using a space.

A: The words in "Augusta Ada" are "Augusta" and "Ada". The letters and their positions in "Augusta" are "[(A, 1), (u, 2), (g, 3), (u, 4), (s, 5), (t, 6), (a, 7)]". The last letter in this sequence is "a". The letters and their positions in "Ada" are "[(A, 1), (d, 2), (a, 3)]". The last letter in this sequence is "a". Concatenating "a", "a" using a space leads to "a a". So, "Augusta Ada" outputs "a a".

Q: Take the last letters of the words in "Augusta Ada King" and concatenate them using a space.

A: "Augusta Ada" outputs "a a". The letters and their positions in "King" are "[(K, 1), (i, 2), (n, 3), (g, 4)]". The last letter in this sequence is "g". Concatenating "a a", "g" using a space leads to "a a g". So, "Augusta Ada King" outputs "a a g".

Q: Take the letters at position 1 of the words in "Alan Mathison" and concatenate them using a space.

A: The words in "Alan Mathison" are "Alan" and "Mathison". The letters and their positions in "Alan" are "[(A, 1), (l, 2), (a, 3), (n, 4)]". The letter at position 1 in this sequence is "A". The letters and their positions in "Mathison" are [(M, 1), (a, 2), (t, 3), (h, 4), (i, 5), (s, 6), (o, 7), (n, 8)]". The letter at position 1 in this sequence is "M". Concatenating "A", "M" using a space leads to "A M". So, "Alan Mathison" outputs "A M".

Q: Take the letters at position 1 of the words in "Alan Mathison Turing" and concatenate them using a space.

A: "Alan Mathison" outputs "A M". The letters and their positions in "Turing" are "[(T, 1), (u, 2), (r, 3), (i, 4), (n, 5), (g, 6)]". The letter at position 1 in this sequence is "T". Concatenating "A M", "T" using a space leads to "A M T". So, "Alan Mathison Turing" outputs "A M T".

Q: Take the letters at position 4 of the words in "Herbert Alexander" and concatenate them using a space.

A: The words in "Herbert Alexander" are "Herbert" and "Alexander". The letters and their positions in "Herbert" are "[(H, 1), (e, 2), (r, 3), (b, 4), (e, 5), (r, 6), (t, 7)]". The letter at position 4 in this sequence is "b". The letters and their positions in "Alexander" are "[(A, 1), (l, 2), (e, 3), (x, 4), (a, 5), (n, 6), (d, 7), (e, 8), (r, 9)]". The letter at position 4 in this sequence is "x". Concatenating "b", "x" using a space leads to "b x". So, "Herbert Alexander" outputs "b x".

Q: Take the letters at position 4 of the words in "Herbert Alexander Simon" and concatenate them using a space.

A: "Herbert Alexander" outputs "b x". The letters and their positions in "Simon" are "[(S, 1), (i, 2), (m, 3), (o, 4), (n, 5)]". The letter at position 4 in this sequence is "o". Concatenating "b x", "o" using a space leads to "b x o". So, "Herbert Alexander Simon" outputs "b x o".

### G.1.5 ALT DECOMP SCHEMA (GENERATE EACH SUB-QUESTION)

```
decomp
```

QC: Take the last letters of the words in "Augusta Ada King" and concatenate them using a space.
QS: [split] What are the words in "Augusta Ada King"?
A: ["Augusta", "Ada", "King"]
QS: [str_position] What is the last letter in "Augusta"?
A: "a"
QS: [str_position] What is the last letter in "Ada"?
A: "a"
QS: [str_position] What is the last letter in "King"?
A: "g"
QS: [merge] Concatenate ["a", "a", "g"] using a space.
A: "a a g"
QS: [EOQ]

QC: Take the letters at position 1 of the words in "Alan Mathison Turing" and concatenate them using a space.
QS: [split] What are the words in "Alan Mathison Turing"?
A: ["Alan", "Mathison", "Turing"]
QS: [str_position] What is the letter at position 1 in "Alan"?
A: "A"
QS: [str_position] What is the letter at position 1 in "Mathison"?
A: "M"
QS: [str_position] What is the letter at position 1 in "Turing"?
A: "T"

QS: [merge] Concatenate ["A", "M", "T"] using a space.
A: "A M T"
QS: [EOQ]

QC: Take the letters at position 4 of the words in "Herbert Alexander Simon" and concatenate them using a space.
QS: [split] What are the words in "Herbert Alexander Simon"?
A: ["Herbert", "Alexander", "Simon"]
QS: [str_position] What is the letter at position 4 in "Herbert"?
A: "b"
QS: [str_position] What is the letter at position 4 in "Alexander"?
A: "x"
QS: [str_position] What is the letter at position 4 in "Simon"?
A: "o"
QS: [merge] Concatenate ["b", "x", "o"] using a space.
A: "b x o"
QS: [EOQ]

## G.2 SEQUENCE REVERSAL

### G.2.1 SPLIT REVERSAL

The prompts in this section implement Algorithm 1.

---

**Algorithm 1** A recursive reversal strategy that splits the sequence in half, reverses each half, and concatenates them. Runs in $O(\log n)$ calls to the LM where $n$ is the number of items in the sequence.

**procedure** SPLITREVERSE($x$)                          ▷ The reversal of $x$
    **if** $|x| < 4$ **then**                              ▷ The base case
        **return** $x_{|x|}, \ldots, x_1$                 ▷ The reversed sequence
    **else**                                             ▷ The inductive case
        $n \leftarrow |x|/2$                             ▷ Half the length of $x$
        $\ell \leftarrow x_1, \ldots, x_n$               ▷ The first half of $x$
        $\ell^R \leftarrow$ SPLITREVERSE($\ell$)          ▷ The reversed first half
        $r \leftarrow x_{n+1}, \ldots, x_{|x|}$           ▷ The second half of $x$
        $r^R \leftarrow$ SPLITREVERSE($r$)               ▷ The reversed second half
        **return** $r_1^R, \ldots, r_n^R, \ell_{n+1}^R, \ldots, \ell_{|x|}^R$   ▷ The concatenated reversed halves
    **end if**
**end procedure**

---

```
reverse
```
QC: Reverse the sequence "driving license, button, packet, identity card, shoe".
QS: [extract] The sequence is "1. driving license, 2. button, 3. packet, 4. identity card, 5. shoe". The sequence is 5 items long, which is more than the minimum length of 4, so we split it. Half of 5 is 5 / 2 = 2.5. Dropping the decimal, we get that the first half will be 2 items long, ending in "2. button". The first half (2 items) is "1. driving license, 2. button".
A: "1. driving license, 2. button"
QS: [extract] The first half of the sequence ends with "2. button", so the second half starts after "2. button" with "3. packet". The full sequence is 5 items long, and the first half is 2 items long, so the second half will be 5 − 2 = 3 items long. The second half of the sequence (3 items) is "3. packet, 4. identity card, 5. shoe".
A: "3. packet, 4. identity card, 5. shoe"
QS: [remove_numbers] Remove the numbers from #1.
A: "driving license, button"
QS: [remove_numbers] Remove the numbers from #2.
A: "packet, identity card, shoe"
QS: [reverse] Reverse the sequence #3.
A: "button, driving license"

QS: [reverse] Reverse the sequence #4.
A: "shoe, identity card, packet"
QS: [join] #6 #5
A: "shoe, identity card, packet, button, driving license"
QS: [EOQ]

QC: Reverse the sequence "laptop, photo, clip".
QS: [extract] The sequence is "1. laptop, 2. photo, 3. clip". The sequence is 3 items long, which is less than the minimum length of 4, so we don't need to split it. All we need to do is reverse " laptop, photo, clip".
A: "laptop, photo, clip"
QS: [cot] Reverse the sequence #1.
A: "clip, photo, laptop"
QS: [EOQ]

QC: Reverse the sequence "newspaper, glasses, laptop, bottle".
QS: [extract] The sequence is "1. newspaper, 2. glasses, 3. laptop, 4. bottle". The sequence is 4 items long, which is equal to the minimum length of 4, so we split it. Half of 4 is 4 / 2 = 2.0. Dropping the decimal, we get that the first half will be 2 items long. The first half (2 items) of the sequence is "1. newspaper, 2. glasses".
A: "1. newspaper, 2. glasses"
QS: [extract] The first half of the sequence ends with "2. glasses", so the second half starts after "2. glasses" with "3. laptop". The full sequence is 4 items long and the first half is 2 items long, so the second half will be 4 − 2 = 2 items long, ending in "2. glasses". The second half of the sequence (2 items) is "3. laptop, 4. bottle".
A: "3. laptop, 4. bottle"
QS: [remove_numbers] Remove the numbers from #1.
A: "newspaper, glasses"
QS: [remove_numbers] Remove the numbers from #2.
A: "laptop, bottle"
QS: [reverse] Reverse the sequence #3.
A: "glasses, newspaper"
QS: [reverse] Reverse the sequence #4.
A: "bottle, laptop"
QS: [join] #6 #5
A: "bottle, laptop, glasses, newspaper"
QS: [EOQ]

remove_numbers

Q: Remove the numbers from "4. bottle, 3. laptop, 2. glasses, 1. newspaper".
A: "bottle, laptop, glasses, newspaper"

Q: Remove the numbers from "1. identity card, 2. packet, 3. button".
A: "identity card, packet, button"

Q: Remove the numbers from "1. player, 2. passport, 3. umbrella, 4. radio".
A: "player, passport, umbrella, radio"

join

Q: "bottle, laptop" "glasses, newspaper".
A: "bottle, laptop, glasses, newspaper"

Q: "identity card, packet, button" "magazine, notebook, glasses".
A: "identity card, packet, button, magazine, notebook, glasses"

Q: "passport, umbrella, radio, mobile phone, photo" "player".
A: "passport, umbrella, radio, mobile phone, photo, player"

Q: "mirror, case" "toothbrush, alarm clock".
A: "mirror, case, toothbrush, alarm clock"

Q: "light bulb, clip, umbrella" "driving licence, watch".
A: "light bulb, clip, umbrella, driving licence, watch"

```
cot
```

QC: Reverse the sequence "newspaper, glasses, laptop, bottle".
QS: First is newspaper. Second is glasses. Third is laptop. Fourth is bottle. Now to reverse, change the order to: Fourth is bottle. Third is laptop. Second is glasses. First is newspaper. So the answer is "bottle, laptop, glasses, newspaper".
A: "bottle, laptop, glasses, newspaper"

QC: Reverse the sequence "laptop, photo, clip".
QS: First is laptop. Second is photo. Third is clip. Now to reverse, change the order to: Third is clip . Second is photo. First is laptop. So the answer is "clip, photo, laptop".
A: "clip, photo, laptop"

QC: Reverse the sequence "driving license, button, packet, identity card, pineapple".
QS: First is driving license. Second is button. Third is packet. Fourth is identity card. Fifth is pineapple. Now to reverse, change the order to: Fifth is pineapple. Fourth is identity card. Third is packet. Second is button. First is driving license. So the answer is "pineapple, identity card, packet, button, driving license".
A: "pineapple, identity card, packet, button, driving license"

```
unrolled_decomp
```

QC: Reverse the sequence "driving license, button, packet, identity card, shoe".
QS: The sequence is "1. driving license, 2. button, 3. packet, 4. identity card, 5. shoe". The sequence is 5 items long, which is more than the minimum length of 4, so we split it. Half of 5 is 5 / 2 = 2.5. Dropping the decimal, we get that the first half will be 2 items long, ending in "2. button". The first half (2 items) is "1. driving license, 2. button". The first half of the sequence ends with "2. button", so the second half starts after "2. button" with "3. packet". The full sequence is 5 items long, and the first half is 2 items long, so the second half will be 5 − 2 = 3 items long. The second half of the sequence (3 items) is "3. packet, 4. identity card, 5. shoe". Removing the numbers from "1. driving license, 2. button", we get "driving license, button". Removing the numbers from "3. packet, 4. identity card, 5. shoe", we get "packet, identity card, shoe". Reversing the sequence "driving license, button", First is driving license. Second is button. Now to reverse, change the order to: Second is button. First is driving license. So the answer is "button, driving license". Reversing the sequence "packet, identity card, shoe", First is packet. Second is identity card. Third is shoe. Now to reverse, change the order to: Third is shoe. Second is identity card. First is packet. So the answer is "shoe, identity card, packet". Joining "shoe, identity card, packet" and "button, driving license", the answer is "shoe, identity card, packet, button, driving license".
A: "shoe, identity card, packet, button, driving license"
QS: [EOQ]

QC: Reverse the sequence "laptop, photo, clip".
QS: The sequence is "1. laptop, 2. photo, 3. clip". The sequence is 3 items long, which is less than the minimum length of 4, so we don't need to split it. All we need to do is reverse "laptop, photo, clip". First is laptop. Second is photo. Third is clip. Now to reverse, change the order to : Third is clip. Second is photo. First is laptop. So the answer is "clip, photo, laptop".
A: "clip, photo, laptop"
QS: [EOQ]

QC: Reverse the sequence "newspaper, glasses, laptop, bottle".
QS: The sequence is "1. newspaper, 2. glasses, 3. laptop, 4. bottle". The sequence is 4 items long, which is equal to the minimum length of 4, so we split it. Half of 4 is 4 / 2 = 2.0. Dropping the decimal, we get that the first half will be 2 items long, ending in "2. glasses". The first half

(2 items) is "1. newspaper, 2. glasses". The first half of the sequence ends with "2. glasses", so the second half starts after "2. glasses" with "3. laptop". The full sequence is 4 items long, and the first half is 2 items long, so the second half will be $4 - 2 = 2$ items long. The second half of the sequence (2 items) is "3. laptop, 4. bottle". Removing the numbers from "1. newspaper, 2. glasses", we get "newspaper, glasses". Removing the numbers from "3. laptop, 4. bottle", we get "laptop, bottle". Reversing the sequence "newspaper, glasses", First is newspaper. Second is glasses. Now to reverse, change the order to: Second is glasses. First is newspaper. So the answer is "glasses, newspaper". Reversing the sequence "laptop, bottle", First is laptop. Second is bottle. Now to reverse, change the order to: Second is bottle. First is laptop. So the answer is "bottle, laptop". Joining "bottle, laptop" and "glasses, newspaper", the answer is "bottle, laptop, glasses, newspaper".

A: "bottle, laptop, glasses, newspaper"

QS: [EOQ]

```
reverse (tail)
```

QC: Reverse the sequence "driving license, button, packet, identity card, shoe".

Q1: [extract] The sequence is "1. driving license, 2. button, 3. packet, 4. identity card, 5. shoe". The sequence is 5 items long, which is more than the minimum length of 4, so we split it. We take the first element in the sequence which is "1. driving license".

#1: "1. driving license"

Q2: [extract] The full sequence is 5 items long, so the remaining sequence will be $5 - 1 = 4$ items long. The tail of the sequence with 4 items is "2. button, 3. packet, 4. identity card, 5. shoe".

#2: "2. button, 3. packet, 4. identity card, 5. shoe"

Q3: [remove_numbers] Remove the numbers from #1.

#3: "driving license"

Q4: [remove_numbers] Remove the numbers from #2.

#4: "button, packet, identity card, shoe"

Q5: [reverse] Reverse the sequence #4.

#5: "shoe, identity card, packet, button"

Q6: [join] #5 #3

#6: "shoe, identity card, packet, button, driving license"

Q7: [EOQ]

QC: Reverse the sequence "laptop, photo, clip".

Q1: [extract] The sequence is "1. laptop, 2. photo, 3. clip". The sequence is 3 items long, which is less than the minimum length of 4, so we don't need to split it. All we need to do is reverse "laptop, photo, clip".

#1: "laptop, photo, clip"

Q2: [cot] Reverse the sequence #1.

#2: "clip, photo, laptop"

Q3: [EOQ]

QC: Reverse the sequence "newspaper, glasses, laptop, bottle".

Q1: [extract] The sequence is "1. newspaper, 2. glasses, 3. laptop, 4. bottle". The sequence is 4 items long, which is more than the minimum length of 4, so we split it. We take the first element in the sequence which is "1. newspaper".

#1: "1. newspaper"

Q2: [extract] The full sequence is 4 items long, so the remaining sequence will be $4 - 1 = 3$ items long. The tail of the sequence with 3 items is "2. glasses, 3. laptop, 4. bottle".

#2: "2. glasses, 3. laptop, 4. bottle"

Q3: [remove_numbers] Remove the numbers from #1.

#3: "newspaper"

Q4: [remove_numbers] Remove the numbers from #2.

#4: "glasses, laptop, bottle"

Q5: [reverse] Reverse the sequence #4.

#5: "bottle, laptop, glasses"

Q6: [join] #5 #3

#6: "bottle, laptop, glasses, newspaper"

Q7: [EOQ]

## G.3 Long-Document QA

We show one of the prompts used for CommaQA experiments here. The entire set of prompts is provided as supplemetary material.

### G.3.1 Decomposed Prompting: (coarse)

```
decomp
```

What awards have movies produced by people born in 1910 won?
QS: (select) [qa] Who were born in the year 1910?
A: ["Teeplemole", "Muntaril"]
QS: (project_values_flat_unique) [qa] For which movies was #1 the producer?
A: ["Featsaw", "Zalate", "Premercy"]
QS: (project_values_flat_unique) [qa] Which awards were given to #2?
A: ["Zorgion", "Chowwurst", "Hallowcock"]
QS: [EOQ]

QC: What movies have people from the country Stridery acted in?
QS: (select) [qa] Who is from the country Stridery?
A: ["Gastrat"]
QS: (project_values_flat_unique) [qa] Which movies has #1 been an actor in?
A: ["Partnershipmaker", "Nilitude", "Warpstone"]
QS: [EOQ]

QC: What awards have the actors of the Erowid winning movies received?
QS: (select) [qa] Which movies were given the Erowid award?
A: ["Dewbar", "Caudacite"]
QS: (project_values_flat_unique) [qa] Who are the actors in the movie #1?
A: ["Wetherality", "Lougerière", "Gigabut"]
QS: (project_values_flat_unique) [qa] Which awards were given to #2?
A: ["Aniconder", "Trifogation"]
QS: [EOQ]

QC: What awards did the movies directed by the Modiparity winners receive?
QS: (select) [qa] Who has been awarded the Modiparity award?
A: ["Bioperatology"]
QS: (project_values_flat_unique) [qa] Which movies has #1 directed?
A: ["Pestok", "Vitrilateral"]
QS: (project_values_flat_unique) [qa] Which awards were given to #2?
A: ["Gutney", "Antipositive"]
QS: [EOQ]

QC: What awards have movies written by people born in 1935 won?
QS: (select) [qa] Who were born in the year 1935?
A: ["Sclerocybin", "Zayage"]
QS: (project_values_flat_unique) [qa] What movies has #1 written?
A: ["Noenometer", "Tayenne", "Pneumodendron"]
QS: (project_values_flat_unique) [qa] Which awards were given to #2?
A: ["Brownbeard", "Goosehead", "Handt"]
QS: [EOQ]

QC: What movies have the directors from Legault directed?
QS: (select) [qa] Who is from the country Legault?
A: ["Metatoun", "Sapien"]
QS: (project_values_flat_unique) [qa] What movies has #1 been the director of?
A: ["Coacheship", "Misapportionment"]
QS: [EOQ]

qa

movie: Premercy ; directed by: Muntaril. movie: Skirtsicine ; director: Teeplemole. movie: Featsaw ; directed by: Monsterscar. movie: Zalate ; director: Monsterscar. movie: Zalate ; awarded: Hallowcock. movie: Featsaw ; awarded: Zorgion. movie: Premercy ; award: Chowwurst. movie: Skirtsicine ; award: Hallowcock. award: Goatfly ; winner: Teeplemole. person: Monsterscar ; award: Glodome. person: Muntaril ; award: Goatfly. movie: Featsaw ; release year: 1973. movie: Zalate ; release year: 1964. movie: Skirtsicine ; release year: 1973. movie: Premercy ; year: 1961. Teeplemole was an actor in the movie Skirtsicine. Muntaril was an actor in the movie Skirtsicine. Monsterscar was an actor in the movie Premercy. Muntaril was an actor in the movie Featsaw. Teeplemole was an actor in the movie Zalate. Muntaril was born in the year 1910. Teeplemole was born in 1910. Monsterscar was born in 1942. Teeplemole is from the country of Piperfish. Monsterscar is from the country of Piperfish. Muntaril is from the country of Clony. Muntaril produced the movie Skirtsicine with others. Monsterscar was one of the producers of the movie Featsaw. Monsterscar produced the movie Premercy with others. Monsterscar produced the movie Zalate with others. Teeplemole was one of the producers of the movie Featsaw. Teeplemole produced the movie Zalate with others. Muntaril produced the movie Premercy with others. Monsterscar wrote for the movie Premercy. Muntaril was one of the writers for the movie Zalate. Muntaril wrote for the movie Featsaw. Teeplemole wrote for the movie Featsaw. Monsterscar was one of the writers for the movie Zalate. Teeplemole was one of the writers for the movie Skirtsicine.
Q: For which movies was Teeplemole the producer?
A: ["Featsaw", "Zalate"]

Q: Which awards were given to Featsaw?
A: ["Zorgion"]

movie: Misgendery ; directed by: Wetherality. movie: Dewbar ; director: Gigabut. movie: Caudacite ; director: Lougerière. movie: Tayenne ; directed by: Lougerière. movie: Misgendery ; awarded: Microsouenesis. movie: Dewbar ; awarded: Erowid. movie: Tayenne ; awarded: Cockspit. movie: Caudacite ; award: Erowid. award: Aniconder ; winner: Wetherality. award: Aniconder ; winner: Lougerière. person: Gigabut ; award: Trifogation. movie: Dewbar ; release year: 1991. movie: Tayenne ; year: 2013. movie: Caudacite ; release year: 2008. movie: Misgendery ; year: 1991. Wetherality was an actor in the movie Dewbar. Gigabut was an actor in the movie Tayenne. Lougerière was an actor in the movie Tayenne. Lougerière acted in the movie Caudacite. Lougerière acted in the movie Misgendery. Gigabut was an actor in the movie Caudacite. Wetherality was an actor in the movie Misgendery. Wetherality was born in the year 1917. Lougerière was born in 1926. Gigabut was born in the year 1917. Gigabut grew up in the nation of Triclops. Lougerière is from the country of Tatkin. Wetherality grew up in the nation of Tatkin. Lougerière produced the movie Dewbar with others. Gigabut produced the movie Tayenne with others. Gigabut produced the movie Dewbar with others. Lougerière was one of the producers of the movie Misgendery. Wetherality was one of the producers of the movie Caudacite. Gigabut was one of the producers of the movie Caudacite. Wetherality produced the movie Misgendery with others. Wetherality produced the movie Tayenne with others. Wetherality wrote for the movie Tayenne. Gigabut wrote for the movie Misgendery. Lougerière was one of the writers for the movie Caudacite. Wetherality wrote for the movie Misgendery. Gigabut wrote for the movie Tayenne. Gigabut wrote for the movie Dewbar. Lougerière wrote for the movie Dewbar. Wetherality wrote for the movie Caudacite.
Q: Who are the actors in the movie Caudacite?
A: ["Lougerière, "Gigabut"]

Q: Which movies were given the Erowid award?
A: ["Dewbar", "Caudacite"]

movie: Pastillobox ; directed by: Firmline. movie: Clenestration ; directed by: Carblock. movie: Pestok ; directed by: Bioperatology. movie: Vitrilateral ; director: Bioperatology. movie: Vitrilateral ; award: Antipositive. movie: Clenestration ; awarded: Handt. movie: Pastillobox ; awarded: Handt. movie: Pestok ; awarded: Gutney. movie: Pestok ; writer: Firmline. movie:

Clenestration ; written by: Carblock. movie: Pastillobox ; written by: Bioperatology. movie: Pestok ; writer: Bioperatology. movie: Clenestration ; written by: Firmline. movie: Vitrilateral ; writer: Bioperatology. movie: Pastillobox ; writer: Carblock. movie: Vitrilateral ; written by : Carblock. movie: Pestok ; release year: 1986. movie: Clenestration ; year: 1986. movie: Vitrilateral ; year: 1999. movie: Pastillobox ; release year: 1984. Carblock was an actor in the movie Pastillobox. Firmline was an actor in the movie Vitrilateral. Bioperatology was an actor in the movie Clenestration. Firmline acted in the movie Pastillobox. Carblock was an actor in the movie Clenestration. Bioperatology was an actor in the movie Pestok. Firmline was born in the year 1904. Bioperatology was born in the year 1935. Carblock was born in 1935. Carblock grew up in the nation of Knoppock. Firmline grew up in the nation of Tatkin. Bioperatology grew up in the nation of Tatkin. Bioperatology won the Modiparity award. Halfbill was awarded to Firmline. Halfbill was awarded to Carblock. Bioperatology was one of the producers of the movie Pestok. Bioperatology produced the movie Vitrilateral with others. Firmline produced the movie Pastillobox with others. Firmline produced the movie Clenestration with others. Carblock was one of the producers of the movie Pastillobox. Carblock produced the movie Vitrilateral with others. Carblock produced the movie Clenestration with others. Firmline was one of the producers of the movie Pestok.

Q: Who has been awarded the Modiparity award?
A: ["Bioperatology"]

Q: Which movies has Bioperatology directed?
A: ["Pestok", "Vitrilateral"]

movie: Nohit ; director: Mimicocycle. movie: Noenometer ; director: Mimicocycle. movie: Tayenne ; directed by: Zayage. movie: Pneumodendron ; director: Sclerocybin. movie: Tayenne ; awarded: Goosehead. movie: Nohit ; awarded: Handt. movie: Pneumodendron ; award: Handt. movie: Noenometer ; awarded: Brownbeard. movie: Nohit ; writer: Mimicocycle. movie: Noenometer ; written by: Sclerocybin. movie: Tayenne ; writer: Sclerocybin. movie: Pneumodendron ; written by: Zayage. movie: Tayenne ; writer: Zayage. movie: Pneumodendron ; written by: Mimicocycle. movie: Noenometer ; release year: 1991. movie: Tayenne ; year: 2013. movie: Nohit ; year: 2005. movie: Pneumodendron ; year: 2005. Mimicocycle was an actor in the movie Tayenne. Zayage acted in the movie Pneumodendron. Zayage was an actor in the movie Nohit. Sclerocybin was an actor in the movie Nohit. Sclerocybin was an actor in the movie Tayenne. Mimicocycle was an actor in the movie Noenometer. Zayage was born in 1935. Sclerocybin was born in 1935. Mimicocycle was born in 1930. Mimicocycle is from the country of Calderita. Sclerocybin grew up in the nation of Calderita. Zayage is from the country of Obility. Quinion was awarded to Zayage. Fannyxist was awarded to Sclerocybin. Fannyxist was awarded to Mimicocycle. Mimicocycle produced the movie Nohit with others. Zayage was one of the producers of the movie Nohit. Sclerocybin was one of the producers of the movie Tayenne. Sclerocybin produced the movie Pneumodendron with others. Zayage produced the movie Pneumodendron with others. Mimicocycle was one of the producers of the movie Tayenne. Sclerocybin was one of the producers of the movie Noenometer. Zayage produced the movie Noenometer with others

Q: What movies has Sclerocybin written?
A: ["Noenometer", "Tayenne"]

Q: Who were born in the year 1935?
A: ["Sclerocybin", "Zayage"]

### G.3.2 DECOMPOSED PROMPTING: (FINE)

```
decomp
```

What awards have movies produced by people born in 1910 won?
QS: (select) [simp_qa] Who were born in the year 1910?
A: ["Teeplemole", "Muntaril"]
QS: (project_values_flat_unique) [pos_qa] For which movies was #1 the producer?
A: ["Featsaw", "Zalate", "Premercy"]
QS: (project_values_flat_unique) [aw_qa] Which awards were given to #2?
A: ["Zorgion", "Chowwurst", "Hallowcock"]

QS: [EOQ]

QC: What movies have people from the country Stridery acted in?
QS: (select) [simp_qa] Who is from the country Stridery?
A: ["Gastrat"]
QS: (project_values_flat_unique) [pos_qa] Which movies has #1 been an actor in?
A: ["Partnershipmaker", "Nilitude", "Warpstone"]
QS: [EOQ]

QC: What awards have the actors of the Erowid winning movies received?
QS: (select) [aw_qa] Which movies were given the Erowid award?
A: ["Dewbar", "Caudacite"]
QS: (project_values_flat_unique) [pos_qa] Who are the actors in the movie #1?
A: ["Wetherality", "Lougerière", "Gigabut"]
QS: (project_values_flat_unique) [aw_qa] Which awards were given to #2?
A: ["Aniconder", "Trifogation"]
QS: [EOQ]

QC: What awards did the movies directed by the Modiparity winners receive?
QS: (select) [aw_qa] Who has been awarded the Modiparity award?
A: ["Bioperatology"]
QS: (project_values_flat_unique) [pos_qa] Which movies has #1 directed?
A: ["Pestok", "Vitrilateral"]
QS: (project_values_flat_unique) [aw_qa] Which awards were given to #2?
A: ["Gutney", "Antipositive"]
QS: [EOQ]

QC: What awards have movies written by people born in 1935 won?
QS: (select) [simp_qa] Who were born in the year 1935?
A: ["Sclerocybin", "Zayage"]
QS: (project_values_flat_unique) [pos_qa] What movies has #1 written?
A: ["Noenometer", "Tayenne", "Pneumodendron"]
QS: (project_values_flat_unique) [aw_qa] Which awards were given to #2?
A: ["Brownbeard", "Goosehead", "Handt"]
QS: [EOQ]

QC: What movies have the directors from Legault directed?
QS: (select) [simp_qa] Who is from the country Legault?
A: ["Metatoun", "Sapien"]
QS: (project_values_flat_unique) [pos_qa] What movies has #1 been the director of?
A: ["Coacheship", "Misapportionment"]
QS: [EOQ]

```
aw_qa
```

movie: Premercy ; directed by: Muntaril. movie: Skirtsicine ; director: Teeplemole. movie: Featsaw ; directed by: Monsterscar. movie: Zalate ; director: Monsterscar. movie: Zalate ; awarded: Hallowcock. movie: Featsaw ; awarded: Zorgion. movie: Premercy ; award: Chowwurst. movie: Skirtsicine ; award: Hallowcock. award: Goatfly ; winner: Teeplemole. person: Monsterscar ; award: Glodome. person: Muntaril ; award: Goatfly. movie: Featsaw ; release year: 1973. movie: Zalate ; release year: 1964. movie: Skirtsicine ; release year: 1973. movie: Premercy ; year: 1961. Teeplemole was an actor in the movie Skirtsicine. Muntaril was an actor in the movie Skirtsicine. Monsterscar was an actor in the movie Premercy. Muntaril was an actor in the movie Featsaw. Teeplemole was an actor in the movie Zalate. Muntaril was born in the year 1910. Teeplemole was born in 1910. Monsterscar was born in 1942. Teeplemole is from the country of Piperfish. Monsterscar is from the country of Piperfish. Muntaril is from the country of Clony. Muntaril produced the movie Skirtsicine with others. Monsterscar was one of the producers of the movie Featsaw. Monsterscar produced the movie Premercy with others. Monsterscar produced the movie Zalate with others. Teeplemole was one of the producers of the movie Featsaw. Teeplemole produced the movie Zalate with

others. Muntaril produced the movie Premercy with others. Monsterscar wrote for the movie Premercy. Muntaril was one of the writers for the movie Zalate. Muntaril wrote for the movie Featsaw. Teeplemole wrote for the movie Featsaw. Monsterscar was one of the writers for the movie Zalate. Teeplemole was one of the writers for the movie Skirtsicine.

Q: Which awards were given to Zalate?
A: ["Hallowcock"]

Q: Which awards were given to Featsaw?
A: ["Zorgion"]

movie: Misgendery ; directed by: Wetherality. movie: Dewbar ; director: Gigabut. movie: Caudacite ; director: Lougerière. movie: Tayenne ; directed by: Lougerière. movie: Misgendery ; awarded: Microsouenesis. movie: Dewbar ; awarded: Erowid. movie: Tayenne ; awarded: Cockspit. movie: Caudacite ; award: Erowid. award: Aniconder ; winner: Wetherality. award: Aniconder ; winner: Lougerière. person: Gigabut ; award: Trifogation. movie: Dewbar ; release year: 1991. movie: Tayenne ; year: 2013. movie: Caudacite ; release year: 2008. movie: Misgendery ; year: 1991. Wetherality was an actor in the movie Dewbar. Gigabut was an actor in the movie Tayenne. Lougerière was an actor in the movie Tayenne. Lougerière acted in the movie Caudacite. Lougerière acted in the movie Misgendery. Gigabut was an actor in the movie Caudacite. Wetherality was an actor in the movie Misgendery. Wetherality was born in the year 1917. Lougerière was born in 1926. Gigabut was born in the year 1917. Gigabut grew up in the nation of Triclops. Lougerière is from the country of Tatkin. Wetherality grew up in the nation of Tatkin. Lougerière produced the movie Dewbar with others. Gigabut produced the movie Tayenne with others. Gigabut produced the movie Dewbar with others. Lougerière was one of the producers of the movie Misgendery. Wetherality was one of the producers of the movie Caudacite. Gigabut was one of the producers of the movie Caudacite. Wetherality produced the movie Misgendery with others. Wetherality produced the movie Tayenne with others. Wetherality wrote for the movie Tayenne. Gigabut wrote for the movie Misgendery. Lougerière was one of the writers for the movie Caudacite. Wetherality wrote for the movie Misgendery. Gigabut wrote for the movie Tayenne. Gigabut wrote for the movie Dewbar. Lougerière wrote for the movie Dewbar. Wetherality wrote for the movie Caudacite.

Q: Which movies were given the Erowid award?
A: ["Dewbar", "Caudacite"]

Q: Which awards were given to Wetherality?
A: ["Aniconder"]

movie: Pastillobox ; directed by: Firmline. movie: Clenestration ; directed by: Carblock. movie: Pestok ; directed by: Bioperatology. movie: Vitrilateral ; director: Bioperatology. movie: Vitrilateral ; award: Antipositive. movie: Clenestration ; awarded: Handt. movie: Pastillobox ; awarded: Handt. movie: Pestok ; awarded: Gutney. movie: Pestok ; writer: Firmline. movie: Clenestration ; written by: Carblock. movie: Pastillobox ; written by: Bioperatology. movie: Pestok ; writer: Bioperatology. movie: Clenestration ; written by: Firmline. movie: Vitrilateral ; writer: Bioperatology. movie: Pastillobox ; writer: Carblock. movie: Vitrilateral ; written by: Carblock. movie: Pestok ; release year: 1986. movie: Clenestration ; year: 1986. movie: Vitrilateral ; year: 1999. movie: Pastillobox ; release year: 1984. Carblock was an actor in the movie Pastillobox. Firmline was an actor in the movie Vitrilateral. Bioperatology was an actor in the movie Clenestration. Firmline acted in the movie Pastillobox. Carblock was an actor in the movie Clenestration. Bioperatology was an actor in the movie Pestok. Firmline was born in the year 1904. Bioperatology was born in the year 1935. Carblock was born in 1935. Carblock grew up in the nation of Knoppock. Firmline grew up in the nation of Tatkin. Bioperatology grew up in the nation of Tatkin. Bioperatology won the Modiparity award. Halfbill was awarded to Firmline. Halfbill was awarded to Carblock. Bioperatology was one of the producers of the movie Pestok. Bioperatology produced the movie Vitrilateral with others. Firmline produced the movie Pastillobox with others. Firmline produced the movie Clenestration with others. Carblock was one of the producers of the movie Pastillobox. Carblock produced the movie Vitrilateral with others. Carblock produced the movie Clenestration with others. Firmline was one of the producers of the movie Pestok.

Q: Who has been awarded the Modiparity award?
A: ["Bioperatology"]

Q: Which awards were given to Pestok?
A: ["Gutney"]

movie: Nohit ; director: Mimicocycle. movie: Noenometer ; director: Mimicocycle. movie: Tayenne ; directed by: Zayage. movie: Pneumodendron ; director: Sclerocybin. movie: Tayenne ; awarded: Goosehead. movie: Nohit ; awarded: Handt. movie: Pneumodendron ; award: Handt. movie: Noenometer ; awarded: Brownbeard. movie: Nohit ; writer: Mimicocycle. movie: Noenometer ; written by: Sclerocybin. movie: Tayenne ; writer: Sclerocybin. movie: Pneumodendron ; written by: Zayage. movie: Tayenne ; writer: Zayage. movie: Pneumodendron ; written by: Mimicocycle. movie: Noenometer ; release year: 1991. movie: Tayenne ; year: 2013. movie: Nohit ; year: 2005. movie: Pneumodendron ; year: 2005. Mimicocycle was an actor in the movie Tayenne. Zayage acted in the movie Pneumodendron. Zayage was an actor in the movie Nohit. Sclerocybin was an actor in the movie Nohit. Sclerocybin was an actor in the movie Tayenne. Mimicocycle was an actor in the movie Noenometer. Zayage was born in 1935. Sclerocybin was born in 1935. Mimicocycle was born in 1930. Mimicocycle is from the country of Calderita. Sclerocybin grew up in the nation of Calderita. Zayage is from the country of Obility. Quinion was awarded to Zayage. Fannyxist was awarded to Sclerocybin. Fannyxist was awarded to Mimicocycle. Mimicocycle produced the movie Nohit with others. Zayage was one of the producers of the movie Nohit. Sclerocybin was one of the producers of the movie Tayenne. Sclerocybin produced the movie Pneumodendron with others. Zayage produced the movie Pneumodendron with others. Mimicocycle was one of the producers of the movie Tayenne. Sclerocybin was one of the producers of the movie Noenometer. Zayage produced the movie Noenometer with others
Q: Which awards were given to Noenometer?
A: ["Brownbeard"]

Q: Which awards were given to Pneumodendron?
A: ["Handt"]

`pos_qa`

movie: Premercy ; directed by: Muntaril. movie: Skirtsicine ; director: Teeplemole. movie: Featsaw ; directed by: Monsterscar. movie: Zalate ; director: Monsterscar. movie: Zalate ; awarded: Hallowcock. movie: Featsaw ; awarded: Zorgion. movie: Premercy ; award: Chowwurst. movie: Skirtsicine ; award: Hallowcock. award: Goatfly ; winner: Teeplemole. person: Monsterscar ; award: Glodome. person: Muntaril ; award: Goatfly. movie: Featsaw ; release year: 1973. movie: Zalate ; release year: 1964. movie: Skirtsicine ; release year: 1973. movie: Premercy ; year: 1961. Teeplemole was an actor in the movie Skirtsicine. Muntaril was an actor in the movie Skirtsicine. Monsterscar was an actor in the movie Premercy. Muntaril was an actor in the movie Featsaw. Teeplemole was an actor in the movie Zalate. Muntaril was born in the year 1910. Teeplemole was born in 1910. Monsterscar was born in 1942. Teeplemole is from the country of Piperfish. Monsterscar is from the country of Piperfish. Muntaril is from the country of Clony. Muntaril produced the movie Skirtsicine with others. Monsterscar was one of the producers of the movie Featsaw. Monsterscar produced the movie Premercy with others. Monsterscar produced the movie Zalate with others. Teeplemole was one of the producers of the movie Featsaw. Teeplemole produced the movie Zalate with others. Muntaril produced the movie Premercy with others. Monsterscar wrote for the movie Premercy. Muntaril was one of the writers for the movie Zalate. Muntaril wrote for the movie Featsaw. Teeplemole wrote for the movie Featsaw. Monsterscar was one of the writers for the movie Zalate. Teeplemole was one of the writers for the movie Skirtsicine.
Q: For which movies was Teeplemole the producer?
A: ["Featsaw", "Zalate"]

Q: For which movies was Muntaril the producer?
A: ["Premercy]

movie: Misgendery ; directed by: Wetherality. movie: Dewbar ; director: Gigabut. movie: Caudacite ; director: Lougerière. movie: Tayenne ; directed by: Lougerière. movie: Misgendery ; awarded: Microsouenesis. movie: Dewbar ; awarded: Erowid. movie: Tayenne ; awarded: Cockspit. movie: Caudacite ; award: Erowid. award: Aniconder ; winner: Wetherality. award: Aniconder ; winner: Lougerière. person: Gigabut ; award: Trifogation. movie: Dewbar ; release year: 1991. movie: Tayenne ; year: 2013. movie: Caudacite ; release year: 2008. movie: Misgendery ; year: 1991. Wetherality was an actor in the movie Dewbar. Gigabut was an actor in the movie Tayenne. Lougerière was an actor in the movie Tayenne. Lougerière acted in the movie Caudacite. Lougerière acted in the movie Misgendery. Gigabut was an actor in the movie Caudacite. Wetherality was an actor in the movie Misgendery. Wetherality was born in the year 1917. Lougerière was born in 1926. Gigabut was born in the year 1917. Gigabut grew up in the nation of Triclops. Lougerière is from the country of Tatkin . Wetherality grew up in the nation of Tatkin. Lougerière produced the movie Dewbar with others. Gigabut produced the movie Tayenne with others. Gigabut produced the movie Dewbar with others. Lougerière was one of the producers of the movie Misgendery. Wetherality was one of the producers of the movie Caudacite. Gigabut was one of the producers of the movie Caudacite. Wetherality produced the movie Misgendery with others. Wetherality produced the movie Tayenne with others. Wetherality wrote for the movie Tayenne. Gigabut wrote for the movie Misgendery. Lougerière was one of the writers for the movie Caudacite. Wetherality wrote for the movie Misgendery. Gigabut wrote for the movie Tayenne. Gigabut wrote for the movie Dewbar. Lougerière wrote for the movie Dewbar. Wetherality wrote for the movie Caudacite.
Q: Who are the actors in the movie Dewbar?
A: ["Wetherality"]

Q: Who are the actors in the movie Caudacite?
A: ["Lougerière, "Gigabut"]

movie: Pastillobox ; directed by: Firmline. movie: Clenestration ; directed by: Carblock. movie: Pestok ; directed by: Bioperatology. movie: Vitrilateral ; director: Bioperatology. movie: Vitrilateral ; award: Antipositive. movie: Clenestration ; awarded: Handt. movie: Pastillobox ; awarded: Handt. movie: Pestok ; awarded: Gutney. movie: Pestok ; writer: Firmline. movie: Clenestration ; written by: Carblock. movie: Pastillobox ; written by: Bioperatology. movie: Pestok ; writer: Bioperatology. movie: Clenestration ; written by: Firmline. movie: Vitrilateral ; writer: Bioperatology. movie: Pastillobox ; writer: Carblock. movie: Vitrilateral ; written by : Carblock. movie: Pestok ; release year: 1986. movie: Clenestration ; year: 1986. movie: Vitrilateral ; year: 1999. movie: Pastillobox ; release year: 1984. Carblock was an actor in the movie Pastillobox. Firmline was an actor in the movie Vitrilateral. Bioperatology was an actor in the movie Clenestration. Firmline acted in the movie Pastillobox. Carblock was an actor in the movie Clenestration. Bioperatology was an actor in the movie Pestok. Firmline was born in the year 1904. Bioperatology was born in the year 1935. Carblock was born in 1935. Carblock grew up in the nation of Knoppock. Firmline grew up in the nation of Tatkin. Bioperatology grew up in the nation of Tatkin. Bioperatology won the Modiparity award. Halfbill was awarded to Firmline. Halfbill was awarded to Carblock. Bioperatology was one of the producers of the movie Pestok. Bioperatology produced the movie Vitrilateral with others. Firmline produced the movie Pastillobox with others. Firmline produced the movie Clenestration with others. Carblock was one of the producers of the movie Pastillobox. Carblock produced the movie Vitrilateral with others. Carblock produced the movie Clenestration with others. Firmline was one of the producers of the movie Pestok.
Q: Which movies has Bioperatology directed?
A: ["Pestok", "Vitrilateral"]

Q: Which movies has Carblock directed?
A: ["Clenestration"]

movie: Nohit ; director: Mimicocycle. movie: Noenometer ; director: Mimicocycle. movie: Tayenne ; directed by: Zayage. movie: Pneumodendron ; director: Sclerocybin. movie: Tayenne ; awarded: Goosehead. movie: Nohit ; awarded: Handt. movie: Pneumodendron ; award: Handt. movie: Noenometer ; awarded: Brownbeard. movie: Nohit ; writer:

Mimicocycle. movie: Noenometer ; written by: Sclerocybin. movie: Tayenne ; writer: Sclerocybin. movie: Pneumodendron ; written by: Zayage. movie: Tayenne ; writer: Zayage. movie: Pneumodendron ; written by: Mimicocycle. movie: Noenometer ; release year: 1991. movie: Tayenne ; year: 2013. movie: Nohit ; year: 2005. movie: Pneumodendron ; year: 2005. Mimicocycle was an actor in the movie Tayenne. Zayage acted in the movie Pneumodendron. Zayage was an actor in the movie Nohit. Sclerocybin was an actor in the movie Nohit. Sclerocybin was an actor in the movie Tayenne. Mimicocycle was an actor in the movie Noenometer. Zayage was born in 1935. Sclerocybin was born in 1935. Mimicocycle was born in 1930. Mimicocycle is from the country of Calderita. Sclerocybin grew up in the nation of Calderita. Zayage is from the country of Obility. Quinion was awarded to Zayage. Fannyxist was awarded to Sclerocybin. Fannyxist was awarded to Mimicocycle. Mimicocycle produced the movie Nohit with others. Zayage was one of the producers of the movie Nohit. Sclerocybin was one of the producers of the movie Tayenne. Sclerocybin produced the movie Pneumodendron with others. Zayage produced the movie Pneumodendron with others. Mimicocycle was one of the producers of the movie Tayenne. Sclerocybin was one of the producers of the movie Noenometer. Zayage produced the movie Noenometer with others

Q: What movies has Sclerocybin written?
A: ["Noenometer", "Tayenne"]

Q: What movies has Zayage written?
A: ["Pneumodendron", "Tayenne"]

`simp_qa`

movie: Premercy ; directed by: Muntaril. movie: Skirtsicine ; director: Teeplemole. movie: Featsaw ; directed by: Monsterscar. movie: Zalate ; director: Monsterscar. movie: Zalate ; awarded: Hallowcock. movie: Featsaw ; awarded: Zorgion. movie: Premercy ; award: Chowwurst. movie: Skirtsicine ; award: Hallowcock. award: Goatfly ; winner: Teeplemole. person: Monsterscar ; award: Glodome. person: Muntaril ; award: Goatfly. movie: Featsaw ; release year: 1973. movie: Zalate ; release year: 1964. movie: Skirtsicine ; release year: 1973. movie: Premercy ; year: 1961. Teeplemole was an actor in the movie Skirtsicine. Muntaril was an actor in the movie Skirtsicine. Monsterscar was an actor in the movie Premercy. Muntaril was an actor in the movie Featsaw. Teeplemole was an actor in the movie Zalate. Muntaril was born in the year 1910. Teeplemole was born in 1910. Monsterscar was born in 1942. Teeplemole is from the country of Piperfish. Monsterscar is from the country of Piperfish. Muntaril is from the country of Clony. Muntaril produced the movie Skirtsicine with others. Monsterscar was one of the producers of the movie Featsaw. Monsterscar produced the movie Premercy with others. Monsterscar produced the movie Zalate with others. Teeplemole was one of the producers of the movie Featsaw. Teeplemole produced the movie Zalate with others. Muntaril produced the movie Premercy with others. Monsterscar wrote for the movie Premercy. Muntaril was one of the writers for the movie Zalate. Muntaril wrote for the movie Featsaw. Teeplemole wrote for the movie Featsaw. Monsterscar was one of the writers for the movie Zalate. Teeplemole was one of the writers for the movie Skirtsicine.
Q: Who were born in the year 1910?
A: ["Teeplemole", "Muntaril"]

Q: From which country is Monsterscar?
A: ["Piperfish"]

movie: Nilitude ; director: Monsterscar. movie: Dewbar ; directed by: Metatoun. movie: Warpstone ; directed by: Gastrat. movie: Partnershipmaker ; director: Metatoun. movie: Dewbar ; award: Tachychronograph. movie: Partnershipmaker ; awarded: Tachychronograph. movie: Nilitude ; award: Paleodactyl. movie: Warpstone ; award: Sabonade. person: Gastrat ; award: Trifogation. award: Polyquadrase ; winner: Monsterscar. award: Trifogation ; winner: Metatoun. movie: Warpstone ; release year: 1956. movie: Dewbar ; release year: 1984. movie: Nilitude ; year: 1984. movie: Partnershipmaker ; year: 1962. Gastrat was an actor in the movie Partnershipmaker. Metatoun was an actor in the movie Partnershipmaker. Metatoun was an actor in the movie Nilitude. Gastrat acted in the movie Nilitude. Monsterscar was an actor in the movie Dewbar. Gastrat acted in the movie Warpstone. Metatoun acted in the movie Warpstone. Metatoun was born in 1939. Gastrat was born in the year 1933.

Monsterscar was born in 1933. Metatoun grew up in the nation of Moulole. Gastrat is from the country of Stridery. Monsterscar grew up in the nation of Moulole. Monsterscar produced the movie Nilitude with others. Monsterscar was one of the producers of the movie Warpstone . Metatoun was one of the producers of the movie Warpstone. Gastrat was one of the producers of the movie Nilitude. Metatoun produced the movie Partnershipmaker with others. Metatoun produced the movie Dewbar with others. Monsterscar was one of the producers of the movie Partnershipmaker. Gastrat produced the movie Dewbar with others. Metatoun wrote for the movie Partnershipmaker. Gastrat wrote for the movie Warpstone. Gastrat was one of the writers for the movie Dewbar. Monsterscar was one of the writers for the movie Nilitude. Metatoun wrote for the movie Warpstone.

Q: Who is from the country Stridery?
A: ["Gastrat"]

Q: Which movies were released in 1984?
A: ["Dewbar", "Nilitude"]

movie: Nohit ; director: Mimicocycle. movie: Noenometer ; director: Mimicocycle. movie: Tayenne ; directed by: Zayage. movie: Pneumodendron ; director: Sclerocybin. movie: Tayenne ; awarded: Goosehead. movie: Nohit ; awarded: Handt. movie: Pneumodendron ; award: Handt. movie: Noenometer ; awarded: Brownbeard. movie: Nohit ; writer: Mimicocycle. movie: Noenometer ; written by: Sclerocybin. movie: Tayenne ; writer: Sclerocybin. movie: Pneumodendron ; written by: Zayage. movie: Tayenne ; writer: Zayage. movie: Pneumodendron ; written by: Mimicocycle. movie: Noenometer ; release year: 1991. movie: Tayenne ; year: 2013. movie: Nohit ; year: 2005. movie: Pneumodendron ; year: 2005. Mimicocycle was an actor in the movie Tayenne. Zayage acted in the movie Pneumodendron. Zayage was an actor in the movie Nohit. Sclerocybin was an actor in the movie Nohit. Sclerocybin was an actor in the movie Tayenne. Mimicocycle was an actor in the movie Noenometer. Zayage was born in 1935. Sclerocybin was born in 1935. Mimicocycle was born in 1930. Mimicocycle is from the country of Calderita. Sclerocybin grew up in the nation of Calderita. Zayage is from the country of Obility. Quinion was awarded to Zayage. Fannyxist was awarded to Sclerocybin. Fannyxist was awarded to Mimicocycle. Mimicocycle produced the movie Nohit with others. Zayage was one of the producers of the movie Nohit. Sclerocybin was one of the producers of the movie Tayenne. Sclerocybin produced the movie Pneumodendron with others. Zayage produced the movie Pneumodendron with others. Mimicocycle was one of the producers of the movie Tayenne. Sclerocybin was one of the producers of the movie Noenometer. Zayage produced the movie Noenometer with others

Q: Who were born in the year 1935?
A: ["Sclerocybin", "Zayage"]

Q: When was the movie Nohit released?
A: ["2005"]

movie: Coacheship ; director: Metatoun. movie: Assamplifier ; director: Kapod. movie: Misapportionment ; director: Sapien. movie: Quinsid ; director: Kapod. movie: Assamplifier ; award: Zorgion. movie: Quinsid ; awarded: Airpipe. movie: Coacheship ; award: Electrodesal. movie: Misapportionment ; award: Airpipe. movie: Coacheship ; written by: Metatoun. movie: Misapportionment ; written by: Kapod. movie: Coacheship ; written by: Kapod. movie : Quinsid ; writer: Sapien. movie: Misapportionment ; written by: Metatoun. movie: Assamplifier ; written by: Kapod. movie: Assamplifier ; written by: Sapien. movie: Assamplifier ; release year: 2000. movie: Coacheship ; year: 2001. movie: Quinsid ; year: 2005. movie: Misapportionment ; year: 2005. Sapien was an actor in the movie Misapportionment. Sapien acted in the movie Assamplifier. Kapod acted in the movie Quinsid . Sapien acted in the movie Coacheship. Metatoun was an actor in the movie Quinsid. Kapod acted in the movie Misapportionment. Metatoun acted in the movie Coacheship. Kapod acted in the movie Assamplifier. Sapien was born in the year 1910. Metatoun was born in 1928. Kapod was born in the year 1910. Metatoun is from the country of Legault. Kapod grew up in the nation of Tatkin. Sapien is from the country of Legault. Malwarp was awarded to Sapien. Metatoun won the Monkeynote award. Kapod won the Monkeynote award. Kapod was one of the producers of the movie Quinsid. Metatoun was one of the producers of the movie

Misapportionment. Metatoun was one of the producers of the movie Quinsid. Sapien was one of the producers of the movie Assamplifier. Sapien produced the movie Coacheship with others. Metatoun was one of the producers of the movie Assamplifier. Kapod was one of the producers of the movie Misapportionment. Kapod was one of the producers of the movie Coacheship.

Q: Who is from the country Legault?
A: ["Metatoun", "Sapien"]

Q: When was Kapod born?
A: ["1910"]

### G.3.3  COT

```
COT
```

movie: Premercy ; directed by: Muntaril. movie: Skirtsicine ; director: Teeplemole. movie: Featsaw ; directed by: Monsterscar. movie: Zalate ; director: Monsterscar. movie: Zalate ; awarded: Hallowcock. movie: Featsaw ; awarded: Zorgion. movie: Premercy ; award: Chowwurst. movie: Skirtsicine ; award: Hallowcock. award: Goatfly ; winner: Teeplemole. person: Monsterscar ; award: Glodome. person: Muntaril ; award: Goatfly. movie: Featsaw ; release year: 1973. movie: Zalate ; release year: 1964. movie: Skirtsicine ; release year: 1973. movie: Premercy ; year: 1961. Teeplemole was an actor in the movie Skirtsicine. Muntaril was an actor in the movie Skirtsicine. Monsterscar was an actor in the movie Premercy. Muntaril was an actor in the movie Featsaw. Teeplemole was an actor in the movie Zalate. Muntaril was born in the year 1910. Teeplemole was born in 1910. Monsterscar was born in 1942. Teeplemole is from the country of Piperfish. Monsterscar is from the country of Piperfish. Muntaril is from the country of Clony. Muntaril produced the movie Skirtsicine with others. Monsterscar was one of the producers of the movie Featsaw. Monsterscar produced the movie Premercy with others. Monsterscar produced the movie Zalate with others. Teeplemole was one of the producers of the movie Featsaw. Teeplemole produced the movie Zalate with others. Muntaril produced the movie Premercy with others. Monsterscar wrote for the movie Premercy. Muntaril was one of the writers for the movie Zalate. Muntaril wrote for the movie Featsaw. Teeplemole wrote for the movie Featsaw. Monsterscar was one of the writers for the movie Zalate. Teeplemole was one of the writers for the movie Skirtsicine.
QC: What awards have movies produced by people born in 1910 won?
QS: The people born in 1910 were Muntaril and Teeplemole. Teeplemole produced the movies Featsaw and Zalate. Muntaril produced the movie Premercy. Featsaw was awarded the Zorgion award. Premercy was awarded the Chowwurst award. Zalate was awarded the Hallowcock award. So the answer is ["Zorgion", "Chowwurst", "Hallowcock"].
A: ["Zorgion", "Chowwurst", "Hallowcock"]
QS: [EOQ]

movie: Misgendery ; directed by: Wetherality. movie: Dewbar ; director: Gigabut. movie: Caudacite ; director: Lougerière. movie: Tayenne ; directed by: Lougerière. movie: Misgendery ; awarded: Microsouenesis. movie: Dewbar ; awarded: Erowid. movie: Tayenne ; awarded: Cockspit. movie: Caudacite ; award: Erowid. award: Aniconder ; winner: Wetherality. award: Aniconder ; winner: Lougerière. person: Gigabut ; award: Trifogation. movie: Dewbar ; release year: 1991. movie: Tayenne ; year: 2013. movie: Caudacite ; release year: 2008. movie: Misgendery ; year: 1991. Wetherality was an actor in the movie Dewbar. Gigabut was an actor in the movie Tayenne. Lougerière was an actor in the movie Tayenne. Lougerière acted in the movie Caudacite. Lougerière acted in the movie Misgendery. Gigabut was an actor in the movie Caudacite. Wetherality was an actor in the movie Misgendery. Wetherality was born in the year 1917. Lougerière was born in 1926. Gigabut was born in the year 1917. Gigabut grew up in the nation of Triclops. Lougerière is from the country of Tatkin. Wetherality grew up in the nation of Tatkin. Lougerière produced the movie Dewbar with others. Gigabut produced the movie Tayenne with others. Gigabut produced the movie Dewbar with others. Lougerière was one of the producers of the movie Misgendery. Wetherality was one of the producers of the movie Caudacite. Gigabut was one of the producers of the movie Caudacite. Wetherality produced the movie Misgendery with others. Wetherality produced the movie Tayenne with others. Wetherality wrote for the movie

Tayenne. Gigabut wrote for the movie Misgendery. Lougerière was one of the writers for the movie Caudacite. Wetherality wrote for the movie Misgendery. Gigabut wrote for the movie Tayenne. Gigabut wrote for the movie Dewbar. Lougerière wrote for the movie Dewbar. Wetherality wrote for the movie Caudacite.

QC: What awards have the actors of the Erowid winning movies received?

QS: The movies that won the Erowid award are Dewbar and Caudacite. Wetherality was an actor in Dewbar. Lougerière and Gigabut acted in Caudacite. Wetherality won the Aniconder award. Lougerière also won the Aniconder award. Gigabut won the Trifogation award. So the answer is ["Aniconder", "Trifogation"].

A: ["Aniconder", "Trifogation"]

QS: [EOQ]

movie: Pastillobox ; directed by: Firmline. movie: Clenestration ; directed by: Carblock. movie: Pestok ; directed by: Bioperatology. movie: Vitrilateral ; director: Bioperatology. movie: Vitrilateral ; award: Antipositive. movie: Clenestration ; awarded: Handt. movie: Pastillobox ; awarded: Handt. movie: Pestok ; awarded: Gutney. movie: Pestok ; writer: Firmline. movie: Clenestration ; written by: Carblock. movie: Pastillobox ; written by: Bioperatology. movie: Pestok ; writer: Bioperatology. movie: Clenestration ; written by: Firmline. movie: Vitrilateral ; writer: Bioperatology. movie: Pastillobox ; writer: Carblock. movie: Vitrilateral ; written by: Carblock. movie: Pestok ; release year: 1986. movie: Clenestration ; year: 1986. movie: Vitrilateral ; year: 1999. movie: Pastillobox ; release year: 1984. Carblock was an actor in the movie Pastillobox. Firmline was an actor in the movie Vitrilateral. Bioperatology was an actor in the movie Clenestration. Firmline acted in the movie Pastillobox. Carblock was an actor in the movie Clenestration. Bioperatology was an actor in the movie Pestok. Firmline was born in the year 1904. Bioperatology was born in the year 1935. Carblock was born in 1935. Carblock grew up in the nation of Knoppock. Firmline grew up in the nation of Tatkin. Bioperatology grew up in the nation of Tatkin. Bioperatology won the Modiparity award. Halfbill was awarded to Firmline. Halfbill was awarded to Carblock. Bioperatology was one of the producers of the movie Pestok. Bioperatology produced the movie Vitrilateral with others. Firmline produced the movie Pastillobox with others. Firmline produced the movie Clenestration with others. Carblock was one of the producers of the movie Pastillobox. Carblock produced the movie Vitrilateral with others. Carblock produced the movie Clenestration with others. Firmline was one of the producers of the movie Pestok.

QC: What awards did the movies directed by the Modiparity winners receive?

QS: The Modiparity winners are Bioperatology. Bioperatology has directed the movies Pestok and Vitrilateral. Pestok has won the Gutney award. Vitrilateral has won the Antipositive award. So the answer is ["Gutney", "Antipositive"].

A: ["Gutney", "Antipositive"]

QS: [EOQ]

movie: Nohit ; director: Mimicocycle. movie: Noenometer ; director: Mimicocycle. movie: Tayenne ; directed by: Zayage. movie: Pneumodendron ; director: Sclerocybin. movie: Tayenne ; awarded: Goosehead. movie: Nohit ; awarded: Handt. movie: Pneumodendron ; award: Handt. movie: Noenometer ; awarded: Brownbeard. movie: Nohit ; writer: Mimicocycle. movie: Noenometer ; written by: Sclerocybin. movie: Tayenne ; writer: Sclerocybin. movie: Pneumodendron ; written by: Zayage. movie: Tayenne ; writer: Zayage. movie: Pneumodendron ; written by: Mimicocycle. movie: Noenometer ; release year: 1991. movie: Tayenne ; year: 2013. movie: Nohit ; year: 2005. movie: Pneumodendron ; year: 2005. Mimicocycle was an actor in the movie Tayenne. Zayage acted in the movie Pneumodendron. Zayage was an actor in the movie Nohit. Sclerocybin was an actor in the movie Nohit. Sclerocybin was an actor in the movie Tayenne. Mimicocycle was an actor in the movie Noenometer. Zayage was born in 1935. Sclerocybin was born in 1935. Mimicocycle was born in 1930. Mimicocycle is from the country of Calderita. Sclerocybin grew up in the nation of Calderita. Zayage is from the country of Obility. Quinion was awarded to Zayage. Fannyxist was awarded to Sclerocybin. Fannyxist was awarded to Mimicocycle. Mimicocycle produced the movie Nohit with others. Zayage was one of the producers of the movie Nohit. Sclerocybin was one of the producers of the movie Tayenne. Sclerocybin produced the movie Pneumodendron with others. Zayage produced the movie Pneumodendron with others.

Mimicocycle was one of the producers of the movie Tayenne. Sclerocybin was one of the producers of the movie Noenometer. Zayage produced the movie Noenometer with others

QC: What awards have movies written by people born in 1935 won?

QS: The people born in 1935 were Sclerocybin and Zayage. Sclerocybin has written the movies Noenometer and Tayenne. Zayage has written the movies Pneumodendron and Tayenne. Noenometer has won the Brownbeard award. Tayenne has won the Goosehead award. Pneumodendron has won the Handt award. So the answer is [”Brownbeard”, ”Goosehead”, ” Handt”].

A: [”Brownbeard”, ”Goosehead”, ”Handt”]

QS: [EOQ]

## G.4 SHORTER PROMPTS FOR SMALLER CONTEXT WINDOWS



`qa(small)`



movie: Premercy ; directed by: Muntaril. movie: Skirtsicine ; director: Teeplemole. movie: Featsaw ; directed by: Monsterscar. movie: Zalate ; director: Monsterscar. movie: Zalate ; awarded: Hallowcock. movie: Featsaw ; awarded: Zorgion. movie: Premercy ; award: Chowwurst. movie: Skirtsicine ; award: Hallowcock. award: Goatfly ; winner: Teeplemole. person: Monsterscar ; award: Glodome. person: Muntaril ; award: Goatfly. movie: Featsaw ; release year: 1973. movie: Zalate ; release year: 1964. movie: Skirtsicine ; release year: 1973. movie: Premercy ; year: 1961. Teeplemole was an actor in the movie Skirtsicine. Muntaril was an actor in the movie Skirtsicine. Monsterscar was an actor in the movie Premercy. Muntaril was an actor in the movie Featsaw. Teeplemole was an actor in the movie Zalate. Muntaril was born in the year 1910. Teeplemole was born in 1910. Monsterscar was born in 1942. Teeplemole is from the country of Piperfish. Monsterscar is from the country of Piperfish. Muntaril is from the country of Clony. Muntaril produced the movie Skirtsicine with others. Monsterscar was one of the producers of the movie Featsaw. Monsterscar produced the movie Premercy with others. Monsterscar produced the movie Zalate with others. Teeplemole was one of the producers of the movie Featsaw. Teeplemole produced the movie Zalate with others. Muntaril produced the movie Premercy with others. Monsterscar wrote for the movie Premercy. Muntaril was one of the writers for the movie Zalate. Muntaril wrote for the movie Featsaw. Teeplemole wrote for the movie Featsaw. Monsterscar was one of the writers for the movie Zalate. Teeplemole was one of the writers for the movie Skirtsicine.

Q: Which awards were given to Zalate?

A: [”Hallowcock”]

Q: For which movies was Muntaril the producer?

A: [”Premercy]

Q: Who are the actors in the movie Premercy?

A: [”Monsterscar”]

movie: Nilitude ; director: Monsterscar. movie: Dewbar ; directed by: Metatoun. movie: Warpstone ; directed by: Gastrat. movie: Partnershipmaker ; director: Metatoun. movie: Dewbar ; award: Tachychronograph. movie: Partnershipmaker ; awarded: Tachychronograph. movie: Nilitude ; award: Paleodactyl. movie: Warpstone ; award: Sabonade. person: Gastrat ; award: Trifogation. award: Polyquadrase ; winner: Monsterscar. award: Trifogation ; winner: Metatoun. movie: Warpstone ; release year: 1956. movie: Dewbar ; release year: 1984. movie: Nilitude ; year: 1984. movie: Partnershipmaker ; year: 1962. Gastrat was an actor in the movie Partnershipmaker. Metatoun was an actor in the movie Partnershipmaker. Metatoun was an actor in the movie Nilitude. Gastrat acted in the movie Nilitude. Monsterscar was an actor in the movie Dewbar. Gastrat acted in the movie Warpstone. Metatoun acted in the movie Warpstone. Metatoun was born in 1939. Gastrat was born in the year 1933. Monsterscar was born in 1933. Metatoun grew up in the nation of Moulole. Gastrat is from the country of Stridery. Monsterscar grew up in the nation of Moulole. Monsterscar produced the movie Nilitude with others. Monsterscar was one of the producers of the movie Warpstone. Metatoun was one of the producers of the movie Warpstone. Gastrat was one of the producers of the movie Nilitude. Metatoun produced the movie Partnershipmaker with others. Metatoun produced the movie Dewbar with others. Monsterscar was one of the producers of

the movie Partnershipmaker. Gastrat produced the movie Dewbar with others. Metatoun wrote for the movie Partnershipmaker. Gastrat wrote for the movie Warpstone. Gastrat was one of the writers for the movie Dewbar. Monsterscar was one of the writers for the movie Nilitude. Metatoun wrote for the movie Warpstone.

Q: Which movies has Gastrat been an actor in?
A: ["Partnershipmaker", "Nilitude", "Warpstone"]

Q: Who is from the country Stridery?
A: ["Gastrat"]

Q: Which movies were given the Tachychronograph award?
A: ["Dewbar", "Partnershipmaker"]



`aw_qa(small)`



movie: Premercy ; directed by: Muntaril. movie: Skirtsicine ; director: Teeplemole. movie: Featsaw ; directed by: Monsterscar. movie: Zalate ; director: Monsterscar. movie: Zalate ; awarded: Hallowcock. movie: Featsaw ; awarded: Zorgion. movie: Premercy ; award: Chowwurst. movie: Skirtsicine ; award: Hallowcock. award: Goatfly ; winner: Teeplemole. person: Monsterscar ; award: Glodome. person: Muntaril ; award: Goatfly. movie: Featsaw ; release year: 1973. movie: Zalate ; release year: 1964. movie: Skirtsicine ; release year: 1973. movie: Premercy ; year: 1961. Teeplemole was an actor in the movie Skirtsicine. Muntaril was an actor in the movie Skirtsicine. Monsterscar was an actor in the movie Premercy. Muntaril was an actor in the movie Featsaw. Teeplemole was an actor in the movie Zalate. Muntaril was born in the year 1910. Teeplemole was born in 1910. Monsterscar was born in 1942. Teeplemole is from the country of Piperfish. Monsterscar is from the country of Piperfish. Muntaril is from the country of Clony. Muntaril produced the movie Skirtsicine with others. Monsterscar was one of the producers of the movie Featsaw. Monsterscar produced the movie Premercy with others. Monsterscar produced the movie Zalate with others. Teeplemole was one of the producers of the movie Featsaw. Teeplemole produced the movie Zalate with others. Muntaril produced the movie Premercy with others. Monsterscar wrote for the movie Premercy. Muntaril was one of the writers for the movie Zalate. Muntaril wrote for the movie Featsaw. Teeplemole wrote for the movie Featsaw. Monsterscar was one of the writers for the movie Zalate. Teeplemole was one of the writers for the movie Skirtsicine.

Q: Which awards were given to Zalate?
A: ["Hallowcock"]

Q: Which awards were given to Premercy?
A: ["Chowwurst"]

movie: Misgendery ; directed by: Wetherality. movie: Dewbar ; director: Gigabut. movie: Caudacite ; director: Lougerière. movie: Tayenne ; directed by: Lougerière. movie: Misgendery ; awarded: Microsouenesis. movie: Dewbar ; awarded: Erowid. movie: Tayenne ; awarded: Cockspit. movie: Caudacite ; award: Erowid. award: Aniconder ; winner: Wetherality. award: Aniconder ; winner: Lougerière. person: Gigabut ; award: Trifogation. movie: Dewbar ; release year: 1991. movie: Tayenne ; year: 2013. movie: Caudacite ; release year: 2008. movie: Misgendery ; year: 1991. Wetherality was an actor in the movie Dewbar. Gigabut was an actor in the movie Tayenne. Lougerière was an actor in the movie Tayenne. Lougerière acted in the movie Caudacite. Lougerière acted in the movie Misgendery. Gigabut was an actor in the movie Caudacite. Wetherality was an actor in the movie Misgendery. Wetherality was born in the year 1917. Lougerière was born in 1926. Gigabut was born in the year 1917. Gigabut grew up in the nation of Triclops. Lougerière is from the country of Tatkin. Wetherality grew up in the nation of Tatkin. Lougerière produced the movie Dewbar with others. Gigabut produced the movie Tayenne with others. Gigabut produced the movie Dewbar with others. Lougerière was one of the producers of the movie Misgendery. Wetherality was one of the producers of the movie Caudacite. Gigabut was one of the producers of the movie Caudacite. Wetherality produced the movie Misgendery with others. Wetherality produced the movie Tayenne with others. Wetherality wrote for the movie Tayenne. Gigabut wrote for the movie Misgendery. Lougerière was one of the writers for the movie Caudacite. Wetherality wrote for the movie Misgendery. Gigabut wrote for the movie

Tayenne. Gigabut wrote for the movie Dewbar. Lougerière wrote for the movie Dewbar. Wetherality wrote for the movie Caudacite.

Q: Which movies were given the Erowid award?
A: ["Dewbar", "Caudacite"]

Q: Which awards were given to Wetherality?
A: ["Aniconder"]



`pos_qa(small)`



movie: Premercy ; directed by: Muntaril. movie: Skirtsicine ; director: Teeplemole. movie: Featsaw ; directed by: Monsterscar. movie: Zalate ; director: Monsterscar. movie: Zalate ; awarded: Hallowcock. movie: Featsaw ; awarded: Zorgion. movie: Premercy ; award: Chowwurst. movie: Skirtsicine ; award: Hallowcock. award: Goatfly ; winner: Teeplemole. person: Monsterscar ; award: Glodome. person: Muntaril ; award: Goatfly. movie: Featsaw ; release year: 1973. movie: Zalate ; release year: 1964. movie: Skirtsicine ; release year: 1973. movie: Premercy ; year: 1961. Teeplemole was an actor in the movie Skirtsicine. Muntaril was an actor in the movie Skirtsicine. Monsterscar was an actor in the movie Premercy. Muntaril was an actor in the movie Featsaw. Teeplemole was an actor in the movie Zalate. Muntaril was born in the year 1910. Teeplemole was born in 1910. Monsterscar was born in 1942. Teeplemole is from the country of Piperfish. Monsterscar is from the country of Piperfish. Muntaril is from the country of Clony. Muntaril produced the movie Skirtsicine with others. Monsterscar was one of the producers of the movie Featsaw. Monsterscar produced the movie Premercy with others. Monsterscar produced the movie Zalate with others. Teeplemole was one of the producers of the movie Featsaw. Teeplemole produced the movie Zalate with others. Muntaril produced the movie Premercy with others. Monsterscar wrote for the movie Premercy. Muntaril was one of the writers for the movie Zalate. Muntaril wrote for the movie Featsaw. Teeplemole wrote for the movie Featsaw. Monsterscar was one of the writers for the movie Zalate. Teeplemole was one of the writers for the movie Skirtsicine.
Q: For which movies was Teeplemole the producer?
A: ["Featsaw", "Zalate"]

Q: For which movies was Muntaril the producer?
A: ["Premercy]

movie: Nilitude ; director: Monsterscar. movie: Dewbar ; directed by: Metatoun. movie: Warpstone ; directed by: Gastrat. movie: Partnershipmaker ; director: Metatoun. movie: Dewbar ; award: Tachychronograph. movie: Partnershipmaker ; awarded: Tachychronograph. movie: Nilitude ; award: Paleodactyl. movie: Warpstone ; award: Sabonade. person: Gastrat ; award: Trifogation. award: Polyquadrase ; winner: Monsterscar. award: Trifogation ; winner: Metatoun. movie: Warpstone ; release year: 1956. movie: Dewbar ; release year: 1984. movie: Nilitude ; year: 1984. movie: Partnershipmaker ; year: 1962. Gastrat was an actor in the movie Partnershipmaker. Metatoun was an actor in the movie Partnershipmaker. Metatoun was an actor in the movie Nilitude. Gastrat acted in the movie Nilitude. Monsterscar was an actor in the movie Dewbar. Gastrat acted in the movie Warpstone. Metatoun acted in the movie Warpstone. Metatoun was born in 1939. Gastrat was born in the year 1933. Monsterscar was born in 1933. Metatoun grew up in the nation of Moulole. Gastrat is from the country of Stridery. Monsterscar grew up in the nation of Moulole. Monsterscar produced the movie Nilitude with others. Monsterscar was one of the producers of the movie Warpstone. Metatoun was one of the producers of the movie Warpstone. Gastrat was one of the producers of the movie Nilitude. Metatoun produced the movie Partnershipmaker with others. Metatoun produced the movie Dewbar with others. Monsterscar was one of the producers of the movie Partnershipmaker. Gastrat produced the movie Dewbar with others. Metatoun wrote for the movie Partnershipmaker. Gastrat wrote for the movie Warpstone. Gastrat was one of the writers for the movie Dewbar. Monsterscar was one of the writers for the movie Nilitude. Metatoun wrote for the movie Warpstone.
Q: Which movies has Gastrat been an actor in?
A: ["Partnershipmaker", "Nilitude", "Warpstone"]

Q: Which movies has Monsterscar been an actor in?

A: ["Dewbar"]



`simp_qa(small)`


movie: Premercy ; directed by: Muntaril. movie: Skirtsicine ; director: Teeplemole. movie: Featsaw ; directed by: Monsterscar. movie: Zalate ; director: Monsterscar. movie: Zalate ; awarded: Hallowcock. movie: Featsaw ; awarded: Zorgion. movie: Premercy ; award: Chowwurst. movie: Skirtsicine ; award: Hallowcock. award: Goatfly ; winner: Teeplemole. person: Monsterscar ; award: Glodome. person: Muntaril ; award: Goatfly. movie: Featsaw ; release year: 1973. movie: Zalate ; release year: 1964. movie: Skirtsicine ; release year: 1973. movie: Premercy ; year: 1961. Teeplemole was an actor in the movie Skirtsicine. Muntaril was an actor in the movie Skirtsicine. Monsterscar was an actor in the movie Premercy. Muntaril was an actor in the movie Featsaw. Teeplemole was an actor in the movie Zalate. Muntaril was born in the year 1910. Teeplemole was born in 1910. Monsterscar was born in 1942. Teeplemole is from the country of Piperfish. Monsterscar is from the country of Piperfish. Muntaril is from the country of Clony. Muntaril produced the movie Skirtsicine with others. Monsterscar was one of the producers of the movie Featsaw. Monsterscar produced the movie Premercy with others. Monsterscar produced the movie Zalate with others. Teeplemole was one of the producers of the movie Featsaw. Teeplemole produced the movie Zalate with others. Muntaril produced the movie Premercy with others. Monsterscar wrote for the movie Premercy. Muntaril was one of the writers for the movie Zalate. Muntaril wrote for the movie Featsaw. Teeplemole wrote for the movie Featsaw. Monsterscar was one of the writers for the movie Zalate. Teeplemole was one of the writers for the movie Skirtsicine.
Q: Who were born in the year 1910?
A: ["Teeplemole", "Muntaril"]

Q: Who is from the country Piperfish?
A: ["Teeplemole", "Monsterscar"]

movie: Nilitude ; director: Monsterscar. movie: Dewbar ; directed by: Metatoun. movie: Warpstone ; directed by: Gastrat. movie: Partnershipmaker ; director: Metatoun. movie: Dewbar ; award: Tachychronograph. movie: Partnershipmaker ; awarded: Tachychronograph. movie: Nilitude ; award: Paleodactyl. movie: Warpstone ; award: Sabonade. person: Gastrat ; award: Trifogation. award: Polyquadrase ; winner: Monsterscar. award: Trifogation ; winner: Metatoun. movie: Warpstone ; release year: 1956. movie: Dewbar ; release year: 1984. movie: Nilitude ; year: 1984. movie: Partnershipmaker ; year: 1962. Gastrat was an actor in the movie Partnershipmaker. Metatoun was an actor in the movie Partnershipmaker. Metatoun was an actor in the movie Nilitude. Gastrat acted in the movie Nilitude. Monsterscar was an actor in the movie Dewbar. Gastrat acted in the movie Warpstone. Metatoun acted in the movie Warpstone. Metatoun was born in 1939. Gastrat was born in the year 1933. Monsterscar was born in 1933. Metatoun grew up in the nation of Moulole. Gastrat is from the country of Stridery. Monsterscar grew up in the nation of Moulole. Monsterscar produced the movie Nilitude with others. Monsterscar was one of the producers of the movie Warpstone. Metatoun was one of the producers of the movie Warpstone. Gastrat was one of the producers of the movie Nilitude. Metatoun produced the movie Partnershipmaker with others. Metatoun produced the movie Dewbar with others. Monsterscar was one of the producers of the movie Partnershipmaker. Gastrat produced the movie Dewbar with others. Metatoun wrote for the movie Partnershipmaker. Gastrat wrote for the movie Warpstone. Gastrat was one of the writers for the movie Dewbar. Monsterscar was one of the writers for the movie Nilitude. Metatoun wrote for the movie Warpstone.
Q: Who is from the country Stridery?
A: ["Gastrat"]

Q: Who were born in the year 1939?
A: ["Metatoun"]



`COT(small)`


movie: Premercy ; directed by: Muntaril. movie: Skirtsicine ; director: Teeplemole. movie: Featsaw ; directed by: Monsterscar. movie: Zalate ; director: Monsterscar. movie: Zalate ; awarded:

Hallowcock. movie: Featsaw ; awarded: Zorgion. movie: Premercy ; award: Chowwurst. movie: Skirtsicine ; award: Hallowcock. award: Goatfly ; winner: Teeplemole. person: Monsterscar ; award: Glodome. person: Muntaril ; award: Goatfly. movie: Featsaw ; release year: 1973. movie: Zalate ; release year: 1964. movie: Skirtsicine ; release year: 1973. movie: Premercy ; year: 1961. Teeplemole was an actor in the movie Skirtsicine. Muntaril was an actor in the movie Skirtsicine. Monsterscar was an actor in the movie Premercy. Muntaril was an actor in the movie Featsaw. Teeplemole was an actor in the movie Zalate. Muntaril was born in the year 1910. Teeplemole was born in 1910. Monsterscar was born in 1942. Teeplemole is from the country of Piperfish. Monsterscar is from the country of Piperfish. Muntaril is from the country of Clony. Muntaril produced the movie Skirtsicine with others. Monsterscar was one of the producers of the movie Featsaw. Monsterscar produced the movie Premercy with others. Monsterscar produced the movie Zalate with others. Teeplemole was one of the producers of the movie Featsaw. Teeplemole produced the movie Zalate with others. Muntaril produced the movie Premercy with others. Monsterscar wrote for the movie Premercy. Muntaril was one of the writers for the movie Zalate. Muntaril wrote for the movie Featsaw. Teeplemole wrote for the movie Featsaw. Monsterscar was one of the writers for the movie Zalate. Teeplemole was one of the writers for the movie Skirtsicine.

QC: What awards have movies produced by people born in 1910 won?

QS: The people born in 1910 were Muntaril and Teeplemole. Teeplemole produced the movies Featsaw and Zalate. Muntaril produced the movie Premercy. Featsaw was awarded the Zorgion award. Premercy was awarded the Chowwurst award. Zalate was awarded the Hallowcock award. So the answer is ["Zorgion", "Chowwurst", "Hallowcock"].

A: ["Zorgion", "Chowwurst", "Hallowcock"]

QS: [EOQ]

movie: Nilitude ; director: Monsterscar. movie: Dewbar ; directed by: Metatoun. movie: Warpstone ; directed by: Gastrat. movie: Partnershipmaker ; director: Metatoun. movie: Dewbar ; award: Tachychronograph. movie: Partnershipmaker ; awarded: Tachychronograph. movie: Nilitude ; award: Paleodactyl. movie: Warpstone ; award: Sabonade. person: Gastrat ; award: Trifogation. award: Polyquadrase ; winner: Monsterscar. award: Trifogation ; winner: Metatoun. movie: Warpstone ; release year: 1956. movie: Dewbar ; release year: 1984. movie: Nilitude ; year: 1984. movie: Partnershipmaker ; year: 1962. Gastrat was an actor in the movie Partnershipmaker. Metatoun was an actor in the movie Partnershipmaker. Metatoun was an actor in the movie Nilitude. Gastrat acted in the movie Nilitude. Monsterscar was an actor in the movie Dewbar. Gastrat acted in the movie Warpstone. Metatoun acted in the movie Warpstone. Metatoun was born in 1939. Gastrat was born in the year 1933. Monsterscar was born in 1933. Metatoun grew up in the nation of Moulole. Gastrat is from the country of Stridery. Monsterscar grew up in the nation of Moulole. Monsterscar produced the movie Nilitude with others. Monsterscar was one of the producers of the movie Warpstone. Metatoun was one of the producers of the movie Warpstone. Gastrat was one of the producers of the movie Nilitude. Metatoun produced the movie Partnershipmaker with others. Metatoun produced the movie Dewbar with others. Monsterscar was one of the producers of the movie Partnershipmaker. Gastrat produced the movie Dewbar with others. Metatoun wrote for the movie Partnershipmaker. Gastrat wrote for the movie Warpstone. Gastrat was one of the writers for the movie Dewbar. Monsterscar was one of the writers for the movie Nilitude. Metatoun wrote for the movie Warpstone.

QC: What movies have people from the country Stridery acted in?

QS: The people born in Stridery are Gastrat. Gastrat acted in Partenrshipmaker, Nilitude and Warpstone. So the answer is ["Partnershipmaker", "Nilitude", "Warpstone"].

A: ["Partnershipmaker", "Nilitude", "Warpstone"]

QS: [EOQ]

## G.5 MATH QA

The decomposer here deterministically calls `cot` to generate the CoT and then calls `gpt_ans` to extract the answer.

`cot`

Q: There are 15 trees in the grove. Grove workers will plant trees in the grove today. After they are done, there will be 21 trees. How many trees did the grove workers plant today?
A: There are 15 trees originally. Then there were 21 trees after some more were planted. So there must have been $21 - 15 = 6$. The answer is 6.

Q: If there are 3 cars in the parking lot and 2 more cars arrive, how many cars are in the parking lot?
A: There are originally 3 cars. 2 more cars arrive. $3 + 2 = 5$. The answer is 5.

Q: Leah had 32 chocolates and her sister had 42. If they ate 35, how many pieces do they have left in total?
A: Originally, Leah had 32 chocolates. Her sister had 42. So in total they had $32 + 42 = 74$. After eating 35, they had $74 - 35 = 39$. The answer is 39.

Q: Jason had 20 lollipops. He gave Denny some lollipops. Now Jason has 12 lollipops. How many lollipops did Jason give to Denny?
A: Jason started with 20 lollipops. Then he had 12 after giving some to Denny. So he gave Denny $20 - 12 = 8$. The answer is 8.

Q: Shawn has five toys. For Christmas, he got two toys each from his mom and dad. How many toys does he have now?
A: Shawn started with 5 toys. If he got 2 toys each from his mom and dad, then that is 4 more toys. $5 + 4 = 9$. The answer is 9.

Q: There were nine computers in the server room. Five more computers were installed each day, from monday to thursday. How many computers are now in the server room?
A: There were originally 9 computers. For each of 4 days, 5 more computers were added. So $5 * 4 = 20$ computers were added. $9 + 20$ is 29. The answer is 29.

Q: Michael had 58 golf balls. On tuesday, he lost 23 golf balls. On wednesday, he lost 2 more. How many golf balls did he have at the end of wednesday?
A: Michael started with 58 golf balls. After losing 23 on tuesday, he had $58 - 23 = 35$. After losing 2 more, he had $35 - 2 = 33$ golf balls. The answer is 33.

Q: Olivia has $23. She bought five bagels for $3 each. How much money does she have left?
A: Olivia had 23 dollars. 5 bagels for 3 dollars each will be 5 x $3 = 15$ dollars. So she has $23 - 15$ dollars left. $23 - 15$ is 8. The answer is 8.

```
gpt_ans
```

Q: There are 15 trees originally. Then there were 21 trees after some more were planted. So there must have been $21 - 15 = 6$ trees planted.
A: 6

Q: There are originally 3 cars. 2 more cars arrive. $3 + 2 = 5$. The answer is 5.
A: 5

Q: Originally, Leah had 32 chocolates. Her sister had 42. So in total they had $32 + 42 = 74$. After eating 35, they had $74 - 35 = 39$. The answer is 39.
A: 39

Q: Jason started with 20 lollipops. Then he had 12 after giving some to Denny. So he gave Denny $20 - 12 = 8$ apples.
A: 8

Q: Shawn started with 5 toys. If he got 2 toys each from his mom and dad, then that is 4 more toys. $5 + 4 = 9$. The answer is 9.
A: 9

Q: There were originally 9 computers. For each of 4 days, 5 more computers were added. So $5 * 4 = 20$ computers were added. $9 + 20$ is 29. The answer is 29.
A: 29

Q: Michael started with 58 golf balls. After losing 23 on tuesday, he had $58 - 23 = 35$. After losing 2 more, he had $35 - 2 = 33$ golf balls.
A: 33

Q: Olivia had 23 dollars. 5 bagels for 3 dollars each will be $5 \times 3 = 15$ dollars. So she has $23 - 15 = 8$ dollars left.
A: 8

G.6  OPEN DOMAIN QA

The prompts in this section implement Decomposed Prompting approach to open-domain multihop QA. For brevity we've included prompts for 5 of 20 randomly sampled questions. The full prompts are attached with the submission and will also be released with the code. Note that we selected a set of 100 questions from the development set to tune the hyperparameter (number of paragraphs to retrieve for all of the retrieval-based approaches).

```
hotpotqa: decomp
```
QC: In which country did this Australian who was detained in Guantanamo Bay detention camp and published "Guantanamo: My Journey" receive para–military training?
QS: (select) [retrieve_odqa] Who is the Australian who was detained in Guantanamo Bay detention camp and published "Guantanamo: My Journey"?
A: {"titles": ["Guantanamo: My Journey", "Bismullah v. Gates", "Guantanamo Bay detention camp"], "answer": ["David Hicks"]}
QS: (select) [retrieve_odqa] In which country did David Hicks receive his para–military training?
A: {"titles": ["John Adams Project", "Camp Echo (Guantanamo Bay)", "Guantanamo Bay Museum of Art and History", "David Hicks"], "answer": ["Afghanistan"]}
QS: (select) [multihop_titleqa] Titles: ["Guantanamo: My Journey", "Bismullah v. Gates", "Guantanamo Bay detention camp", "John Adams Project", "Camp Echo (Guantanamo Bay)", "Guantanamo Bay Museum of Art and History", "David Hicks"]. Question: In which country did this Australian who was detained in Guantanamo Bay detention camp and published "Guantanamo: My Journey" receive para–military training?
A: ["Afghanistan"]
QS: [EOQ]

QC: How many awards did the "A Girl Like Me" singer win at the American Music Awards of 2012?
QS: (select) [retrieve_odqa] Who is the singer of "A Girl Like Me"?
A: {"titles": ["American Music Award", "Rihanna"], "answer": ["Rihanna"]}
QS: (select) [retrieve_odqa] How many awards did Rihana won in the American Music Awards of 2012?
A: {"titles": ["American Music Awards of 2012", "APRA Music Awards of 2012", "List of awards and nominations received by TLC", "Native American Music Awards"], "answer": ["one"]}
QS: (select) [multihop_titleqa] Titles: ["American Music Award", "Rihanna", "American Music Awards of 2012", "APRA Music Awards of 2012", "List of awards and nominations received by TLC", "Native American Music Awards"]. Question: How many awards did the "A Girl Like Me" singer win at the American Music Awards of 2012?
A: ["one"]
QS: [EOQ]

QC: who is older Jeremy Horn or Renato Sobral ?
QS: (select) [retrieve_odqa] When was Jeremy Horn born?

A: {"titles": ["Zaza Tkeshelashvili", "Jeremy Horn", "Jeremy Horn (singer)", "Ricardo Arona"], "answer": ["August 25, 1975"]}
QS: (select) [retrieve_odqa] When was Renato Sobral born?
A: {"titles": ["Brian Warren", "Renato Sobral"], "answer": ["September 7, 1975"]}
QS: (select) [multihop_titleqa] Titles: ["Zaza Tkeshelashvili", "Jeremy Horn", "Jeremy Horn (singer)", "Ricardo Arona", "Brian Warren", "Renato Sobral"]. Question: who is older Jeremy Horn or Renato Sobral ?
A: ["Jeremy Horn"]
QS: [EOQ]

QC: What was the 2014 population of the city where Lake Wales Medical Center is located?
QS: (select) [retrieve_odqa] Lake Wales Medical Center is located in what city?
A: {"titles": ["Baylor College of Medicine", "Lake Wales Medical Center", "Tufts University School of Medicine", "Hanford Community Medical Center"], "answer": ["Polk County, Florida"]}
QS: (select) [retrieve_odqa] What was the population of Polk County in 2014?
A: {"titles": ["Banner University Medical Center Tucson", "Lake Wales, Florida"], "answer": ["15,140"]}
QS: (select) [multihop_titleqa] Titles: ["Baylor College of Medicine", "Lake Wales Medical Center", "Tufts University School of Medicine", "Hanford Community Medical Center", "Banner University Medical Center Tucson", "Lake Wales, Florida"]. Question: What was the 2014 population of the city where Lake Wales Medical Center is located?
A: ["15,140"]
QS: [EOQ]

QC: Nobody Loves You was written by John Lennon and released on what album that was issued by Apple Records, and was written, recorded, and released during his 18 month separation from Yoko Ono?
QS: (select) [retrieve_odqa] What album was issued by Apple Records, and written, recorded, and released during John Lennon's 18 month separation from Yoko Ono?
A: {"titles": ["John Lennon/Plastic Ono Band", "Milk and Honey (album)", "Walls and Bridges"], "answer": ["Walls and Bridges"]}
QS: (select) [retrieve_odqa] Nobody Loves You was written by John Lennon on what album?
A: {"titles": ["John Lennon Museum", "Nobody Loves You (When You're Down and Out)"], "answer": ["Walls and Bridges"]}
QS: (select) [multihop_titleqa] Titles: ["John Lennon/Plastic Ono Band", "Milk and Honey (album)", "Walls and Bridges", "John Lennon Museum", "Nobody Loves You (When You're Down and Out)"]. Question: Nobody Loves You was written by John Lennon and released on what album that was issued by Apple Records, and was written, recorded, and released during his 18 month separation from Yoko Ono?
A: ["Walls and Bridges"]
QS: [EOQ]

```
hotpotqa: retrieve_odqa
```

QC: Who is the Australian who was detained in Guantanamo Bay detention camp and published "Guantanamo: My Journey"?
QS: (select) [retrieve] Who is the Australian who was detained in Guantanamo Bay detention camp and published "Guantanamo: My Journey"?
A: ["Guantanamo: My Journey", "Bismullah v. Gates", "Guantanamo Bay detention camp"]
QS: (select) [singlehop_titleqa] Titles: ["Guantanamo: My Journey", "Bismullah v. Gates", "Guantanamo Bay detention camp"]. Question: Who is the Australian who was detained in Guantanamo Bay detention camp and published "Guantanamo: My Journey"?
A: {"titles": ["Guantanamo: My Journey", "Bismullah v. Gates", "Guantanamo Bay detention camp"], "answer": ["David Hicks"]}
QS: [EOQ]

QC: In which country did David Hicks receive his para−military training?
QS: (select) [retrieve] In which country did David Hicks receive his para−military training?
A: ["John Adams Project", "Camp Echo (Guantanamo Bay)", "Guantanamo Bay Museum of Art and History", "David Hicks"]
QS: (select) [singlehop_titleqa] Titles: ["John Adams Project", "Camp Echo (Guantanamo Bay)", "Guantanamo Bay Museum of Art and History", "David Hicks"]. Question: In which country did David Hicks receive his para−military training?
A: {"titles": ["John Adams Project", "Camp Echo (Guantanamo Bay)", "Guantanamo Bay Museum of Art and History", "David Hicks"], "answer": ["Afghanistan"]}
QS: [EOQ]

QC: Who is the singer of "A Girl Like Me"?
QS: (select) [retrieve] Who is the singer of "A Girl Like Me"?
A: ["American Music Award", "Rihanna"]
QS: (select) [singlehop_titleqa] Titles: ["American Music Award", "Rihanna"]. Question: Who is the singer of "A Girl Like Me"?
A: {"titles": ["American Music Award", "Rihanna"], "answer": ["Rihanna"]}
QS: [EOQ]

QC: How many awards did Rihana won in the American Music Awards of 2012?
QS: (select) [retrieve] How many awards did Rihana won in the American Music Awards of 2012?
A: ["American Music Awards of 2012", "APRA Music Awards of 2012", "List of awards and nominations received by TLC", "Native American Music Awards"]
QS: (select) [singlehop_titleqa] Titles: ["American Music Awards of 2012", "APRA Music Awards of 2012", "List of awards and nominations received by TLC", "Native American Music Awards"]. Question: How many awards did Rihana won in the American Music Awards of 2012?
A: {"titles": ["American Music Awards of 2012", "APRA Music Awards of 2012", "List of awards and nominations received by TLC", "Native American Music Awards"], "answer": ["one"]}
QS: [EOQ]

QC: When was Jeremy Horn born?
QS: (select) [retrieve] When was Jeremy Horn born?
A: ["Zaza Tkeshelashvili", "Jeremy Horn", "Jeremy Horn (singer)", "Ricardo Arona"]
QS: (select) [singlehop_titleqa] Titles: ["Zaza Tkeshelashvili", "Jeremy Horn", "Jeremy Horn (singer)", "Ricardo Arona"]. Question: When was Jeremy Horn born?
A: {"titles": ["Zaza Tkeshelashvili", "Jeremy Horn", "Jeremy Horn (singer)", "Ricardo Arona"], "answer": ["August 25, 1975"]}
QS: [EOQ]

QC: When was Renato Sobral born?
QS: (select) [retrieve] When was Renato Sobral born?
A: ["Brian Warren", "Renato Sobral"]
QS: (select) [singlehop_titleqa] Titles: ["Brian Warren", "Renato Sobral"]. Question: When was Renato Sobral born?
A: {"titles": ["Brian Warren", "Renato Sobral"], "answer": ["September 7, 1975"]}
QS: [EOQ]

QC: Lake Wales Medical Center is located in what city?
QS: (select) [retrieve] Lake Wales Medical Center is located in what city?
A: ["Baylor College of Medicine", "Lake Wales Medical Center", "Tufts University School of Medicine", "Hanford Community Medical Center"]

QS: (select) [singlehop_titleqa] Titles: ["Baylor College of Medicine", "Lake Wales Medical Center", "Tufts University School of Medicine", "Hanford Community Medical Center"]. Question: Lake Wales Medical Center is located in what city?
A: {"titles": ["Baylor College of Medicine", "Lake Wales Medical Center", "Tufts University School of Medicine", "Hanford Community Medical Center"], "answer": ["Polk County, Florida"]}
QS: [EOQ]

QC: What was the population of Polk County in 2014?
QS: (select) [retrieve] What was the population of Polk County in 2014?
A: ["Banner University Medical Center Tucson", "Lake Wales, Florida"]
QS: (select) [singlehop_titleqa] Titles: ["Banner University Medical Center Tucson", "Lake Wales, Florida"]. Question: What was the population of Polk County in 2014?
A: {"titles": ["Banner University Medical Center Tucson", "Lake Wales, Florida"], "answer": ["15,140"]}
QS: [EOQ]

QC: What album was issued by Apple Records, and written, recorded, and released during John Lennon's 18 month separation from Yoko Ono?
QS: (select) [retrieve] What album was issued by Apple Records, and written, recorded, and released during John Lennon's 18 month separation from Yoko Ono?
A: ["John Lennon/Plastic Ono Band", "Milk and Honey (album)", "Walls and Bridges"]
QS: (select) [singlehop_titleqa] Titles: ["John Lennon/Plastic Ono Band", "Milk and Honey (album)", "Walls and Bridges"]. Question: What album was issued by Apple Records, and written, recorded, and released during John Lennon's 18 month separation from Yoko Ono?
A: {"titles": ["John Lennon/Plastic Ono Band", "Milk and Honey (album)", "Walls and Bridges"], "answer": ["Walls and Bridges"]}
QS: [EOQ]

QC: Nobody Loves You was written by John Lennon on what album?
QS: (select) [retrieve] Nobody Loves You was written by John Lennon on what album?
A: ["John Lennon Museum", "Nobody Loves You (When You're Down and Out)"]
QS: (select) [singlehop_titleqa] Titles: ["John Lennon Museum", "Nobody Loves You (When You're Down and Out)"]. Question: Nobody Loves You was written by John Lennon on what album?
A: {"titles": ["John Lennon Museum", "Nobody Loves You (When You're Down and Out)"], "answer": ["Walls and Bridges"]}
QS: [EOQ]

```
hotpotqa:  singlehop_titleqa
```

Wikipedia Title: David Hicks
<hidden for brevity>

Wikipedia Title: Guantanamo: My Journey
<hidden for brevity>

Wikipedia Title: Murat Kurnaz
<hidden for brevity>

Q: Who is the Australian who was detained in Guantanamo Bay detention camp and published "Guantanamo: My Journey"?
A: ["David Hicks"]

Q: In which country did David Hicks receive his para−military training?
A: ["Afghanistan"]

Wikipedia Title: American Music Award
<hidden for brevity>

Wikipedia Title: Rihanna
<hidden for brevity>

Wikipedia Title: American Music Awards of 2012
<hidden for brevity>

Q: Who is the singer of "A Girl Like Me"?
A: ["Rihanna"]

Q: How many awards did Rihana won in the American Music Awards of 2012?
A: ["one"]

Wikipedia Title: Renato Sobral
<hidden for brevity>

Wikipedia Title: Zaza Tkeshelashvili
<hidden for brevity>

Wikipedia Title: Jeremy Horn
<hidden for brevity>

Q: When was Jeremy Horn born?
A: ["August 25, 1975"]

Q: When was Renato Sobral born?
A: ["September 7, 1975"]

Wikipedia Title: Lake Wales Medical Center
<hidden for brevity>

Wikipedia Title: Lake Wales, Florida
<hidden for brevity>

Wikipedia Title: Tufts University School of Medicine
<hidden for brevity>

Q: Lake Wales Medical Center is located in what city?
A: ["Polk County, Florida"]

Q: What was the population of Polk County in 2014?
A: ["15,140"]

Wikipedia Title: Nobody Loves You (When You're Down and Out)
<hidden for brevity>

Wikipedia Title: Walls and Bridges
<hidden for brevity>

Wikipedia Title: Mother (John Lennon song)
<hidden for brevity>

Q: What album was issued by Apple Records, and written, recorded, and released during John
   Lennon's 18 month separation from Yoko Ono?
A: ["Walls and Bridges"]

Q: Nobody Loves You was written by John Lennon on what album?
A: ["Walls and Bridges"]

```
hotpotqa: multihop_titleqa (direct)
```
Wikipedia Title: David Hicks
<hidden for brevity>

Wikipedia Title: Guantanamo: My Journey
<hidden for brevity>

Wikipedia Title: Murat Kurnaz
<hidden for brevity>

Q: In which country did this Australian who was detained in Guantanamo Bay detention camp and
   published "Guantanamo: My Journey" receive para−military training?
A: ["Afghanistan"]

Wikipedia Title: American Music Award
<hidden for brevity>

Wikipedia Title: Rihanna
<hidden for brevity>

Wikipedia Title: American Music Awards of 2012
<hidden for brevity>

Q: How many awards did the "A Girl Like Me" singer win at the American Music Awards of
   2012?
A: ["one"]

Wikipedia Title: Renato Sobral
<hidden for brevity>

Wikipedia Title: Zaza Tkeshelashvili
<hidden for brevity>

Wikipedia Title: Jeremy Horn
<hidden for brevity>

Q: who is older Jeremy Horn or Renato Sobral ?
A: ["Jeremy Horn"]

Wikipedia Title: Lake Wales Medical Center
<hidden for brevity>

Wikipedia Title: Lake Wales, Florida
<hidden for brevity>

Wikipedia Title: Tufts University School of Medicine
<hidden for brevity>

Q: What was the 2014 population of the city where Lake Wales Medical Center is located?

A: ["15,140"]

Wikipedia Title: Nobody Loves You (When You're Down and Out)
<hidden for brevity>

Wikipedia Title: Walls and Bridges
<hidden for brevity>

Wikipedia Title: Mother (John Lennon song)
<hidden for brevity>

Q: Nobody Loves You was written by John Lennon and released on what album that was issued by
    Apple Records, and was written, recorded, and released during his 18 month separation from
    Yoko Ono?
A: ["Walls and Bridges"]

```
                           hotpotqa: multihop_titleqa (cot)
```
Wikipedia Title: David Hicks
<hidden for brevity>

Wikipedia Title: Guantanamo: My Journey
<hidden for brevity>

Wikipedia Title: Murat Kurnaz
<hidden for brevity>

Q: In which country did this Australian who was detained in Guantanamo Bay detention camp and
    published "Guantanamo: My Journey" receive para−military training?
A: The Australian who was detained in Guantanamo Bay detention camp and published "
    Guantanamo: My Journey" is David Hicks. David Hicks received his para−military training in
    Afghanistan. So the answer is: Afghanistan.

Wikipedia Title: American Music Award
<hidden for brevity>

Wikipedia Title: Rihanna
<hidden for brevity>

Wikipedia Title: American Music Awards of 2012
<hidden for brevity>

Q: How many awards did the "A Girl Like Me" singer win at the American Music Awards of
    2012?
A: The singer of "A Girl Like Me" singer is Rihanna. In the American Music Awards of 2012,
    Rihana won one award. So the answer is: one.

Wikipedia Title: Renato Sobral
<hidden for brevity>

Wikipedia Title: Zaza Tkeshelashvili
<hidden for brevity>

Wikipedia Title: Jeremy Horn
<hidden for brevity>

Q: who is older Jeremy Horn or Renato Sobral ?

A: Jeremy Horn was born on August 25, 1975. Renato Sobral was born on September 7, 1975. Thus, Jeremy Horn is older. So the answer is: Jeremy Horn.

Wikipedia Title: Lake Wales Medical Center
<hidden for brevity>

Wikipedia Title: Lake Wales, Florida
<hidden for brevity>

Wikipedia Title: Tufts University School of Medicine
<hidden for brevity>

Q: What was the 2014 population of the city where Lake Wales Medical Center is located?
A: Lake Wales Medical Center is located in the city of Polk County, Florida. The population of Polk County in 2014 was 15,140. So the answer is: 15,140.

Wikipedia Title: Nobody Loves You (When You're Down and Out)
<hidden for brevity>

Wikipedia Title: Walls and Bridges
<hidden for brevity>

Wikipedia Title: Mother (John Lennon song)
<hidden for brevity>

Q: Nobody Loves You was written by John Lennon and released on what album that was issued by Apple Records, and was written, recorded, and released during his 18 month separation from Yoko Ono?
A: The album issued by Apple Records, and written, recorded, and released during John Lennon's 18 month separation from Yoko Ono is Walls and Bridges. Nobody Loves You was written by John Lennon on Walls and Bridges album. So the answer is: Walls and Bridges.

2wikimultihopqa: decomp

QC: Do director of film Coolie No. 1 (1995 Film) and director of film The Sensational Trial have the same nationality?
QS: (select) [retrieve_odqa] Who is the director of the film Coolie No. 1 (1995 film)?
A: {"titles": ["Hanro Smitsman", "Coolie No. 1 (1995 film)"], "answer": ["David Dhawan"]}
QS: (select) [retrieve_odqa] Who is the director of the film The Sensational Trial?
A: {"titles": ["Rachel Feldman", "The Sensational Trial"], "answer": ["Karl Freund"]}
QS: (select) [retrieve_odqa] What is David Dhawan's nationality?
A: {"titles": ["David Dhawan", "Brian Johnson (special effects artist)"], "answer": ["India"]}
QS: (select) [retrieve_odqa] What is Karl Freund's nationality?
A: {"titles": ["Karl Freund", "Ian Barry (director)"], "answer": ["Germany"]}
QS: (select) [multihop_titleqa] Titles: ["Hanro Smitsman", "Coolie No. 1 (1995 film)", "Rachel Feldman", "The Sensational Trial", "David Dhawan", "Brian Johnson (special effects artist)", "Karl Freund", "Ian Barry (director)"]. Question: Do director of film Coolie No. 1 (1995 Film) and director of film The Sensational Trial have the same nationality?
A: ["no"]
QS: [EOQ]

QC: Which film has the director died later, The Gal Who Took the West or Twenty Plus Two?
QS: (select) [retrieve_odqa] Who is the director of the film Twenty Plus?
A: {"titles": ["Riki Gal", "Twenty Plus Two"], "answer": ["Joseph M. Newman"]}
QS: (select) [retrieve_odqa] Who is the director of the film The Gal Who Took the West?
A: {"titles": ["Querelle", "The Gal Who Took the West"], "answer": ["Frederick de Cordova"]}
QS: (select) [retrieve_odqa] When did Joseph M. Newman die?

A: {"titles": ["Joseph M. Newman", "Thulasi (1987 film)"], "answer": ["January 23, 2006"]}
QS: (select) [retrieve_odqa] When did Fred de Cordova die?
A: {"titles": ["Fred de Cordova", "Le Masque de la Meduse"], "answer": ["September 15, 2001"]}
QS: (select) [multihop_titleqa] Titles: ["Riki Gal", "Twenty Plus Two", "Querelle", "The Gal Who Took the West", "Joseph M. Newman", "Thulasi (1987 film)", "Fred de Cordova", "Le Masque de la Meduse"]. Question: Which film has the director died later, The Gal Who Took the West or Twenty Plus Two?
A: ["Twenty Plus Two"]
QS: [EOQ]

QC: Who is the grandchild of Krishna Shah (Nepalese Royal)?
QS: (select) [retrieve_odqa] Who is the child of Krishna Shah?
A: {"titles": ["Ana Gruzinsky−Golitsyn", "Krishna Shah (Nepalese royal)", "Diana Weston", "Albina du Boisrouvray"], "answer": ["Rudra Shah"]}
QS: (select) [retrieve_odqa] Who is the child of Rudra Shah?
A: {"titles": ["Jim Ramel Kjellgren", "Constance Anne Herschel", "Rudra Shah"], "answer": ["Prithvipati Shah"]}
QS: (select) [multihop_titleqa] Titles: ["Ana Gruzinsky−Golitsyn", "Krishna Shah (Nepalese royal)", "Diana Weston", "Albina du Boisrouvray", "Jim Ramel Kjellgren", "Constance Anne Herschel", "Rudra Shah"]. Question: Who is the grandchild of Krishna Shah (Nepalese Royal)?
A: ["Prithvipati Shah"]
QS: [EOQ]

QC: Where did the director of film Maddalena (1954 Film) die?
QS: (select) [retrieve_odqa] Who is the director of the film Maddalena?
A: {"titles": ["Ian Barry (director)", "Maddalena (1954 film)", "Brian Kennedy (gallery director)"], "answer": ["Augusto Genina"]}
QS: (select) [retrieve_odqa] Where did Augusto Genina die?
A: {"titles": ["Dana Blankstein", "Augusto Genina", "Peter Levin", "Olav Aaraas"], "answer": ["Rome"]}
QS: (select) [multihop_titleqa] Titles: ["Ian Barry (director)", "Maddalena (1954 film)", "Brian Kennedy (gallery director)", "Dana Blankstein", "Augusto Genina", "Peter Levin", "Olav Aaraas"]. Question: Where did the director of film Maddalena (1954 Film) die?
A: ["Rome"]
QS: [EOQ]

QC: Who was born first out of Martin Hodge and Ivania Martinich?
QS: (select) [retrieve_odqa] When was Martin Hodge born?
A: {"titles": ["Greg A. Hill (artist)", "Martin Hodge", "John Allen (Oxford University cricketer)"], "answer": ["4 February 1959"]}
QS: (select) [retrieve_odqa] When was Ivania Martinich born?
A: {"titles": ["Tom Dickinson", "Ivania Martinich", "Wesley Barresi", "Henry Moore (cricketer)"], "answer": ["25 July 1995"]}
QS: (select) [multihop_titleqa] Titles: ["Greg A. Hill (artist)", "Martin Hodge", "John Allen (Oxford University cricketer)", "Tom Dickinson", "Ivania Martinich", "Wesley Barresi", "Henry Moore (cricketer)"]. Question: Who was born first out of Martin Hodge and Ivania Martinich?
A: ["Martin Hodge"]
QS: [EOQ]

2wikimultihopqa: retrieve_odqa

QC: Who is the director of the film Coolie No. 1 (1995 film)?
QS: (select) [retrieve] Who is the director of the film Coolie No. 1 (1995 film)?
A: ["Hanro Smitsman", "Coolie No. 1 (1995 film)"]

QS: (select) [singlehop_titleqa] Titles: ["Hanro Smitsman", "Coolie No. 1 (1995 film)"]. Question: Who is the director of the film Coolie No. 1 (1995 film)?
A: {"titles": ["Hanro Smitsman", "Coolie No. 1 (1995 film)"], "answer": ["David Dhawan"]}
QS: [EOQ]

QC: Who is the director of the film The Sensational Trial?
QS: (select) [retrieve] Who is the director of the film The Sensational Trial?
A: ["Rachel Feldman", "The Sensational Trial"]
QS: (select) [singlehop_titleqa] Titles: ["Rachel Feldman", "The Sensational Trial"]. Question: Who is the director of the film The Sensational Trial?
A: {"titles": ["Rachel Feldman", "The Sensational Trial"], "answer": ["Karl Freund"]}
QS: [EOQ]

QC: What is David Dhawan's nationality?
QS: (select) [retrieve] What is David Dhawan's nationality?
A: ["David Dhawan", "Brian Johnson (special effects artist)"]
QS: (select) [singlehop_titleqa] Titles: ["David Dhawan", "Brian Johnson (special effects artist)"]. Question: What is David Dhawan's nationality?
A: {"titles": ["David Dhawan", "Brian Johnson (special effects artist)"], "answer": ["India"]}
QS: [EOQ]

QC: What is Karl Freund's nationality?
QS: (select) [retrieve] What is Karl Freund's nationality?
A: ["Karl Freund", "Ian Barry (director)"]
QS: (select) [singlehop_titleqa] Titles: ["Karl Freund", "Ian Barry (director)"]. Question: What is Karl Freund's nationality?
A: {"titles": ["Karl Freund", "Ian Barry (director)"], "answer": ["Germany"]}
QS: [EOQ]

QC: Who is the director of the film Twenty Plus?
QS: (select) [retrieve] Who is the director of the film Twenty Plus?
A: ["Riki Gal", "Twenty Plus Two"]
QS: (select) [singlehop_titleqa] Titles: ["Riki Gal", "Twenty Plus Two"]. Question: Who is the director of the film Twenty Plus?
A: {"titles": ["Riki Gal", "Twenty Plus Two"], "answer": ["Joseph M. Newman"]}
QS: [EOQ]

QC: Who is the director of the film The Gal Who Took the West?
QS: (select) [retrieve] Who is the director of the film The Gal Who Took the West?
A: ["Querelle", "The Gal Who Took the West"]
QS: (select) [singlehop_titleqa] Titles: ["Querelle", "The Gal Who Took the West"]. Question: Who is the director of the film The Gal Who Took the West?
A: {"titles": ["Querelle", "The Gal Who Took the West"], "answer": ["Frederick de Cordova"]}
QS: [EOQ]

QC: When did Joseph M. Newman die?
QS: (select) [retrieve] When did Joseph M. Newman die?
A: ["Joseph M. Newman", "Thulasi (1987 film)"]
QS: (select) [singlehop_titleqa] Titles: ["Joseph M. Newman", "Thulasi (1987 film)"]. Question: When did Joseph M. Newman die?
A: {"titles": ["Joseph M. Newman", "Thulasi (1987 film)"], "answer": ["January 23, 2006"]}
QS: [EOQ]

QC: When did Fred de Cordova die?
QS: (select) [retrieve] When did Fred de Cordova die?
A: ["Fred de Cordova", "Le Masque de la Meduse"]
QS: (select) [singlehop_titleqa] Titles: ["Fred de Cordova", "Le Masque de la Meduse"]. Question: When did Fred de Cordova die?
A: {"titles": ["Fred de Cordova", "Le Masque de la Meduse"], "answer": ["September 15, 2001"]}
QS: [EOQ]

QC: Who is the child of Krishna Shah?
QS: (select) [retrieve] Who is the child of Krishna Shah?
A: ["Ana Gruzinsky−Golitsyn", "Krishna Shah (Nepalese royal)", "Diana Weston", "Albina du Boisrouvray"]
QS: (select) [singlehop_titleqa] Titles: ["Ana Gruzinsky−Golitsyn", "Krishna Shah (Nepalese royal)", "Diana Weston", "Albina du Boisrouvray"]. Question: Who is the child of Krishna Shah?
A: {"titles": ["Ana Gruzinsky−Golitsyn", "Krishna Shah (Nepalese royal)", "Diana Weston", "Albina du Boisrouvray"], "answer": ["Rudra Shah"]}
QS: [EOQ]

QC: Who is the child of Rudra Shah?
QS: (select) [retrieve] Who is the child of Rudra Shah?
A: ["Jim Ramel Kjellgren", "Constance Anne Herschel", "Rudra Shah"]
QS: (select) [singlehop_titleqa] Titles: ["Jim Ramel Kjellgren", "Constance Anne Herschel", "Rudra Shah"]. Question: Who is the child of Rudra Shah?
A: {"titles": ["Jim Ramel Kjellgren", "Constance Anne Herschel", "Rudra Shah"], "answer": ["Prithvipati Shah"]}
QS: [EOQ]

QC: Who is the director of the film Maddalena?
QS: (select) [retrieve] Who is the director of the film Maddalena?
A: ["Ian Barry (director)", "Maddalena (1954 film)", "Brian Kennedy (gallery director)"]
QS: (select) [singlehop_titleqa] Titles: ["Ian Barry (director)", "Maddalena (1954 film)", "Brian Kennedy (gallery director)"]. Question: Who is the director of the film Maddalena?
A: {"titles": ["Ian Barry (director)", "Maddalena (1954 film)", "Brian Kennedy (gallery director)"], "answer": ["Augusto Genina"]}
QS: [EOQ]

QC: Where did Augusto Genina die?
QS: (select) [retrieve] Where did Augusto Genina die?
A: ["Dana Blankstein", "Augusto Genina", "Peter Levin", "Olav Aaraas"]
QS: (select) [singlehop_titleqa] Titles: ["Dana Blankstein", "Augusto Genina", "Peter Levin", "Olav Aaraas"]. Question: Where did Augusto Genina die?
A: {"titles": ["Dana Blankstein", "Augusto Genina", "Peter Levin", "Olav Aaraas"], "answer": ["Rome"]}
QS: [EOQ]

QC: When was Martin Hodge born?
QS: (select) [retrieve] When was Martin Hodge born?
A: ["Greg A. Hill (artist)", "Martin Hodge", "John Allen (Oxford University cricketer)"]
QS: (select) [singlehop_titleqa] Titles: ["Greg A. Hill (artist)", "Martin Hodge", "John Allen (Oxford University cricketer)"]. Question: When was Martin Hodge born?
A: {"titles": ["Greg A. Hill (artist)", "Martin Hodge", "John Allen (Oxford University cricketer)"], "answer": ["4 February 1959"]}
QS: [EOQ]

QC: When was Ivania Martinich born?
QS: (select) [retrieve] When was Ivania Martinich born?
A: ["Tom Dickinson", "Ivania Martinich", "Wesley Barresi", "Henry Moore (cricketer)"]
QS: (select) [singlehop_titleqa] Titles: ["Tom Dickinson", "Ivania Martinich", "Wesley Barresi", "Henry Moore (cricketer)"]. Question: When was Ivania Martinich born?
A: {"titles": ["Tom Dickinson", "Ivania Martinich", "Wesley Barresi", "Henry Moore (cricketer)"], "answer": ["25 July 1995"]}
QS: [EOQ]

`2wikimultihopqa:  singlehop_titleqa`

Wikipedia Title: David Dhawan
<hidden for brevity>

Wikipedia Title: Howard W. Koch
<hidden for brevity>

Wikipedia Title: The Sensational Trial
<hidden for brevity>

Wikipedia Title: Coolie No. 1 (1995 film)
<hidden for brevity>

Wikipedia Title: Karl Freund
<hidden for brevity>

Q: Who is the director of the film Coolie No. 1 (1995 film)?
A: ["David Dhawan"]

Q: Who is the director of the film The Sensational Trial?
A: ["Karl Freund"]

Q: What is David Dhawan's nationality?
A: ["India"]

Q: What is Karl Freund's nationality?
A: ["Germany"]

Wikipedia Title: Fred de Cordova
<hidden for brevity>

Wikipedia Title: Thulasi (1987 film)
<hidden for brevity>

Wikipedia Title: Joseph M. Newman
<hidden for brevity>

Wikipedia Title: The Gal Who Took the West
<hidden for brevity>

Wikipedia Title: Twenty Plus Two
<hidden for brevity>

Q: Who is the director of the film Twenty Plus?
A: ["Joseph M. Newman"]

Q: Who is the director of the film The Gal Who Took the West?

A: ["Frederick de Cordova"]

Q: When did Joseph M. Newman die?
A: ["January 23, 2006"]

Q: When did Fred de Cordova die?
A: ["September 15, 2001"]

Wikipedia Title: Ana Gruzinsky−Golitsyn
<hidden for brevity>

Wikipedia Title: Rudra Shah
<hidden for brevity>

Wikipedia Title: Krishna Shah (Nepalese royal)
<hidden for brevity>

Q: Who is the child of Krishna Shah?
A: ["Rudra Shah"]

Q: Who is the child of Rudra Shah?
A: ["Prithvipati Shah"]

Wikipedia Title: Augusto Genina
<hidden for brevity>

Wikipedia Title: Ian Barry (director)
<hidden for brevity>

Wikipedia Title: Maddalena (1954 film)
<hidden for brevity>

Q: Who is the director of the film Maddalena?
A: ["Augusto Genina"]

Q: Where did Augusto Genina die?
A: ["Rome"]

Wikipedia Title: Martin Hodge
<hidden for brevity>

Wikipedia Title: Ivania Martinich
<hidden for brevity>

Wikipedia Title: Tom Dickinson
<hidden for brevity>

Q: When was Martin Hodge born?
A: ["4 February 1959"]

Q: When was Ivania Martinich born?
A: ["25 July 1995"]

```
              2wikimultihopqa:  multihop_titleqa (direct)
```
Wikipedia Title: David Dhawan
<hidden for brevity>

Wikipedia Title: Howard W. Koch
<hidden for brevity>

Wikipedia Title: The Sensational Trial
<hidden for brevity>

Wikipedia Title: Coolie No. 1 (1995 film)
<hidden for brevity>

Wikipedia Title: Karl Freund
<hidden for brevity>

Q: Do director of film Coolie No. 1 (1995 Film) and director of film The Sensational Trial have the
    same nationality?
A: ["no"]

Wikipedia Title: Fred de Cordova
<hidden for brevity>

Wikipedia Title: Thulasi (1987 film)
<hidden for brevity>

Wikipedia Title: Joseph M. Newman
<hidden for brevity>

Wikipedia Title: The Gal Who Took the West
<hidden for brevity>

Wikipedia Title: Twenty Plus Two
<hidden for brevity>

Q: Which film has the director died later, The Gal Who Took the West or Twenty Plus Two?
A: ["Twenty Plus Two"]

Wikipedia Title: Ana Gruzinsky−Golitsyn
<hidden for brevity>

Wikipedia Title: Rudra Shah
<hidden for brevity>

Wikipedia Title: Krishna Shah (Nepalese royal)
<hidden for brevity>

Q: Who is the grandchild of Krishna Shah (Nepalese Royal)?
A: ["Prithvipati Shah"]

Wikipedia Title: Augusto Genina
<hidden for brevity>

Wikipedia Title: Ian Barry (director)
<hidden for brevity>

Wikipedia Title: Maddalena (1954 film)
<hidden for brevity>

Q: Where did the director of film Maddalena (1954 Film) die?
A: ["Rome"]

Wikipedia Title: Martin Hodge
<hidden for brevity>

Wikipedia Title: Ivania Martinich
<hidden for brevity>

Wikipedia Title: Tom Dickinson
<hidden for brevity>

Q: Who was born first out of Martin Hodge and Ivania Martinich?
A: ["Martin Hodge"]

```
                    2wikimultihopqa:  multihop_titleqa (cot)
```
Wikipedia Title: David Dhawan
<hidden for brevity>

Wikipedia Title: Howard W. Koch
<hidden for brevity>

Wikipedia Title: The Sensational Trial
<hidden for brevity>

Wikipedia Title: Coolie No. 1 (1995 film)
<hidden for brevity>

Wikipedia Title: Karl Freund
<hidden for brevity>

Q: Do director of film Coolie No. 1 (1995 Film) and director of film The Sensational Trial have the
    same nationality?
A: Coolie No. 1 (1995 film) was directed by David Dhawan. The Sensational Trial was directed by
    Karl Freund. David Dhawan's nationality is India. Karl Freund's nationality is Germany.
    Thus, they do not have the same nationality. So the answer is: no.

Wikipedia Title: Fred de Cordova
<hidden for brevity>

Wikipedia Title: Thulasi (1987 film)
<hidden for brevity>

Wikipedia Title: Joseph M. Newman
<hidden for brevity>

Wikipedia Title: The Gal Who Took the West
<hidden for brevity>

Wikipedia Title: Twenty Plus Two
<hidden for brevity>

Q: Which film has the director died later, The Gal Who Took the West or Twenty Plus Two?
A: The film Twenty Plus Two was directed by Joseph M. Newman. The Gal Who Took the West
    was directed by Frederick de Cordova. Joseph M. Newman died on January 23, 2006. Fred de
    Cordova died on September 15, 2001. Thus, the person to die later from the two is Twenty
    Plus Two. So the answer is: Twenty Plus Two.

Wikipedia Title: Ana Gruzinsky–Golitsyn
<hidden for brevity>

Wikipedia Title: Rudra Shah
<hidden for brevity>

Wikipedia Title: Krishna Shah (Nepalese royal)
<hidden for brevity>

Q: Who is the grandchild of Krishna Shah (Nepalese Royal)?
A: Krishna Shah has a child named Rudra Shah. Rudra Shah has a child named Prithvipati Shah. Thus, Krishna Shah has a grandchild named Prithvipati Shah. So the answer is: Prithvipati Shah.

Wikipedia Title: Augusto Genina
<hidden for brevity>

Wikipedia Title: Ian Barry (director)
<hidden for brevity>

Wikipedia Title: Maddalena (1954 film)
<hidden for brevity>

Q: Where did the director of film Maddalena (1954 Film) die?
A: The film Maddalena is directed by Augusto Genina. Augusto Genina died in Rome. So the answer is: Rome.

Wikipedia Title: Martin Hodge
<hidden for brevity>

Wikipedia Title: Ivania Martinich
<hidden for brevity>

Wikipedia Title: Tom Dickinson
<hidden for brevity>

Q: Who was born first out of Martin Hodge and Ivania Martinich?
A: Martin Hodge was born on 4 February 1959. Ivania Martinich was born on 25 July 1995. Thus, Martin Hodge was born first. So the answer is: Martin Hodge.

`musique_ans: decomp`

QC: When did the first large winter carnival take place in the city where CIMI–FM is licensed to broadcast?
QS: (select) [retrieve_odqa] What city is CIMI–FM is licensed to broadcast in?
A: {"titles": ["KWMZ–FM", "CIMI–FM", "KSAO (FM)", "WTTL–FM", "KOLU", "WORW", "WLRX (FM)"], "answer": ["Quebec City"]}
QS: (select) [retrieve_odqa] When did the first large winter carnival in Quebec City take place?
A: {"titles": ["WRQY", "Quebec Winter Carnival", "KBCR–FM"], "answer": ["1894"]}
QS: (select) [multihop_titleqa] Titles: ["KWMZ–FM", "CIMI–FM", "KSAO (FM)", "WTTL–FM", "KOLU", "WORW", "WLRX (FM)", "WRQY", "Quebec Winter Carnival", "KBCR–FM"]. Question: When did the first large winter carnival take place in the city where CIMI–FM is licensed to broadcast?
A: ["1894"]
QS: [EOQ]

QC: What is the headquarters for the organization who sets the standards for ISO 21500?
QS: (select) [retrieve_odqa] Who sets the standards for ISO 21500?
A: {"titles": ["ISO 3166−2:BM", "ISO 3166−2:IS", "ISO 3166−2:IT", "ISO 3166−2:BB", "ISO 21500", "ISO 3166−2:ET", "ISO 3166−2:AT", "ISO 3166−2:CG", "ISO 22000"], "answer": ["International Organization for Standardization"]}
QS: (select) [retrieve_odqa] Where is the headquarters for International Organization for Standardization?
A: {"titles": ["International Organization for Standardization", "ISO 4031", "ISO 3166−2:CN", "Unicode", "ISO 3166−2:GH", "ISO 3166−2:AO", "ISO/TC 68", "ISO 7001", "ISO 3307"], "answer": ["Geneva"]}
QS: (select) [multihop_titleqa] Titles: ["ISO 3166−2:BM", "ISO 3166−2:IS", "ISO 3166−2:IT", "ISO 3166−2:BB", "ISO 21500", "ISO 3166−2:ET", "ISO 3166−2:AT", "ISO 3166−2:CG", "ISO 22000", "International Organization for Standardization", "ISO 4031", "ISO 3166−2:CN", "Unicode", "ISO 3166−2:GH", "ISO 3166−2:AO", "ISO/TC 68", "ISO 7001", "ISO 3307"]. Question: What is the headquarters for the organization who sets the standards for ISO 21500?
A: ["Geneva"]
QS: [EOQ]

QC: How long is the US border with the country that borders the state where Finding Dory takes place?
QS: (select) [retrieve_odqa] In which state does Finding Dory take place?
A: {"titles": ["Finding Dory", "Latvia", "List of countries that border only one other country", "Removal of Hungary's border fence with Austria"], "answer": ["California"]}
QS: (select) [retrieve_odqa] Which country shares a border with California?
A: {"titles": ["Mexico−United States border", "Kingdom of Gera", "Pesticide", "Currie Cup"], "answer": ["Mexico"]}
QS: (select) [retrieve_odqa] What is the length of the US border with Mexico?
A: {"titles": ["Piscataqua River border dispute", "Share a Coke", "Mexico−United States border", "Hotel Arbez"], "answer": ["1,989 mi"]}
QS: (select) [multihop_titleqa] Titles: ["Finding Dory", "Latvia", "List of countries that border only one other country", "Removal of Hungary's border fence with Austria", "Mexico−United States border", "Kingdom of Gera", "Pesticide", "Currie Cup", "Piscataqua River border dispute", "Share a Coke", "Mexico−United States border", "Hotel Arbez"]. Question: How long is the US border with the country that borders the state where Finding Dory takes place?
A: ["1,989 mi"]
QS: [EOQ]

QC: When was Neville A. Stanton's employer founded?
QS: (select) [retrieve_odqa] Who is the employer of Neville A. Stanton?
A: {"titles": ["Harriot Stanton Blatch", "Robichaud v Canada (Treasury Board)", "The Peggy Neville Show", "Stanton Township, Champaign County, Illinois", "Women's suffrage in the United States", "Neville A. Stanton"], "answer": ["University of Southampton"]}
QS: (select) [retrieve_odqa] When was the University of Southampton founded?
A: {"titles": ["Southampton", "Presley Neville"], "answer": ["1862"]}
QS: (select) [multihop_titleqa] Titles: ["Harriot Stanton Blatch", "Robichaud v Canada (Treasury Board)", "The Peggy Neville Show", "Stanton Township, Champaign County, Illinois", "Women's suffrage in the United States", "Neville A. Stanton", "Southampton", "Presley Neville"]. Question: When was Neville A. Stanton's employer founded?
A: ["1862"]
QS: [EOQ]

QC: In which county was the birthplace of the Smoke in tha City performer?
QS: (select) [retrieve_odqa] Who is the performer of Smoke in the City?

A: {"titles": ["Nicotine marketing", "Khabarovsky District", "Beyonce", "Asthma", "Smoke in tha City"], "answer": ["MC Eiht"]}
QS: (select) [retrieve_odqa] What is the birthplace of MC Eiht?
A: {"titles": ["Cherokee City, Arkansas", "MC Eiht", "Biysky District"], "answer": ["Compton"]}
QS: (select) [retrieve_odqa] Compton is located in which county?
A: {"titles": ["Vilnius County", "Biblioteca Ayacucho", "Compton, California"], "answer": ["Los Angeles County"]}
QS: (select) [multihop_titleqa] Titles: ["Nicotine marketing", "Khabarovsky District", "Beyonce", "Asthma", "Smoke in tha City", "Cherokee City, Arkansas", "MC Eiht", "Biysky District", "Vilnius County", "Biblioteca Ayacucho", "Compton, California"]. Question: In which county was the birthplace of the Smoke in tha City performer?
A: ["Los Angeles County"]
QS: [EOQ]

                    musique_ans:  retrieve_odqa

QC: What city is CIMI–FM is licensed to broadcast in?
QS: (select) [retrieve] What city is CIMI–FM is licensed to broadcast in?
A: ["KWMZ–FM", "CIMI–FM", "KSAO (FM)", "WTTL–FM", "KOLU", "WORW", "WLRX (FM)"]
QS: (select) [singlehop_titleqa] Titles: ["KWMZ–FM", "CIMI–FM", "KSAO (FM)", "WTTL–FM", "KOLU", "WORW", "WLRX (FM)"]. Question: What city is CIMI–FM is licensed to broadcast in?
A: {"titles": ["KWMZ–FM", "CIMI–FM", "KSAO (FM)", "WTTL–FM", "KOLU", "WORW", "WLRX (FM)"], "answer": ["Quebec City"]}
QS: [EOQ]

QC: When did the first large winter carnival in Quebec City take place?
QS: (select) [retrieve] When did the first large winter carnival in Quebec City take place?
A: ["WRQY", "Quebec Winter Carnival", "KBCR–FM"]
QS: (select) [singlehop_titleqa] Titles: ["WRQY", "Quebec Winter Carnival", "KBCR–FM"]. Question: When did the first large winter carnival in Quebec City take place?
A: {"titles": ["WRQY", "Quebec Winter Carnival", "KBCR–FM"], "answer": ["1894"]}
QS: [EOQ]

QC: Who sets the standards for ISO 21500?
QS: (select) [retrieve] Who sets the standards for ISO 21500?
A: ["ISO 3166–2:BM", "ISO 3166–2:IS", "ISO 3166–2:IT", "ISO 3166–2:BB", "ISO 21500", "ISO 3166–2:ET", "ISO 3166–2:AT", "ISO 3166–2:CG", "ISO 22000"]
QS: (select) [singlehop_titleqa] Titles: ["ISO 3166–2:BM", "ISO 3166–2:IS", "ISO 3166–2:IT", "ISO 3166–2:BB", "ISO 21500", "ISO 3166–2:ET", "ISO 3166–2:AT", "ISO 3166–2:CG", "ISO 22000"]. Question: Who sets the standards for ISO 21500?
A: {"titles": ["ISO 3166–2:BM", "ISO 3166–2:IS", "ISO 3166–2:IT", "ISO 3166–2:BB", "ISO 21500", "ISO 3166–2:ET", "ISO 3166–2:AT", "ISO 3166–2:CG", "ISO 22000"], "answer": ["International Organization for Standardization"]}
QS: [EOQ]

QC: Where is the headquarters for International Organization for Standardization?
QS: (select) [retrieve] Where is the headquarters for International Organization for Standardization?
A: ["International Organization for Standardization", "ISO 4031", "ISO 3166–2:CN", "Unicode", "ISO 3166–2:GH", "ISO 3166–2:AO", "ISO/TC 68", "ISO 7001", "ISO 3307"]
QS: (select) [singlehop_titleqa] Titles: ["International Organization for Standardization", "ISO 4031", "ISO 3166–2:CN", "Unicode", "ISO 3166–2:GH", "ISO 3166–2:AO", "ISO/TC 68", "ISO 7001", "ISO 3307"]. Question: Where is the headquarters for International Organization for Standardization?

A: {"titles": ["International Organization for Standardization", "ISO 4031", "ISO 3166−2:CN", "Unicode", "ISO 3166−2:GH", "ISO 3166−2:AO", "ISO/TC 68", "ISO 7001", "ISO 3307"], "answer": ["Geneva"]}
QS: [EOQ]

QC: In which state does Finding Dory take place?
QS: (select) [retrieve] In which state does Finding Dory take place?
A: ["Finding Dory", "Latvia", "List of countries that border only one other country", "Removal of Hungary's border fence with Austria"]
QS: (select) [singlehop_titleqa] Titles: ["Finding Dory", "Latvia", "List of countries that border only one other country", "Removal of Hungary's border fence with Austria"]. Question: In which state does Finding Dory take place?
A: {"titles": ["Finding Dory", "Latvia", "List of countries that border only one other country", "Removal of Hungary's border fence with Austria"], "answer": ["California"]}
QS: [EOQ]

QC: Which country shares a border with California?
QS: (select) [retrieve] Which country shares a border with California?
A: ["Mexico−United States border", "Kingdom of Gera", "Pesticide", "Currie Cup"]
QS: (select) [singlehop_titleqa] Titles: ["Mexico−United States border", "Kingdom of Gera", "Pesticide", "Currie Cup"]. Question: Which country shares a border with California?
A: {"titles": ["Mexico−United States border", "Kingdom of Gera", "Pesticide", "Currie Cup"], "answer": ["Mexico"]}
QS: [EOQ]

QC: What is the length of the US border with Mexico?
QS: (select) [retrieve] What is the length of the US border with Mexico?
A: ["Piscataqua River border dispute", "Share a Coke", "Mexico−United States border", "Hotel Arbez"]
QS: (select) [singlehop_titleqa] Titles: ["Piscataqua River border dispute", "Share a Coke", "Mexico−United States border", "Hotel Arbez"]. Question: What is the length of the US border with Mexico?
A: {"titles": ["Piscataqua River border dispute", "Share a Coke", "Mexico−United States border", "Hotel Arbez"], "answer": ["1,989 mi"]}
QS: [EOQ]

QC: Who is the employer of Neville A. Stanton?
QS: (select) [retrieve] Who is the employer of Neville A. Stanton?
A: ["Harriot Stanton Blatch", "Robichaud v Canada (Treasury Board)", "The Peggy Neville Show", "Stanton Township, Champaign County, Illinois", "Women's suffrage in the United States", "Neville A. Stanton"]
QS: (select) [singlehop_titleqa] Titles: ["Harriot Stanton Blatch", "Robichaud v Canada (Treasury Board)", "The Peggy Neville Show", "Stanton Township, Champaign County, Illinois", "Women's suffrage in the United States", "Neville A. Stanton"]. Question: Who is the employer of Neville A. Stanton?
A: {"titles": ["Harriot Stanton Blatch", "Robichaud v Canada (Treasury Board)", "The Peggy Neville Show", "Stanton Township, Champaign County, Illinois", "Women's suffrage in the United States", "Neville A. Stanton"], "answer": ["University of Southampton"]}
QS: [EOQ]

QC: When was the University of Southampton founded?
QS: (select) [retrieve] When was the University of Southampton founded?
A: ["Southampton", "Presley Neville"]

QS: (select) [singlehop_titleqa] Titles: ["Southampton", "Presley Neville"]. Question: When was the University of Southampton founded?
A: {"titles": ["Southampton", "Presley Neville"], "answer": ["1862"]}
QS: [EOQ]

QC: Who is the performer of Smoke in the City?
QS: (select) [retrieve] Who is the performer of Smoke in the City?
A: ["Nicotine marketing", "Khabarovsky District", "Beyonce", "Asthma", "Smoke in tha City"]
QS: (select) [singlehop_titleqa] Titles: ["Nicotine marketing", "Khabarovsky District", "Beyonce", "Asthma", "Smoke in tha City"]. Question: Who is the performer of Smoke in the City?
A: {"titles": ["Nicotine marketing", "Khabarovsky District", "Beyonce", "Asthma", "Smoke in tha City"], "answer": ["MC Eiht"]}
QS: [EOQ]

QC: What is the birthplace of MC Eiht?
QS: (select) [retrieve] What is the birthplace of MC Eiht?
A: ["Cherokee City, Arkansas", "MC Eiht", "Biysky District"]
QS: (select) [singlehop_titleqa] Titles: ["Cherokee City, Arkansas", "MC Eiht", "Biysky District"]. Question: What is the birthplace of MC Eiht?
A: {"titles": ["Cherokee City, Arkansas", "MC Eiht", "Biysky District"], "answer": ["Compton"]}
QS: [EOQ]

QC: Compton is located in which county?
QS: (select) [retrieve] Compton is located in which county?
A: ["Vilnius County", "Biblioteca Ayacucho", "Compton, California"]
QS: (select) [singlehop_titleqa] Titles: ["Vilnius County", "Biblioteca Ayacucho", "Compton, California"]. Question: Compton is located in which county?
A: {"titles": ["Vilnius County", "Biblioteca Ayacucho", "Compton, California"], "answer": ["Los Angeles County"]}
QS: [EOQ]

```
musique_ans:  singlehop_titleqa
```

Wikipedia Title: CIMI−FM
<hidden for brevity>

Wikipedia Title: Quebec Winter Carnival
<hidden for brevity>

Wikipedia Title: KELD−FM
<hidden for brevity>

Q: What city is CIMI−FM is licensed to broadcast in?
A: ["Quebec City"]

Q: When did the first large winter carnival in Quebec City take place?
A: ["1894"]

Wikipedia Title: ISO/TC 68
<hidden for brevity>

Wikipedia Title: ISO 21500
<hidden for brevity>

Wikipedia Title: ISO 3166−2:GH
<hidden for brevity>

Q: Who sets the standards for ISO 21500?
A: ["International Organization for Standardization"]

Q: Where is the headquarters for International Organization for Standardization?
A: ["Geneva"]

Wikipedia Title: Mexico–United States border
<hidden for brevity>

Wikipedia Title: Mexico–United States border
<hidden for brevity>

Wikipedia Title: Finding Dory
<hidden for brevity>

Wikipedia Title: India
<hidden for brevity>

Q: In which state does Finding Dory take place?
A: ["California"]

Q: Which country shares a border with California?
A: ["Mexico"]

Q: What is the length of the US border with Mexico?
A: ["1,989 mi"]

Wikipedia Title: Southampton
<hidden for brevity>

Wikipedia Title: Jonathan Stanton
<hidden for brevity>

Wikipedia Title: Neville A. Stanton
<hidden for brevity>

Q: Who is the employer of Neville A. Stanton?
A: ["University of Southampton"]

Q: When was the University of Southampton founded?
A: ["1862"]

Wikipedia Title: Compton, California
<hidden for brevity>

Wikipedia Title: Smoke in tha City
<hidden for brevity>

Wikipedia Title: MC Eiht
<hidden for brevity>

Wikipedia Title: Eastern Bengal and Assam
<hidden for brevity>

Q: Who is the performer of Smoke in the City?

A: ["MC Eiht"]

Q: What is the birthplace of MC Eiht?
A: ["Compton"]

Q: Compton is located in which county?
A: ["Los Angeles County"]

```
musique_ans:  multihop_titleqa (direct)
```

Wikipedia Title: CIMI−FM
<hidden for brevity>

Wikipedia Title: Quebec Winter Carnival
<hidden for brevity>

Wikipedia Title: KELD−FM
<hidden for brevity>

Q: When did the first large winter carnival take place in the city where CIMI−FM is licensed to
    broadcast?
A: ["1894"]

Wikipedia Title: ISO/TC 68
<hidden for brevity>

Wikipedia Title: ISO 21500
<hidden for brevity>

Wikipedia Title: ISO 3166−2:GH
<hidden for brevity>

Q: What is the headquarters for the organization who sets the standards for ISO 21500?
A: ["Geneva"]

Wikipedia Title: Mexico−United States border
<hidden for brevity>

Wikipedia Title: Mexico−United States border
<hidden for brevity>

Wikipedia Title: Finding Dory
<hidden for brevity>

Wikipedia Title: India
<hidden for brevity>

Q: How long is the US border with the country that borders the state where Finding Dory takes
    place?
A: ["1,989 mi"]

Wikipedia Title: Southampton
<hidden for brevity>

Wikipedia Title: Jonathan Stanton
<hidden for brevity>

Wikipedia Title: Neville A. Stanton
<hidden for brevity>

Q: When was Neville A. Stanton's employer founded?
A: ["1862"]

Wikipedia Title: Compton, California
<hidden for brevity>

Wikipedia Title: Smoke in tha City
<hidden for brevity>

Wikipedia Title: MC Eiht
<hidden for brevity>

Wikipedia Title: Eastern Bengal and Assam
<hidden for brevity>

Q: In which county was the birthplace of the Smoke in tha City performer?
A: ["Los Angeles County"]

```
                    musique_ans:  multihop_titleqa (cot)
```
Wikipedia Title: CIMI−FM
<hidden for brevity>

Wikipedia Title: Quebec Winter Carnival
<hidden for brevity>

Wikipedia Title: KELD−FM
<hidden for brevity>

Q: When did the first large winter carnival take place in the city where CIMI−FM is licensed to
    broadcast?
A: CIMI−FM is licensed to broadcast in Quebec City. The first large winter carnival in Quebec
    City took place in 1894. So the answer is: 1894.

Wikipedia Title: ISO/TC 68
<hidden for brevity>

Wikipedia Title: ISO 21500
<hidden for brevity>

Wikipedia Title: ISO 3166−2:GH
<hidden for brevity>

Q: What is the headquarters for the organization who sets the standards for ISO 21500?
A: The standards for ISO 21500 were set by International Organization for Standardization. The
    International Organization for Standardization has headquarters in Geneva. So the answer is:
    Geneva.

Wikipedia Title: Mexico−United States border
<hidden for brevity>

Wikipedia Title: Mexico−United States border
<hidden for brevity>

Wikipedia Title: Finding Dory
<hidden for brevity>

Wikipedia Title: India
<hidden for brevity>

Q: How long is the US border with the country that borders the state where Finding Dory takes place?
A: Finding Dory is supposed to take place in California. The country that shares a border with California is Mexico. The length of the us border with Mexico is 1,989 mi. So the answer is: 1,989 mi.

Wikipedia Title: Southampton
<hidden for brevity>

Wikipedia Title: Jonathan Stanton
<hidden for brevity>

Wikipedia Title: Neville A. Stanton
<hidden for brevity>

Q: When was Neville A. Stanton's employer founded?
A: The employer of Neville A. Stanton is University of Southampton. The University of Southampton was founded in 1862. So the answer is: 1862.

Wikipedia Title: Compton, California
<hidden for brevity>

Wikipedia Title: Smoke in tha City
<hidden for brevity>

Wikipedia Title: MC Eiht
<hidden for brevity>

Wikipedia Title: Eastern Bengal and Assam
<hidden for brevity>

Q: In which county was the birthplace of the Smoke in tha City performer?
A: The performer of Smoke in tha City is MC Eiht. MC Eiht's birthplace is Compton. Compton is located in the county of Los Angeles County. So the answer is: Los Angeles County.

