# OpenReview forum: "Decomposed Prompting: A Modular Approach for Solving Complex Tasks"
_ICLR.cc/2023/Conference — ICLR 2023 poster_

### Official Review · Reviewer_AJBU · 2022-10-24

**Confidence:** 3
**Correctness:** 3
**Technical Novelty And Significance:** 2
**Empirical Novelty And Significance:** 2
**Recommendation:** 6

**Clarity, Quality, Novelty And Reproducibility:**

- Clarity: the paper is mostly clear and easy to follow.

- Novelty: The idea seems very intuitive and a prompting version of previous ideas such as modular network, so I don't know how to comment on novelty.

- Reproducibility: appendix provides prompts used in experiments.

**Strength And Weaknesses:**

- Strength: the method seems a natural step in the direction of prompting, and empirical results across different tasks are good.

- Weakness: experiments seem to only focus on establishing better results than baselines (e.g. chain-of-thought), but I feel we need better analysis about the method itself. A key thing missing is failure modes and limitations -- I'd argue such insights are more valuable for practitioners interested in adopting Decomposed Prompting.

Also, there is some recent and related work around modular use of LLMs, such as Socratic Models [1]. Although motivation and tasks are not the same, some discussion is needed.

[1] Zeng et al., Socratic Models: Composing Zero-Shot Multimodal Reasoning with Language.


**Summary Of The Paper:**

The paper proposes Decomposed Prompting, a new prompting method to decompose a task into subtasks and handle them by modular prompts. This allows hierarchical and recursive decomposition, as well as calling external APIs. Across symbolic manipulation (k-th letter concatenation, list reversal) and question answering (CommaQA-E, HotpotQA) tasks, Decomposed Prompting is shown to outperform Chain-of-thought prompting.

**Summary Of The Review:**

I believe Decomposed Prompting is a natural next step for prompting, and results here look good. We probably need more analysis and insights so that people can design decomposed prompts for their tasks.

---

> ### Author Response · Authors · 2022-11-19
> **Analysis**
>
> Thank you for your comments and recognizing the potential of Decomp as the next step for prompting.
> > error analysis
>
> Thank you for your suggestion. We have added the error cases to the Appendix. Not surprisingly, the errors are due to mistakes made by the sub-task handlers. Improving these handlers via weak supervision (from the end-task) or developing better task-following models are potential directions of future research. We will expand this analysis to the other datasets for the final paper.
>
> Additionally, as we mention in our 2WikiMultihopQA experiments, producing the decomposition structure can be a challenging task for smaller models e.g. flan-t5. Either simplifying the structure into more natural language or fine-tuning these models on various decomposition structures are also potential avenues for future work. We will present these decomposition failure cases in our final paper.
>
>
> > socratic models
>
> Thank you for this pointer. We will add them to the paper.

---

### Official Review · Reviewer_Pqnp · 2022-10-24

**Confidence:** 4
**Correctness:** 3
**Technical Novelty And Significance:** 3
**Empirical Novelty And Significance:** 3
**Recommendation:** 8

**Clarity, Quality, Novelty And Reproducibility:**

Quality, clarity: Good; see above.

Originality: This is hard to assess given that there are so many preprints on similar topics appearing over the last several months (as the authors surely know, judging by the related work section!). However, I think this particular idea is new and placed well in the context of other works.

Reproducibility: Although code is not provided, I would likely have no trouble reproducing the results on these tasks and extending them to new ones.

**Strength And Weaknesses:**

Strengths:
- (++) I like the approach of separating controller and subtask modules. This work contributes to an increasing realization that linear attention -- where the entire reasoning sequence is a single chain of tokens -- could be insufficient for hard reasoning problems; the chosen comparisons with CoT illustrate this well. The ability, even if only hypothetical, to teach a controller to query external structured retrieval systems, execute programs, etc., is an exciting prospect.
- (+) Interesting combination of few-shot prompting (to teach the decomposer/controller) with question decomposition along the lines of [Perez et al., 2020; Wolfson et al., 2020].

Weaknesses / questions:
- (--) The breadth of applications is a little unsatisfying given how general the proposed approach can be. The examples currently given are either ones that can be turned into exact programs (list and string operations) or ones that use a chain of questions for knowledge retrieval (QA). But where can decomposed prompting be applied beyond these cases? The most interesting tasks may be the ones that require a mix of formal steps  (that a simple program could do) / querying of symbolic systems and knowledge generation / retrieval.
- (-) The following would be important to illustrate:
  - How does direct prompting -- no CoT -- performs on these tasks?
  - How sensitive is the system to the model size and prompt format? Can GPT-2 XL or the smallest variant of GPT-3 be used for (a) the decomposer and (b) the submodules, and if so, which ones? (From my own experience playing with CoT-like approaches, I would expect large differences even among davinci, text-davinci-001, and text-davinci-002 variants of GPT-3 for some problems.)
- When the submodules are expected to produce structured output, such as a list of strings, how reliably do they generate text in the correct format? (What would happen if a submodule outputs something not of the form "[ item1, item2, ... ]", but the following call contains "foreach"?)

**Summary Of The Paper:**

A framework called DecomP for solving multi-step reasoning problems with large language models is proposed. A task is "decomposed" into a sequence of smaller subtasks, where each subtask calls a reusable submodule (itself a LLM call or an external knowledge retrieval); a simple syntax for control flow allowing substitution of past subtask results and iteration over lists is introduced. Experiments on two string and list operation tasks and two question-answering tasks show that DecomP compares favourably with chain-of-thought approaches.

**Summary Of The Review:**

The paper is quite well written and adds some interesting ideas to our understanding of multi-step reasoning with LLMs. On the other hand, given all the recent findings that go in a similar direction, the strong results with this particular method did not surprise me very much. This isn't a reason not to publish the paper, of course, but I would hope to see something that would make it stand out more, such as results on a broader (more diverse) set of tasks that illustrate more variety of subtasks and interesting ways to combine them. ~Therefore, I am giving a cautious "marginally below", but am willing to be convinced to raise the score.~ See post-rebuttal comment below.

---

> ### Author Response · Authors · 2022-11-19
> **Broader Results**
>
> Thank you for your encouraging comments and recognizing the potential of our approach to go beyond a single linear output sequence via decomposition.
>
> > general approach but breadth of applications is a little unsatisfying
>
> We are glad that you recognize the flexibility of the framework. Our goal was to show the different possible structures possible with our framework and their benefits compared to prior work. We believe that similar structures can be applied on more datasets (e.g. claim decomposition) and novel decomposition structures could also be developed (e.g. generate- and-simplify) as future work. We have now added an additional multi-step reasoning dataset:: 2WikiMultihopQA (in App. A) and two math qa datasets: GMS8K and MultiArith (in App. B). Hopefully these seven datasets provide sufficient evidence that DecomP is a useful prompting technique and would be applicable to other complex tasks.
>
>
> > direct QA?
>
> Thanks for pointing this out. We did not expect direct QA to solve these tasks any better than CoT. E.g., DirectQA model gets an accuracy of 0-3 on the letter concatenation task and about 10 pts worse than CoT on CommaQA. We will add them for completeness to the final version. Note that No-Context models in HotpotQa are direct QA models.
>
> > sensitivity to prompt format
>
> Thanks for this interesting suggestion. First, we would like to mention that we have used three different models in our experiments (text-davinci-002 for shorter contexts, text-davinci-001 on the easy reversal task and code-davinci-002 on the longer HotpotQA context). We also used three different prompts to show the insensitivity to prompts. See the updated appendix (App. D) for individual numbers on each prompt -- DecomP always outperforms CoT across prompt formats.
>
> > sensitivity to model size
>
> Given your suggestion, we evaluated GPT3 models (curie, davinci-001 and davinci-002) on the CommAQA dataset. In summary, the weaker models do perform worse as expected but the decomposition approach is generally better than CoT (except curie where all the scores are close to zero) We added these numbers to the appendix (App. C).
>
> We also evaluated the flan-t5 models on the 2WikiMultihopQA dataset. Unfortunately, the flan-t5 models are currently not able to handle the decomposition structure and we are currently restricted to GPT3 models for the decomposition task. Regarding the sub-tasks, most of the sub-tasks were too challenging for flan-t5 (even identifying the letters in a word). The factoid question answering sub-task (singlehop_rcqa in Fig. 6) was more aligned with the tasks that can be handled with flan-t5. So we experimented with different sizes of flan-t5 (large, xl and xxl) on the 2WikiMultihopQA replacing the singlehop qa sub-task model with flan-t5. As we show in Fig. 13, flan-t5-xxl, a 10x smaller model than GPT3, can marginally outperform GPT3 on this sub-task. We hope that future work can even replace the decomposers with flan-t5 models obviating the need of large limited-access models.
>
> > structured output from sub-modules
>
> We are currently restricted to GPT3 for the decomposition task as well as structured outputs, since only these models are reliably able to produce such outputs. We hope that future work can even replace these modules with open-access and free models obviating the need of large limited-access models.
>
>
>  > What would happen if a submodule outputs something not of the form "[ item1, item2, ... ]", but the following call contains "foreach"?)
>
> The structure is important for the foreach operator and GPT3 models have produced this structure successfully in all our datasets. At the same time, our decompositions can be written  without relying on the answer structure. E.g. we added new results on the letter concatenation task where we replaced the foreach loop with
> ```
> QS: [str_position] What is the last letter in "Augusta"?
> A: "a"
> QS: [str_position] What is the last letter in "Ada"?
> A: "a"
> QS: [str_position] What is the last letter in "King"?
> A: "g"
> ```
> I.e. We used the LLM to identify all the sub-questions from the answer. By using the model to generate these sub-questions we can be more robust to formatting issues in the output answers (at the cost of potential precision or recall mistakes in generating these questions).
>
>
> > success is not surprising and would have liked to see broader results
>
> Chain-of-thought has been very successful on many tasks (also shown by Suzgun et al’ 22) achieving close to human performance. So we do believe that finding a novel approach that can outperform CoT is surprising, even though it appears to be not so surprising once the approach is presented. To further make our case, we did expand our datasets to two math QA datasets (GSM8k and MultiArith) as well as 2WikiMultihopQA. Hopefully these seven datasets provide sufficient evidence that DecomP is a broadly useful technique.

---

> > ### Comment · Reviewer_Pqnp · 2022-11-30
> > **Response to response**
> >
> > Thank you for the answers and updates. I have no further questions and have increased the score.

---

### Official Review · Reviewer_WUGg · 2022-10-25

**Confidence:** 4
**Correctness:** 3
**Technical Novelty And Significance:** 3
**Empirical Novelty And Significance:** 2
**Recommendation:** 6

**Clarity, Quality, Novelty And Reproducibility:**

Clarity: The paper is well-written and organized.

Quality: The experiments are solid but the scope is limited. See weaknesses for details.

Novelty: To me, the novelty of this paper is limited. It is a solid extension of the Least-to-most prompting (Zhou et al., 2022).

Reproducibility: With all the details provided, it won't be too hard to reproduce the results.


**Strength And Weaknesses:**

Strengths:
- The proposed approach is well-motivated.
- Strong empirical results across different datasets.
- Comprehensive experiments to show the flexibility of the proposed framework.

Weaknesses:
- To me, the main weakness of this paper is the lack of novelty. It is more of an extension of the least-to-most prompting (Zhou et al., 2022). The least-to-most prompting can also generate the decompositions iteratively based on the outputs from previous steps. What's really new in this paper is that instead of solving the decompositions all at once by a pretrained language model, the authors show that solving them by dedicated LLMs or other systems (e.g., external API calls) is even better.
- The scope of the experiments is not broad enough given the flexibility of the proposed framework. It would be better if the authors could test it in more challenging settings. There are two possible settings I think might be interesting:
    - Answering the "why" questions in long-form questions answering (Fan et al., 2019). For example, to answer "why were the main predators in New Zealand large birds, compared to other places in the world?", we could possibly decompose it into several sub-questions to gather relevant information and synthesize them:
        1.  What are some of the main predators in the world?
        2.  Why New Zealand doesn't have *results from #1* as the main predators?
    - Fact-checking real-world complex claims through decomposition (Chen et al., 2022). For example, to check a claim like "Joe Biden stated on August 31, 2020, in a speech: "When I was vice president, violent crime fell 15% in this country. ... The murder rate now is up 26% across the nation this year under Donald Trump." We may need to know the answers to the following sub-questions and synthesize them to produce final veracity for the claim:
        1. Did the crime rate fall by 15% during Joe Biden's presidency?
        2. Did the murder rate in 2020 increase by 26% from 2019?
        3. Is Biden comparing crime rates from the same time interval in his statement?
        4. Is the violent crime rate and murder rate directly comparable?

    It would be better to show to what extent the decomposed prompts work in such real-world scenarios. Can you generate the decompositions iteratively? It is possible to synthesize the final answer by the decompositions?

References:
 - Angela Fan, Yacine Jernite, Ethan Perez, David Grangier, Jason Weston, and Michael Auli. 2019. ELI5: Long Form Question Answering. In Proceedings of the 57th Annual Meeting of the Association for Computational Linguistics, pages 3558–3567, Florence, Italy. Association for Computational Linguistics.
 - Chen, Jifan, Aniruddh Sriram, Eunsol Choi, and Greg Durrett. "Generating Literal and Implied Subquestions to Fact-check Complex Claims." arXiv preprint arXiv:2205.06938 (2022).


**Summary Of The Paper:**

This paper proposed to use decomposed prompting to solve complex NLP tasks. The idea behind decomposed prompting is very simple: divided and conquer. For a given complex task, they first decompose it into simpler sub-tasks via task prompting and then solve each sub-task by prompting-based LLMs dedicated to the sub-task. They conduct experiments on four different tasks including $k^{th}$ LETTER CONCATENATION, LIST REVERSAL, LONG-CONTEXT QUESTION ANSWERING, and OPEN-DOMAIN QUESTION ANSWERING. The empirical results not only show that the proposed framework improves the few-shot performance over CoT prompting but also demonstrate flexibility: it is easy to incorporate symbolic operations and information retrieval systems into the framework.


**Summary Of The Review:**

Although the idea is not brand new, the experiments are well-designed and demonstrate how the proposed approach could be used in different scenarios. I think the general community would find this useful and many researchers would benefit from the design choices for different tasks. Personally, I don't feel quite excited after reading this paper. It is a solid extension of the previous literature, but I expect more from an ICLR publication. Therefore, I lean toward a weak rejection.

---

> ### Author Response · Authors · 2022-11-19
> **Novelty**
>
> Thank you for your comments and recognizing the comprehensiveness of our experiments and the strength of our empirical results.
>
> >  extension of the least-to-most prompting
>
> As we clarified in the paper (and will expand upon further in our revision), least-to-most is just one particular decomposition structure and has a limited scope. E.g., by definition, least-to-most asks questions from easiest to the hardest and requires an LLM to eventually answer the complete question (“most” in least-to-most). We don’t have any such restriction; in fact, most of our datasets don’t even require an LLM to answer the complete question. E.g, every DecomP sub-question in the letter concatenation task is a simple one and we never ask the LLM to solve the complete question. Additionally, hierarchy, top-down recursion, and other non-linear decomposition structures are not even possible with least-to-most prompting, but very natural in the DecomP framework. We have added a comparison to least-to-most prompting on the letter concatenation task (see Fig. 16) and its performance is similar to chain-of-thought and far behind DecomP.
>
> The only commonality between these two methods is the usage of question decomposition, which is a broad idea used across many prior works (Talmor & Berant, 2018, Min et al. 2019, Perez et al., 2020; Khot et al., 2021, inter alia). We believe that sharing such a fundamental and broad concept with prior works should not make a paper incremental. We hope you agree.
>
> > not broad enough given the flexibility of the proposed framework
>
> We are glad that you recognize the flexibility of the framework. Our goal was to show a flavor of the many possible reasoning structures possible with our framework and their benefits compared to prior work. We have added two arithmetic QA datasets (GSM8k, MultiArith) and a new QA dataset (2WikiMultiHopQA) to further show the broad applicability of our approach. We believe that similar decomposition structures can be applied on more datasets (e.g., claim decomposition) and novel decomposition structures can also be developed in this framework (e.g., generate-and-simplify) as future work.
>
> > real-world scenarios (e.g. ELI5, ClaimDecomp)
>
> Thanks for your suggestions. We have added an additional dataset that requires multi-step reasoning: 2WikiMultihopQA. On this dataset too, we show that our approach leads to improved performance over CoT reasoning. We believe our current experiments with now 7 datasets (including the two new math QA datasets) provide sufficient evidence that DecomP is a useful prompting technique and that it might be useful on even more complex tasks.
>
> We also looked at ClaimDecomp but unfortunately this is currently mainly a decomposition dataset and has no associated corpora for QA. ELI5 could benefit from question decomposition and possible even other interesting decomposition structures (e.g. generate an explanation and then use a LLM to simplify that explanation). These are extremely challenging datasets at the frontier of prompting-based methods and even supervised methods and therefore would require a more thorough investigation.. We hope to explore these datasets for decomposed prompting in future work.

---

> > ### Comment · Reviewer_WUGg · 2022-12-12
> > **Raised my score**
> >
> > Thanks for the clarification and the newly added experiments! The results look stronger and more convincing to me. I am happy to raise my score to 6.

---

### Official Review · Reviewer_H95Z · 2022-10-31

**Confidence:** 4
**Correctness:** 4
**Technical Novelty And Significance:** 4
**Empirical Novelty And Significance:** 3
**Recommendation:** 6

**Clarity, Quality, Novelty And Reproducibility:**

The idea of decomposition is novel.  Reproducing the results should be easy based on the provided prompts.

**Strength And Weaknesses:**

- Following some predecessors, solving a difficult task by devide-and-conquer, decomposing them into simpler one, is quite novel to me.

- The potential of the proposed is on solving challenging tasks, such as arithmetics.  I would encourage the authors try on challenging GSM8K dataset. It's not only because it's a dataset that CoT haven't solved it well, but I believe it could be a great fit to demonstrate the strength of the proposed algorithm. Many complex  arithmetics problems can be decomposed into simple sub-problems. I would expect each subproblem can be solved well by LLM, so as the whole problem under the decomposition scheme.

- I have a few questions.  In practice, how do we decide which decomposition scheme to use? also, how do  we decide the question of each subproblem? is it required human design to tailor for each application like CoT? if so, how sensitive the results is to the design of the questions and decomposition?

- How does this model work under a smaller LLM? In CoT, an interesting finding is it works better with larger LLM. How about the proposed one?

**Summary Of The Paper:**

The authors propose a novel prompting algorithm based on a decomposition or divide-and-conquer idea. The idea is instead of feeding long CoT into the LLM, it decomposes the difficult task into easier subtask. The advantage is the smaller task can either be solved better with LLM or can be handled by external knowledges. Several experiments have been conducted to prove the effectiveness of the proposed method.

**Summary Of The Review:**

The paper provides a novel idea, and several experiments to support its claim. It would be even better to experiment on challenging task, such as GSM8K as suggested.

---

> ### Author Response · Authors · 2022-11-19
> **Math datasets**
>
> Thank you for your supportive comments and recognizing the novelty of our divide-and-conquer approach.
> >  try on challenging GSM8K
>
> Thanks for your suggestion. A key advantage of our approach is that we can identify the sub-task where a system, say CoT,  makes errors and choose to decompose the problem such that a targeted sub-task handler could focus on this source of error. E.g. we can reduce the errors made on detecting the k^th letter by decomposing that particular sub-task. Based on your suggestion, we focused on the mistakes due to relying on patterns to extract answers from chain-of-thought. By reducing these types of errors, our decomposed prompts result in a 17 pt improvement on MultiArith (78 -> 95) and 14 pt improvement on GSM8K(36 -> 50.6) compared to CoT. We have added these results in the paper (see App. B). We will evaluate and report numbers on other math reasoning datasets for the final version.
>
> Specifically, CoT prompts use a final sentence ending in “The answer is …” to extract the predicted answer. E.g.
>
> ```
> Q: There are 15 trees in the grove. Grove workers will plant trees in the grove today. After they are done, there will be 21 trees. How many trees did the grove workers plant today?
> A: There are 15 trees originally. Then there were 21 trees after some more were planted. So there must have been 21 - 15 = 6. The answer is 6.
> ```
> However, GPT3 does not always produce outputs with a clearly marked answer (even with the standard prompt of 8 exs from Wei et al ‘22). E.g.
>
> ```
> Q: Nancy's old washing machine could only wash 9 pieces of clothing at a time. If she had to wash 19 shirts and 8 sweaters how many loads would she have to do?
> A: Nancy's old washing machine could only wash 9 pieces of clothing at a time. So she would have to do 19 shirts in 9 + 9 + 1 loads, and she would have to do 8 sweaters in 8 + 8 loads. So she would have to do 19 + 8 = 27 loads in total.
> COT ⇒ Fails to extract the answer.
> ```
> We use DecomP to decompose the problem into first using CoT to perform the required math reasoning and then using a second GPT3 module (gpt_ans) to extract the answer from the CoT. Here is the sample output using our decomposition:
> ```
> QC: Nancy's old washing machine could only wash 9 pieces of clothing at a time. If she had to wash 19 shirts and 8 sweaters how many loads would she have to do?
> Q1: [cot] Nancy's old washing machine could only wash 9 pieces of clothing at a time. If she had to wash 19 shirts and 8 sweaters how many loads would she have to do?
> #1: Nancy's old washing machine could only wash 9 pieces of clothing at a time. So she would have to do 19 shirts in 9 + 9 + 1 loads, and she would have to do 8 sweaters in 8 + 8 loads. So she would have to do 19 + 8 = 27 loads in total.
> Q2: [gpt_ans] #1
> #2: 27
> Q3: [EOQ]
> ```
> where gpt_ans has examples such as:
> ```
> Q: Michael started with 58 golf balls. After losing 23 on tuesday, he had 58 - 23 = 35. After losing 2 more, he had 35 - 2 = 33 golf balls.
> A: 33
> ```
> We have added these prompts to the appendix too. As we mentioned earlier, this simple change resulted in a 14-17 pt improvement on two datasets. While this is a simple change, it opens up the possibility of using DecomP for other complex answer types, e.g. non-extractive answer generation from CoT.
>
> The other key source of error are mistakes in decomposition due to incorrect question understanding (we ignore math calculation errors which have been fixed  heuristically by using a calculator  in prior work). As we get stronger models with better math reasoning capabilities, it would be trivial to delegate the decomposition sub-task to these special models -- a key advantage of our framework over CoT.

---

> ### Author Response · Authors · 2022-11-19
> **System sensitivity**
>
>
> > how to decide the decomposition scheme and questions? How sensitive is the system?
>
> Our experiments used three prompts with low deviation across results. We have added the results for all the prompts in the appendix (App. D) and show DecomP outperforms CoT on all these prompts.
>
> Regarding the decomposition scheme, we used the most natural decomposition structure in our experiments. It is, of course, possible to write certain decomposition schemes where it is challenging for the model to generalize from the prompt to the test instances. But the key advantage of our work is that such failures are easy to observe and debug. E.g. if the decomposition scheme is too challenging, it will be observed in incorrect decompositions. Similarly, errors in the sub-task agents' answers to questions can be used to detect bad agent design. Additionally each sub-task agent can be unit-tested independent of the complex task and improved by either providing more examples or using more task-appropriate models.
>
> Given your suggestion, we did experiment with two other simple decomposition structures in letter concatenation and reversal.
>
> For letter concatenation, we replaced
> ```
> QS: (foreach) [str_position] What is the last letter in "#1"?
> ```
> with
> ```
> QS: [str_position] What is the last letter in "Augusta"?
> A: "a"
> QS: [str_position] What is the last letter in "Ada"?
> A: "a"
> QS: [str_position] What is the last letter in "King"?
> A: "g"
> ```
> ,i.e., we use GPT3 to generate all the sub-questions from the array answer.
>
>
> For list reversal, instead of splitting into halves, we take the tail of the list, reverse it and then concatenate it to the head. I.e. reverse(list) = reverse(list[1:]) + list[0]. This requires  more GPT3 calls (O(n)) compared to the original approach of splitting with O(ln n) calls.
>
>
> In both these cases, we noticed that the performance did not drop (see new App.E) and assume this will be true as long as the decomposition structure is natural (and not adversarial). On the letter concatenation task, the results were exactly the same.  The new reversal formalism was actually stronger on longer inputs at the cost of more calls to GPT3 (O(ln(n)) using binary splits vs O(n) one element at a time). Both these decomposition schemes are still better than CoT.
>
>
> > smaller LLMs?
>
> Thank you for your suggestion. We added experiments where we modified the model size within the GPT3 family for CommaQA and used different flan-t5 models for 2WikiMultiHopQA (a new dataset. In general, the weaker models do perform worse as expected but the decomposition approach is generally better than CoT (except curie where all the scores are close to zero). We added these numbers to the appendix A and C.

---

### Author Response · Authors · 2022-11-19
**Executive Summary**

We thank all the reviewers for their insightful comments and suggestions. We appreciate the reviewers recognizing the novelty and flexibility of our approach as well as the comprehensiveness of our experiments.

We want to bring attention to few key empirical comparisons that we have added in response to the reviewer comments:

- As suggested by H95Z, we applied DecomP to math qa datasets: GSM8K and MultiArith. By adding a new answer extraction sub-module, we were able to improve the accuracy of GPT3 models by 14-17 pts (see App. B). While this is a simple change, it illustrates the possibility of using DecomP for other complex answer types, e.g. non-extractive answer generation from CoT.
- To further broaden our results, we also added experiments for a new multi-hop QA dataset: 2WikiMultihopQA (App. A). We show that using our decomposed IR approach results in 6 pts improvement over just one-step IR.
- As suggested by H95Z and Pqnp, we evaluated different model sizes on the CommaQA dataset and 2WikiMultihopQA. While weaker models observe a drop in score, DecomP still outperforms CoT across all models (except on the curie model where all approaches scored close to zero).
- In the final version of our paper, we will also add results on the MuSiQue dataset [Trivedi et al '22] containing 2-4 hop questions where our DecomP approach achieves a score of 23.9 compared to the 21.0 of the one-step model. We will also report numbers of the FLAN model on this dataset.

Bib:
[♫ MuSiQue: Multihop Questions via Single-hop Question Composition](https://aclanthology.org/2022.tacl-1.31) (Trivedi et al., TACL 2022)

---

### Decision · Program_Chairs · 2023-01-20

**Decision:**

Accept: poster

**Justification For Why Not Higher Score:**

There were largely mixed feelings about this paper, especially considering its similarity to chain-of-thought prompting and least-to-most prompting.

**Justification For Why Not Lower Score:**

Note: Reviewer WUGg wrote that they plan to change their score to a 6 but hasn't done it despite prodding. So the scores should be 6 6 6 8 with an average of 6.5

**Metareview: Summary, Strengths And Weaknesses:**

This is an interesting paper with a new prompting strategy that decomposes a task into smaller tasks that can also be solved by prompting. It is closely related to but potentially more powerful than chain-of-thought prompting. Experimental results show that this approach improves performance across a variety of tasks.

While reviewers originally felt that the experiments were limited, some further experiments were added. Also, concerns about novelty were addressed in the rebuttal. As a result of the changes made to the paper, reviewers were satisfied with the resulting paper. Some reviewers felt that more analysis and insights would be helpful to make it easier for people to use this prompt decomposition approach.

**Note From Pc:**

if the above contains the word "oral" or "spotlight" please see: "oral" presentation means -> notable-top-5% and "spotlight" means -> notable-top-25%. As stated in our emails, we are disassociating presentation type from AC recommendations

**Summary Of Ac-Reviewer Meeting:**

At the time of scheduling, this was not a borderline paper, but the scores later changed.